

# Coral Skeletal Proxy Records Database for the Great Barrier Reef, Australia

Ariella K. Arzey[1], Helen V. McGregor[1,2], Tara R. Clark[1,3], Jody M. Webster[4], Stephen E. Lewis[5], Jennie Mallela[6], Nicholas P. McKay[7], Hugo W. Fahey[1], Supriyo Chakraborty[8], Tries B. Razak[9,10] and Matt J. Fischer[11]

[1]Environmental Futures, School of Earth, Atmospheric and Life Sciences, University of Wollongong, Wollongong, 2522, Australia
[2]Securing Antarctica's Environmental Future, University of Wollongong, Wollongong, 2522, Australia
[3]Radiogenic Isotope Facility, School of the Environment, The University of Queensland, Brisbane, 4072, Australia
[4]Geocoastal Research Group, School of Geosciences, The University of Sydney, Camperdown, 2006, Australia
[5]Catchment to Reef Research Group, Centre for Tropical Water and Aquatic Ecosystem Research, James Cook University, Townsville, 4811, Australia
[6]Research School of Biology, The Australian National University, Canberra, 2601, Australia
[7]School of Earth and Sustainability, Northern Arizona University, Flagstaff, 86011, USA
[8]Indian Institute of Tropical Meteorology, Ministry of Earth Sciences (MoES), Pune, 411008, India
[9]Department of Marine Science and Technology, Faculty of Fisheries and Marine Sciences, IPB University, Bogor, 16680, Indonesia
[10]School of Coral Reef Restoration (SCORES), Faculty of Fisheries and Marine Science, IPB University, Bogor, 16680, Indonesia
[11]Environment Research and Technology Group, Australian Nuclear Science and Technology Organisation, Lucas Heights, 2234, Australia

*Correspondence to*: Ariella K. Arzey (aka548@uowmail.edu.au)





**Abstract.** The Great Barrier Reef (GBR), Australia has a long history of palaeoenvironmental coral research. However, it can
be logistically difficult to find the relevant research and records, which are often unpublished or exist as 'grey literature'. This
hinders researchers' ability to efficiently assess the current state of coral core studies on the GBR and thus identify any key
knowledge gaps. This study presents the Great Barrier Reef Coral Skeletal Records Database (GBRCD), which compiles 208
records from coral skeletal research conducted since the early 1990s. The database includes records from the Holocene, from
~8,000 years ago, to the present day; from the northern, central, and southern GBR from inshore and offshore locations.
Massive *Porites* spp. coral records comprise the majority (92.5 %) of the database, and the remaining records are from
*Acropora*, *Isopora* or *Cyphastrea* spp. The database includes 78 variables, with Sr/Ca, U/Ca and Ba/Ca the most frequently
measured. Most records measure data over 10 or more years and are at monthly or lower resolution. The GBRCD is machine
readable and easily searchable so users can find records relevant to their research, for example, by filtering for site names, time
period, or coral type. It is publicly available as comma-separated values (CSV) data and metadata files with entries linked by
the unique record ID and as Linked Paleo Data (LiPD) files. The GBRCD is publicly available from the NOAA National
Center for Environmental Information's Paleoclimate Data Archive at https://doi.org/10.25921/hqxk-8h74 (Arzey et al. 2024).
The collection and curation of existing GBR coral research provides researchers with the ability to analyse common proxies
such as Sr/Ca across multiple locations and/or examine regional to reef scale trends. The database is also suitable for multi-
proxy comparisons and combination or composite analyses to determine overarching changes recorded by the proxies. This
database represents the first comprehensive compilation of coral records from the GBR. It enables the investigation of multiple
environmental factors via various proxy systems for the GBR, northeastern Australia and potentially the broader Indo-Pacific.

## 1 Introduction

Scleractinian coral skeletons have long been used to reconstruct past changes of reef environments and climates, and to
understand reef responses to those changes (e.g. Druffel & Griffin 1993; Webster et al. 2018; Thompson 2022; Clark et al.
2017; Felis et al. 2014). Foundational to the field are studies on corals from the World Heritage listed Great Barrier Reef
(GBR). Studies on GBR corals have established, for example, the relationships between coral skeletal Sr/Ca and $\delta^{18}O$ with sea
surface temperature (SST) (Weber & Woodhead 1969; Weber & Woodhead 1970; Weber & Woodhead 1972; Weber 1973;
McCulloch et al. 1994), coral luminescence with riverine inputs (Isdale 1984; Boto & Isdale 1985; Isdale et al. 1998), coral
radiocarbon with oceanographic processes (Druffel & Griffin 1993), the environmental signature recorded in coral density
bands (Lough & Barnes 1990; Lough & Barnes 1997), evidence of sediment and nutrient exposure (McCulloch et al. 2003;
Wyndham et al. 2004; Sammarco et al. 1999) and a host of new analytical methods and approaches (Gagan et al. 1998; Sinclair
et al. 1998; Barnes et al. 2003). In addition, palaeoclimate and palaeoenvironmental research based on GBR corals (typically
massive *Porites* spp.) have advanced our knowledge on the most pressing threats to the GBR and coral reef ecosystems more
broadly (McCulloch et al. 2003; De'ath et al. 2009; Lough et al. 2015; Wei et al. 2009; Koop et al. 2001), tropical climate
processes (Hendy et al. 2002; Lough 2007; Druffel & Griffin 1993), reef responses to climate and sea-level change (Webster
et al. 2018; Sanborn et al. 2020; Yokoyama et al. 2011; Leonard et al. 2020), and coral bleaching (Suzuki et al. 2003; D'Olivo
& McCulloch 2017; De'ath et al. 2012).

Despite the long history of coral palaeoclimate and environmental research on the GBR, like other research fields, GBR coral
research data are being lost to time (Vines et al. 2014). While there is a recent push to ensure that coral geochemical records
are publicly available upon publication (Kaufman & team 2018; Khider et al. 2019; Dassié et al. 2017), many GBR coral core
records are not yet publicly archived, and thus we risk losing these valuable datasets. The need to publicly archive GBR coral
geochemical data is even more pressing because corals are under threat from human-induced stresses (Hughes et al. 2017;
Maynard et al. 2015; Guan et al. 2020; Ortiz et al. 2018). Long term monitoring of the GBR indicates reef-wide hard coral
cover was down to 18 % in 2017 (Australian Institute of Marine Science 2017). Since then hard coral cover has increased, but



recovery has primarily been driven by fast growing *Acropora* species (Australian Institute of Marine Science 2017, 2022), which are less suitable for producing continuous multi-century records. The future of GBR corals, and therefore the valuable geochemical records they contain, faces an uncertain future, which makes existing GBR records a precious and potentially finite resource.

This study presents a comprehensive database of GBR coral records, compiling both published (201) coral geochemical and
luminescence records and records published here for the first time (7). The GBR Coral Skeletal Records Database (GBRCD) transfers data into a standardised machine-readable format (structured using a consistent logic readable by and between machines) and makes the records available to researchers in a searchable database. The database consists of two main parts: 1) individual comma-separated values (CSV) files consisting of a single coral's geochemical and luminescence measurements and the associated age estimate, and 2) a metadata file for the entire suite of corals, which includes details of location, species,
analytical techniques, publications, cross-references, and other details. Alternatively, the GBRCD is available as Linked Paleo Data (LiPD) files and serialisations in Python, MATLAB and R, which includes the same fields standardised to the LiPD format and terminology. Basic analysis of the scope of the database is presented, as is a discussion of database caveats and future use. The GBRCD addresses both the potential loss of existing GBR coral data and provides the ability to easily explore what records exist. The GBRCD is a valuable resource for exploratory analysis to understand the GBR (and the wider Pacific)
environment (e.g. Henley et al. In press) and preserves the records for future use.

## 1.1 GBR Setting

The GBR is made up of more than 3,000 reefs that span 14 degrees of latitude, and the World Heritage Area comprises an area of 348,000 km$^2$. The width of the GBR across the continental shelf changes with latitude, ranging from 50 to 250 km wide
(Steinberg 2007; Hopley et al. 2007). Thus, coral reefs in the GBR can be classified based on their proximity to the Queensland coast and sea floor depth, as either inner-shelf, mid-shelf, or outer-shelf reefs and, depending on their classification, these reefs will experience various gradients of terrigenous or open ocean influences. There is a seasonal atmospheric circulation that is characterised by an annual summer monsoon (December to March), which brings rainfall and increased river discharge to neighbouring GBR marine waters.

Average sea surface temperature (SST) varies latitudinally, with a 1.65°C difference between the northern region (11-17° S) and the southern region (20–24° S)(Fig. 1) for the 1981–2010 period (HadISST; Rayner et al. 2003). The timing of the annual minimum and maximum also varies slightly; as mean maximum SST in the 1981–2010 period (HadISST1.1) most frequently occurs in February in the central and southern GBR, and in January in the northern GBR, while minimum SSTs most frequently occur in August in the northern and southern GBR and in July in the central GBR.

The modern GBR coral reef ecosystem initiated ~8–9 thousand years ago (ka) following the Holocene marine transgression (Davies & Hopley 1983; Davies et al. 1985; Dechnik et al. 2015; Hopley et al. 2007), although the timing of individual coral reef initiation and vertical accretion rates varied considerably (range of 1–16 m/1000 years)(Dechnik et al. 2017; Sanborn et al. 2020; Davies & Hopley 1983; Hopley et al. 2007; Leonard et al. 2020; Ryan et al. 2018). Decreases in reef growth rate occurs where the reef surface is within ~2–3m of sea level due to reduced growth and/or erosional loss (Davies & Hopley
1983). Evidence from reef drilling suggests that there was little latitudinal variation in reef growth rate throughout the GBR, although the depth to the Pleistocene reef foundations in northern and southern GBR regions is shallower than the central region (e.g. Davies et al. 1985; Hopley et al. 2007). Additionally, there are reef-flat growth hiatuses in both the northern and southern GBR, but not in the central GBR, due to a relative fall in sea level of ~0.5 m in the other regions that was not evident in the central GBR due to subsidence in the central region (Halifax Basin) from hydro-isostatic adjustment (Dechnik et al.
105 2017).

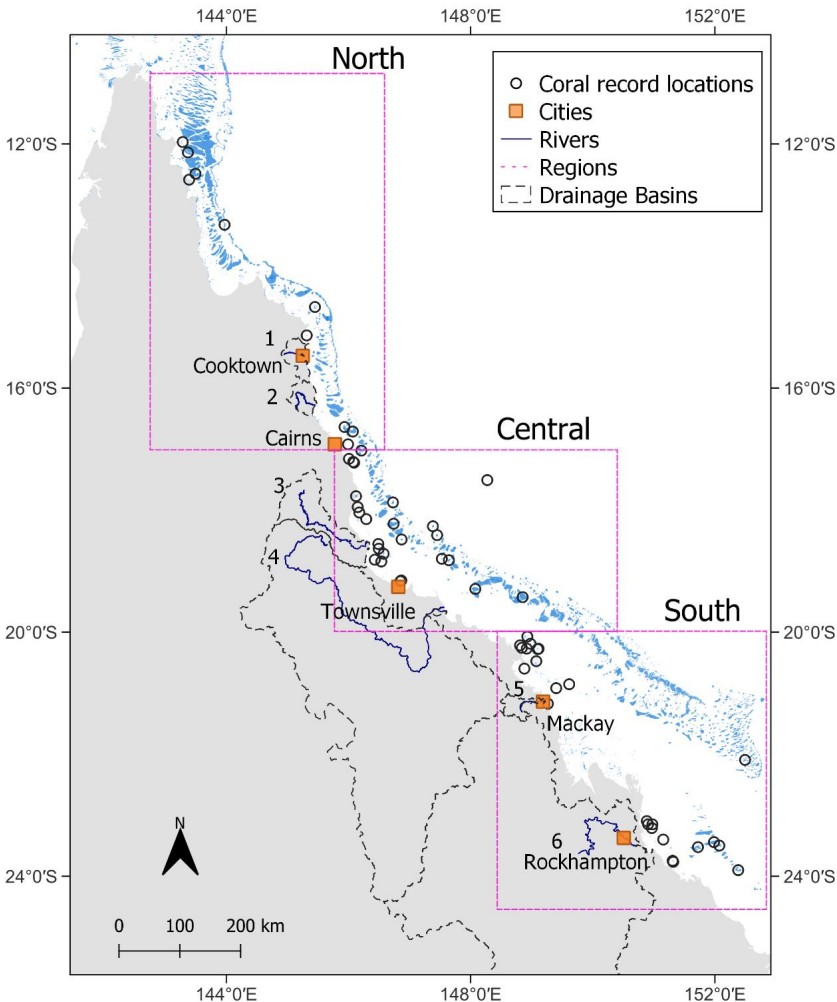

**Figure 1. Map of locations with coral records in the Great Barrier Reef Coral Skeletal Records Database. Record locations are indicated by circles, select cities are indicated by orange squares, select rivers are indicated by dark blue lines, reefs are in light blue and the nominal regions of the north, central and south GBR are indicated by the dashed pink lines. Numbers identify the major rivers; 1 – Endeavour River, 2 – Daintree River, 3 – Herbert River, 4 – Burdekin River, 5 – Pioneer River and 6 – Fitzroy River. The drainage basin boundary associated with the included rivers is indicated by the dashed black lines. GIS layers © State of Queensland (Department of Resources) 2023. This material is licensed under a Creative Commons - Attribution 4.0 International licence. See Appendix G for full list of included GIS layers.**

## 1.2 Palaeoclimate and palaeoenvironment reconstructed from GBR coral skeletons

Research on the GBR has a long history of discovery and scientific development, with ventures as far back as the 1928–1929 GBR expedition lauded as having promoted the development of empirical and analytical approaches important to the foundation of modern coral reef science (Spencer et al. 2021). Coral skeletons provide in situ records of the reef environment as they incorporate trace elements, isotopes, and organic materials in proportion to climate and environmental variations in the marine environment. These palaeoclimate and palaeoenvironment 'proxies' are a long-established method to assess the reef

environment in the past and present, and provide a means to quantify relative changes over space and time. Proxy records from coral skeletons offer the ability to extract high-resolution (weekly–monthly) information to assess a range of climate and environmental variables, most commonly SST and rainfall, and at variety of scales, for example near-weekly records that





extend from the present day back several centuries prior to the instrumental record (e.g.Lewis et al. 2012; D'Olivo & McCulloch 2022).

Beginning in the late 1960s, studies using GBR corals provided empirical evidence of a relationship between skeletal oxygen isotopes, luminescence, trace elements, radiocarbon and density and the coral's marine environment (Weber & Woodhead 1969; Weber & Woodhead 1970; Weber & Woodhead 1972; Isdale 1984; Druffel & Griffin 1993; Lough & Barnes 1990; McCulloch et al. 2003) and were at the forefront of the development of new technology and methods (Isdale et al. 1998; Gagan et al. 1998; Sinclair et al. 1998; Barnes et al. 2003; D'Olivo et al. 2018).

After the description of the utility of Sr/Ca as an SST proxy by Beck et al. (1992), coral skeleton proxy research in the GBR focused on development and calibration of proxy-SST relationships in the 1990s. GBR coral records from this period are generally short (<10 years) and were analysed using a variety of spectrometry methods (Alibert & McCulloch 1997; Druffel & Griffin 1993, 1999; McCulloch et al. 1994; Gagan et al. 1998; Sinclair et al. 1998). Research in the 1990s also included the first application of paired Sr/Ca and $\delta^{18}$O proxies to reconstruct the GBR palaeoenvironment (McCulloch et al. 1994; Gagan

et al. 1998) and the further exploration of coral luminescence as a proxy for river discharge (Isdale et al. 1998). These approaches were applied to extend our knowledge of GBR SST, riverine input and oceanographic variability prior to the 1800s (Lough 2007, 2011a; Hendy et al. 2002; Hendy et al. 2003a; Calvo et al. 2007), and during the Holocene (Gagan et al. 1998; Leonard et al. 2016; Sadler et al. 2016b; Lough et al. 2014; Roche et al. 2014) .

Since the 2000s, partially facilitated by the development of the Australian Institute of Marine Science luminometer and the

Australian National University laser ablation methods, research on coral proxies in the GBR has focused on proxies for terrigenous inputs. These include indicators of terrestrial runoff and sediment exposure (e.g. Ba/Ca and rare earth elements (REEs) (Jupiter et al. 2008; Jupiter 2008; Saha et al. 2021; McCulloch et al. 2003; Lewis et al. 2012; Leonard et al. 2019)), land-use changes (e.g. $\delta^{15}$N and Mn (Lewis et al. 2007; Lewis et al. 2012; Erler et al. 2016; Erler et al. 2020; Marion et al. 2021)) and river flow patterns (e.g. Ba/Ca and luminescence (Lough et al. 2002; Lough 2007; Lough et al. 2014; D'Olivo &

McCulloch 2022)). Additionally, there is continuing research that aims to explore new proxies (e.g. V/Ca (Saha et al. 2019a), $\delta^{98}$Mo (Wang et al. 2019)) as well as to develop proxies relevant to global issues such as ocean acidification (e.g. $\delta^{11}$B (Pelejero et al. 2005; Wei et al. 2009; D'Olivo et al. 2015; McCulloch et al. 2017)).

## 2 Overview of the GBR Coral Skeletal Records Database

### 2.1 Selection criteria for inclusion in GBRCD

The GBRCD was not assembled around a specific research goal unlike other coral palaeoclimate databases, such as the CoralHydro2k database (Walter et al. 2023), which was designed to support "the project's goal of reconstructing tropical hydroclimatic variability at seasonal and longer timescales". The GBRCD is along the lines of the Coral Trait Database (Madin et al. 2016), but focuses on a specific region and coral proxy measurements (i.e. geochemical and luminescence measurements). Therefore, only three broad selection criteria are considered for including a record in the GBRCD:

▪ Location. The GBRCD compiles coral records from the Great Barrier Reef and, due to its regional context and latitudinal relevance, Flinders Reef.

▪ Length. The records included were continuous (multi-year) measurements from coral skeletons.

▪ Age model. Each record needed to include relevant chronology (i.e. an estimate of age as the independent variable) or have a chronology that could be recreated relatively easily based on available data.

All available geochemical or luminescence data were included in the GBRCD if they met these three criteria. No screening was applied on the quality of the data and no records were excluded based on analytical uncertainties, diagenesis screening, resolution (e.g. weekly, monthly, annual), significance of the correlation between proxy and target parameter or record length. It is at the discretion of researchers using data from the GBRCD to determine suitability for their aims.





## 2.2 Data sources

The majority of GBR coral records are from peer-reviewed published studies (see Appendix Table A1). In many cases, the published records were publicly archived in repositories such as the NOAA National Centers for Environmental Information (NCEI) World Data Service for Paleoclimatology and World Data Center PANGAEA. The GBRCD includes digital object identifier (DOI) information linking back to these original sources and publications. In some cases, published studies included additional records that were not archived in public repositories. Instead, these data were obtained from the publication's

supplemental material, or by reaching out to the corresponding authors.

Seven records have not been published previously and are published in the GBRCD for the first time (Table 1 & Appendix A & C). These records are accompanied by their associated metadata record and appropriate background information (see Appendix A, C, D). If further information about the record is required, users are encouraged to reach out to the first author/contact listed in the GBRCD publication metadata.


**Table 1. Records first published in the GBRCD.**

| Record ID | Variables | Reef Location |
|---|---|---|
| AR24DIP01 | Ba/Ca; Mg/Ca; Sr/Ca; U/Ca; Y/Ca | Dip Reef |
| AR24OTI01 | Sr/Ca | One Tree Island |
| AR24OTI02 | Ba/Ca; B/Ca; Ca/Ca; Mg/Ca; Sr/Ca; U/Ca | One Tree Island |
| AR24OTI03 | Ba/Ca; B/Ca; Ca/Ca; Mg/Ca; Sr/Ca; U/Ca | One Tree Island |
| AR24OTI04 | Ba/Ca; B/Ca; Ca/Ca; Mg/Ca; Sr/Ca; U/Ca | One Tree Island |
| AR24OTI05 | Ba/Ca; B/Ca; Ca/Ca; Mg/Ca; Sr/Ca; U/Ca | One Tree Island |
| AR24SLY01 | $\delta^{13}C$; $\Delta^{14}C$; $\delta^{18}O$ | Stanley Reef |

Records included in the database are multi-year measurements of coral geochemistry and luminescence signals. In total, there are 78 variables available to investigate the GBR environment, or to determine new proxy-environment relationships (Table

2). The records include commonly measured isotopes and trace elements (TE) such as $\delta^{18}O$, Ba/Ca, Sr/Ca, and U/Ca, as well as less explored isotopes (such as $\delta^{11}B$ and $\delta^{15}N$) and emerging TE proxies such as Li/Mg and rare earth elements (REEs). Luminescence records are included in the database as coral luminescence is a proven hydrological proxy in the GBR (Lough 2011, Lough et al. 2014).

Care is advised for the use of all records in the GBRCD as the relevance of the measured variables may vary by location, or

the sampling resolution of the records may not be suitable. Therefore, further processing needs be considered before integrating or comparing the records.

## 2.3 Database structure

The database is provided in a split form and includes a table of metadata describing each record, as well as individual CSV files that include the coral variables such as age, distance down core, measured trace elements and so on. It is also provided as

LiPD serialisations with the metadata fields structured as per the LiPD format (see Table B1). The database is available in both formats to facilitate ease of use. The CSV files are easily accessible and do not need programming languages (e.g. R or python) to explore the datasets, whereas the LiPD files require programming languages to fully access and is designed as structured data to enable interoperability and quick analysis of palaeoclimate data (McKay & Emile-Geay 2016).

The metadata supplied alongside the coral record data provides a thorough background on each record, with assumptions and

limitations made clear for researchers. While every best attempt has been made to provide information for all metadata fields



that are relevant, there are some fields where the information was not available (i.e. flagged with NA). Metadata fields allow for filtering, for example, by: site, coral species, analysis method, proxy type, etc.

Individual CSV and LiPD files are provided for each coral record. In general, an individual record is defined as being from a single coral, whether a coral head was drilled directly, or the coral was retrieved from a core drilled into coral reef matrix

(coral reef matrices can intersect individual coral heads as well as the reef framework in which they are preserved).

In some instances, multiple geochemical and/or luminescence records have been measured from the same coral head. In this situation, the multiple records have their own separate entries as they are usually either part of an included composite record, measured using different techniques, or have sufficient differences in the metadata or dataset to provide a good reason not to attempt to join the datasets. However, they are identified and cross-referenced in the database as being from the same coral or

core.

## 2.4 Data processing

Published data, as was available or supplied, have been archived as per the database format described here. Potentially abnormal data (e.g. values <0 for Mg/Ca) were not removed from the datasets as they may be indicative of climate or environmental events, and it is up to the user to determine appropriate screening procedures when using the database.

A basic quality check was conducted on records published for the first time in the GBRCD (see Appendix C & F). This consisted of creating or adjusting the coral age model based on available information and checking for diagenetic alteration where there was a physical sample available.

Coral records that were archived or supplied with the ages in standard date format (e.g. DD/MM/YYYY) had all ages converted to decimal date in Common Era (CE)/Before Common Era (BCE) format. This means that there is no year zero, and negative

(BCE) numbers start at "-1". Due to rounding issues with decimal dates assigned to the first day of the month (at midnight), 12 hours were added to each date to ensure the age remained assigned to the correct month of the year.

For archiving purposes, 16 records were assigned a new age model. This was primarily because the coral data were not available with an age model, or did not have one appropriate for archiving (e.g. date not assigned to each data point). A note is included in the relevant field about the chronology for records where the chronology was (re)created for archiving in the

GBRCD and further details are provided in Appendix C. Nine fossil records were supplied or previously archived with chronologies relative to the record itself (i.e. 0 = first or youngest year of record). The available dating information was used to transform these chronologies to Common Era ages.

Before Common Era fossil coral data are included in the GBRCD. Absolute age estimates for these corals are based on uranium-thorium (U-Th, or U-series) dating and radiocarbon ([14]C) measurements, with basic information about the dating

supplied in the GBRCD metadata. Five coral records dated by radiocarbon measurements were re-calibrated using CALIB rev. 8 (Stuiver & Reimer 1993) and the Marine20 radiocarbon age calibration curve (Heaton et al. 2020) to standardise the age estimates and details (including the local delta-R reservoir correction) are available in the dating_notes field. Users of the GBRCD should determine if the radiocarbon ages of the fossil records should be revised based on future updates to the calibration curves and/or reservoir ages. For records where U-Th dating information is reported for the first time in this study,

information is provided in the GBRCD metadata and complete relevant U-Th dating information (Dutton et al. 2017) is included in the Appendix (Table D1).

Relevant metadata was obtained from source publications, and where necessary subsequent or cited publications, and data archives. Coral IDs were also matched across publications using literature searches, the AIMS reef core database, and within the GBRCD.





## 2.5 Database Records and Identification

Records are identified here as datasets from a single coral that may include one or more measured variables unified with a single age model.

Similar to the Iso2k database protocols (Konecky et al. 2020), each record in the database was given a nine-digit alphanumeric ID. The dataset ID was created from the first two letters of the first author's last name, the last two numbers of the year of publication, the first three letters of the location (or in cases where appropriate, a commonly accepted shortening or three-letter acronym for the location was used), and two numbers that represent the core number. If data were not previously published, publication year was substituted with the year of this compilation. For reef locations in the GBR that are currently represented by a number rather than a name, and for records that are the composite of multiple locations, 'GBR' has been used as the three-letter reef location for core naming purposes.

However, the Iso2k nine-digit alphanumeric identifier was insufficient to separate records where researchers have published multiple papers from the same GBR location within the same year. To separate these records a 10th digit was added to the ID, adding an alphabetical end code of 'a', 'b' and so on following the order of publication (Table 2), similar to many journal standards for uniquely identifying references. Additionally, where the core record has been split due to notable differences in the record (for example due to data resolution or age model), a split identifier of '_1' or '_2' has been appended to the end of the identifier (Table 2).

**Table 2. Example of process of giving core records a unique identifier.**

| GBRCD ID | Core ID from publication | Publication Reference | Location | Variables | Resolution | Split | Comments |
|---|---|---|---|---|---|---|---|
| SA16HER01a | AR HL 3 | Sadler et al. (2016a) | Heron Reef | Sr/Ca | Biannual | Core 1 | Suffix identifier for record as different coral(s) with publication(s) in the same year at the same reef location. |
| SA16HER01b | P HS 1 | Sadler et al. (2016b) | Heron Reef | Sr/Ca | Bimonthly | Core 1 | Suffix identifier for record as different coral(s) with publication(s) in the same year at the same reef location. |
| WE09ARL01_1 | AREO 4 | Wei et al. (2009) | Arlington Reef | Ba/Ca; $\delta^{11}$B; $\delta^{13}$C; $\delta^{18}$O; Mg/Ca; Sr/Ca; $\delta^{11}$B pH | Annual | Core 1, split 1 | Record split due to difference in measurement resolution. |
| WE09ARL01_2 | AREO 4 | Wei et al. (2009) | Arlington Reef | Ba/Ca; $\delta^{11}$B; $\delta^{13}$C; $\delta^{18}$O; Mg/Ca; Sr/Ca; $\delta^{11}$B pH | >Annual | Core 1, split 2 | Record split due to difference in measurement resolution. |



## 2.6 Database Variables, Units and Standards

Records are all standardly formatted as CSV and LiPD files, with each record containing an age field and variable field(s) as
per Table 3. Each GBRCD record includes between 1 and 33 variables.

The database collates 78 variables obtained from coral records: 72 measured and 6 calculated variables (Table 4).

Units reported in the database were standardised to enable interoperability between records. All trace element/calcium (TE/Ca) ratios were included as mmol/mol and, where relevant, the calibrations (proxy-SST) for the ratios included in the record metadata are in the same units. Isotopes were archived as permille (‰), and, where relevant, the calibrations for the ratios are
in the same units. The majority of non-ratio elements were archived as parts per million. The exceptions to this were the rare earth elements (REEs) (lanthanum (La), cerium (Ce), praseodymium (Pr), neodymium (Nd), samarium (Sm), europium (Eu), gadolinium (Gd), terbium (Tb), dysprosium (Dy), holmium (Ho), erbium (Er), thulium (Tm), ytterbium (Yb), and lutetium (Lu)), yttrium (Y) and zirconium (Zr), which were archived as parts per billion (i.e. $10^9$).

The list of included dataset field names for each record is provided in the metadata file (i.e. meths_primaryVariableList or
meths_additionalVariableList; Table 4).

All records include an Age field that is the age of each sample as determined by the coral age model. For records that are at greater than annual resolution the age supplied is representative of the grouping of years included in each data point. For example, the Druffel & Griffin (1999) biennial data lists the middle year of the sampled coral (i.e. October 1889 to October 1891 has the age 1890.8, while Hendy et al. (2002) quinquennial resolution data lists the middle year of the sampled coral (i.e.
1981 to 1985 has the age 1983).

**Table 3. Great Barrier Reef Database record data format. Describes the format of the CSV files of each record.**

| Field name | Variable | Type | Description |
|---|---|---|---|
| Age | Age | numeric | Time data for the record. All time data are expressed as a decimal date. Annual or greater resolution default to [year] (equivalent of [year].0) unless original publication specified otherwise. Units: CE |
| [variable] | Data | numeric | A Nx1 vector of proxy data. Data type is specified from list of proxy and additional variables listed in Table 4. Units are specified in Table 4. |

**Table 4. Data variables. Describes variables included in the database, units, type, and brief description. The list of variables is alphabetical.**

| Field name | Variable | Type | Units | Unit abbreviation | Description |
|---|---|---|---|---|---|
| Arag | $\Omega_{arag}$ | calculated | dimensionless | | Aragonite saturation state determined from calculated $\delta^{11}$B pH. |
| Ba | Ba | measured | parts per million | ppm | Barium |
| Ba138Ca | Ba/Ca | measured | millimole/mole | mmol/mol | Barium/Calcium A secondary Ba/Ca measurement for laser ablation. This is a measure of the total Ba/Ca using the count intensities of the Ba$^{138}$ isotope. |



| Field name | Variable | Type | Units | Unit abbreviation | Description |
|---|---|---|---|---|---|
| BaCa | Ba/Ca | measured | millimole/mole | mmol/mol | Barium/Calcium. For laser ablation this is a measure of the total Ba/Ca using the count intensities of the Ba[137] isotope. |
| BCa | B/Ca | measured | millimole/mole | mmol/mol | Boron/Calcium |
| BMg | B/Mg | measured | millimole/mole | mmol/mol | Boron/Magnesium |
| Ca | Ca | measured | parts per million | ppm | Calcium |
| CaCa | Ca/Ca | measured | millimole/mole | mmol/mol | A measure of the total Ca/Ca ratio using the count intensities of the Calcium[48]/Calcium[43] isotopes. |
| Calcn | Calcification rate | calculated | grams/centimetre/year | g/cm/yr | Annual coral calcification rate. |
| Cd | Cd | measured | parts per million | ppm | Cadmium |
| CdCa | Cd/Ca | measured | millimole/mole | mmol/mol | Cadmium/Calcium |
| Ce | Ce | measured | parts per million | ppm | Cerium (REE) |
| CeCa | Ce/Ca | measured | millimole/mole | mmol/mol | Cerium/Calcium |
| CeCe_anom | Ce/Ce* | calculated | dimensionless[a] | | Cerium anomaly calculated[b] as: $$\left(\frac{Ce}{Ce}\right) = \left(\frac{Ce_{sn}}{Pr_{sn} \times \left(\frac{Pr}{Nd}\right)_{sn}}\right) (1)$$ |
| d11B | $\delta^{11}B$ | measured | permille | permil (‰) | $^{11}B/^{10}B$ |
| d11B_pH | $\delta^{11}B$ pH | calculated | dimensionless | | pH calculated from $\delta^{11}B$. |
| d11B_pHcf_pres | $\delta^{11}B$ pH$_{cf}$ | calculated | dimensionless | | pH of calcifying fluid calculated from $\delta^{11}B$. Values are pressure corrected. |
| d13C | $\delta^{13}C$ | measured | permille | permil (‰) | $^{13}C/^{12}C$ |
| D14C | $\Delta^{14}C$ | measured | permille | permil (‰) | $\delta^{14}C$ ($^{14}C/^{12}C$) deviation from a standard and corrected for $\delta^{13}C$. |
| d15N | CS-$\delta^{15}N$ | measured | permille | permil (‰) | $^{15}N/^{14}N$. $\delta^{15}N$ of skeleton bound organic matter. |
| d18O | $\delta^{18}O$ | measured | permille | permil (‰) | $^{18}O/^{16}O$ |
| d18Osw | $\delta^{18}O_{seawater}$ | calculated | permille | permil (‰) | $\delta^{18}O$ of seawater calculated from paired $\delta^{18}O$ and Sr/Ca. |
| d66Zn | $\delta^{66}Zn$ | measured | permille | permil (‰) | $^{66}Zn/^{64}Zn$ |
| d98Mo | $\delta^{98}Mo$ | measured | permille | permil (‰) | $^{98}Mo/^{95}Mo$ |
| Density | Density | measured | grams/centimetre | g/cm | Annual average coral density. |
| Distance | Distance | measured | millimetres | mm | Distance measured from outer edge of coral or core. |
| Dy | Dy | measured | parts per billion | ppb | Dysprosium (REE) |
| Er | Er | measured | parts per billion | ppb | Erbium (REE) |



| Field name | Variable | Type | Units | Unit abbreviation | Description |
|---|---|---|---|---|---|
| Eu | Eu | measured | parts per billion | ppb | Europium (REE) |
| EuCa | Eu/Ca | measured | millimole/mole | mmol/mol | Europium/Calcium |
| Extn | Extension rate | measured | millimetres/year | mm/yr | Annual linear extension rate of coral. |
| Fe | Fe | measured | parts per million | ppm | Iron |
| FeCa | Fe/Ca | measured | millimole/mole | mmol/mol | Iron/Calcium |
| Gd | Gd | measured | parts per billion | ppb | Gadolinium (REE) |
| GdCa | Gd/Ca | measured | millimole/mole | mmol/mol | Gadolinium/Calcium |
| Ho | Ho | measured | parts per billion | ppb | Holmium (REE) |
| La | La | measured | parts per billion | ppb | Lanthanum (REE) |
| LaCa | La/Ca | measured | millimole/mole | mmol/mol | Lanthanum/Calcium |
| LiCa | Li/Ca | measured | millimole/mole | mmol/mol | Lithium/Calcium |
| LiMg | Li/Mg | measured | millimole/mole | mmol/mol | Lithium/Magnesium |
| Lu | Lu | measured | parts per billion | ppb | Lutetium (REE) |
| LuCa | Lu/Ca | measured | millimole/mole | mmol/mol | Lutetium/Calcium |
| Lumin | Luminescence | measured | dimensionless | | Coral luminescence as measured by luminometer. |
| LuminGB | Luminescence G/B | measured | dimensionless | | Luminescence calculated from spectral green/blue ratio measured by spectral luminescence scanning. |
| LuminInd | Luminescence visual indices | measured | dimensionless | | Annual coral luminescence indices as determined by visual assessment of annual luminescence lines. |
| LuminRange | Luminescence range | measured | dimensionless | | Annual luminescence range as measured by luminometer; determined by difference between annual (summer) maximum luminescence and preceding annual (winter) minimum luminescence. |
| LuminSF | Luminescence sum | measured | dimensionless | | Standardised annual fluorescence/luminescence calculated by interpolating and summing 12 values between annual minimum (winter) luminescence values. |
| MgCa | Mg/Ca | measured | millimole/mole | mmol/mol | Magnesium/Calcium |
| Mn | Mn | measured | parts per million | ppm | Manganese |
| MnCa | Mn/Ca | measured | millimole/mole | mmol/mol | Manganese/Calcium |
| Mo | Mo | measured | parts per million | ppm | Molybdenum |
| Nd | Nd | measured | parts per billion | ppb | Neodymium (REE) |



| Field name | Variable | Type | Units | Unit abbreviation | Description |
|---|---|---|---|---|---|
| NdYb | Nd/Yb | measured | dimensionless[a] | | Neodymium/Ytterbium (REE) |
| P | P | measured | parts per million | ppm | Phosphorus |
| PbCa | Pb/Ca | measured | millimole/mole | mmol/mol | Lead/Calcium |
| PCa | P/Ca | measured | millimole/mole | mmol/mol | Phosphorus/Calcium |
| Pr | Pr | measured | parts per billion | ppb | Praseodymium (REE) |
| PrCa | Pr/Ca | measured | millimole/mole | mmol/mol | Praseodymium/Calcium |
| REEsCa | ΣREE/Ca | measured | dimensionless[a] | | Sum of REEs/Calcium |
| Sm | Sm | measured | parts per billion | ppb | Samarium (REE) |
| SmCa | Sm/Ca | measured | millimole/mole | mmol/mol | Samarium/Calcium |
| SrCa | Sr/Ca | measured | millimole/mole | mmol/mol | Strontium/Calcium |
| Tb | Tb | measured | parts per billion | ppb | Terbium (REE) |
| Th | Th | measured | parts per million | ppb | [232]Thorium |
| Ti | Ti | measured | parts per million | ppm | Titanium |
| Tm | Tm | measured | parts per billion | ppb | Thulium (REE) |
| UB | U/B | measured | millimole/mole | mmol/mol | Uranium/Boron Calculated from U/Ca and B/Ca. |
| UCa | U/Ca | measured | millimole/mole | mmol/mol | Uranium/Calcium |
| USr | U/Sr | measured | millimole/mole | mmol/mol | Uranium/Strontium Calculated from U/Ca and Sr/Ca. |
| VCa | V/Ca | measured | millimole/mole | mmol/mol | Vanadium/Calcium |
| Y | Y | measured | parts per billion | ppb | Yttrium (REY) |
| Yb | Yb | measured | parts per billion | ppb | Ytterbium (REE) |
| YbCa | Yb/Ca | measured | millimole/mole | mmol/mol | Ytterbium/Calcium |
| YCa | Y/Ca | measured | millimole/mole | mmol/mol | Yttrium/Calcium |
| YHo | Y/Ho | measured | parts per billion | ppb | Yttrium/Holmium as mass ratio. |
| Zn | Zn | measured | parts per million | ppm | Zinc |
| ZnCa | Zn/Ca | measured | millimole/mole | mmol/mol | Zinc/Calcium |
| Zr | Zr | measured | parts per billion | ppb | Zirconium |

a – data considered dimensionless (pers. comm. N. Saha)

b – Ce/Ce* formula (Eq. (1)) from Saha et al. (2019b) per Lawrence et al. (2006)

## 2.7 Coral Database Metadata

The GBRCD contains 103 metadata fields in the CSV version (reduced or combined into 75 fields in the LiPD version; Appendix Table B1) that enable identification, and investigation of the coral records. Information for the metadata fields have been sourced from publications, supplementary materials, data archive information and personal communications with researchers.

Nine fields are notes or free-form text fields that provide useful information for future investigations and understanding the record that cannot be succinctly reduced to controlled vocabulary and/or a numerical value.

The metadata included in the GBR coral record database was divided into six categories (Identifier, Geographic, Publication, Analysis, Calibration and Dating) based on best practice standards suggested by Marine Annually Resolved Proxy Archives



(MARPA) (Dassié et al. 2017), Paleoclimate Community reporTing Standard (PaCTS) 1.0 (Khider et al. 2019), Paleoenvironmental Standard Terms (PaST) Thesaurus (Morrill et al. 2021) and implemented in other palaeo-archive databases such as Iso2k and CoralHydro2k (Walter et al. 2023; Konecky et al. 2020).

**2.7.1 Identifier Metadata**

Identifier fields were used to distinguish and characterise each coral record and identify potential crossover (or duplication) between records in the database and other research (Table 5). Identifier fields include core names, (as specified in the original publication(s)) core collection time, coral archive species and the record period (max and min year). The metadata for the record period aims to maximise users' ability to find relevant records that may only include measurements for part of a year,

for example, the max year for records ending in March 2000 or November 2000 would be 2000 for both.

The database includes 67 duplicate core names, which generally indicate different types of analyses on the same coral head, reflecting the development and use of new analysis techniques. For example, TE/Ca may be measured from the same core using different methods (e.g. inductively coupled plasma mass spectrometer (ICP-AES) and laser ablation inductively coupled mass spectrometer (LA-ICP-MS)) or may be measurements of different variables (e.g. TE/Ca and luminescence). However, it

also demonstrates the concentration of analyses on particular coral heads. For example, five of the 21 records from Magnetic Island in the central GBR are analyses from MAG01D and include an: ~weekly resolution multi-trace element LA-ICP-MS record, ~annual resolution stable isotope record, biennial resolution multi-trace element inductively coupled plasma mass spectrometer (ICP-MS) record, quinquennial resolution stable isotope and trace element record, and annual resolution luminescence record.

A notes field (cdata_coralNotes; Table 5) is also included for additional information on the coral, including source of latitude and longitude data (if not from record publication), issues and observations for consideration if using the record.

**Table 5. Identifier metadata for the CSV version of the GBRCD for each coral record including database ID, publication core name, species, and time span.**

| Field name | Variable | Type | Description |
|---|---|---|---|
| cdata_datasetID | Dataset ID | text | A specific identifier assigned to all records, with specifics for the publication and site. |
| cdata_coreName | Core name published | text | Core name as specified in text in publications. Allows for tracing of coral records through past and future publications.[a] |
| cdata_altCoreName | Core name alternate | text | Alternate core name to cdata_coreName if more than one was used in the publication or where a different name was specified in datasets or secondary sources. |
| cdata_collectTime | Time Collected | text | The year and month (if available) the coral core was collected from living coral. Format is [year CE-month]. Alternatively, if core was from non-living coral, it was described as "Fossil" in this field. |
| cdata_minYear | Min year | numeric | First/oldest year of record. Recorded as integer years CE. Coral age decimal values were rounded down to the calendar year integer. Negative values are years BCE. |



| Field name | Variable | Type | Description |
|---|---|---|---|
| cdata_maxYear | Max year | numeric | Last/youngest year of record. Recorded as integer years CE. Coral age decimal values were rounded down to the calendar year integer. Negative values are years BCE. |
| cdata_archiveSpecies | Coral species | text | Coral genus and species (if known). Records where species name was unknown or not given were written as '[Genus] sp.' [a] Note that species level identification of coral species (particularly *Porites* spp.) is open to disagreement and thus carries a level of uncertainty. |
| cdata_isDatabaseDuplicate | Duplicate coral core name flag | logic | Indicates whether coral core name appears more than once in GBRCD. Flags if coral records were split, and/or where the core name (cdata_coralName & cdata_altCoralName) was identical between publications. |
| cdata_dataCoverageGroup | Record coverage | numeric | The group the record was sorted into based on the length of the record. Groups range from 1 to 3 as described in Table 6. |
| cdata_coralNotes | Coral notes | text | Any notes on coral and additional information that does not fit into other fields e.g. list of core names (if known) for composite records, record observations and considerations. |

a – Descriptions based on Walter et al. (2023) CoralHydro2k database fields.

**Table 6. Coral record coverage group and description relating to the field cdata_dataCoverageGroup.**

| Category | Description |
|---|---|
| 1 | Record length >100 years |
| 2 | Record length 10–100 years |
| 3 | Record length <10 years |

**2.7.2 Geographic Metadata**

Geographic fields (Table 7) provide further core identification information and give the physical location of the coral archive: geographic coordinates, site name and depth of the colony (termed 'elevation' below).

Coral records that did not have listed GPS coordinates for the data were assigned GPS coordinates using "Reefs and shoals – Queensland" dataset (EPSG:4283 - GDA94 coordinate reference system) from State of Queensland (Department of Natural Resources, Mines and Energy) 2021, available under a Creative Commons - Attribution 4.0 International licence. GPS coordinates were approximately matched to the supplied location if a sufficiently detailed map was included in the publication, or the coordinates approximating the reef centre were used. Where coordinates were supplied for the GBRCD, or the

coordinates have been changed or updated from what was published in the original publication, this is noted in the field cdata_coralNotes.

Elevation information is supplied for coral records in the geo_elevation field. This value includes uncertainties inherent in the published data as elevation may describe depth below sea level to top or bottom of the coral and this is not always explicitly described. Likewise, the choice of height datum used to describe sea level is also not always explicitly stated. Fossil coral





records from drill core matrices have the depth down core range (if known), as well as an estimated depth within that range, noted in the geo_notes field. As there is uncertainty around the estimated depth down core, as well as uncertainty due to past changes in GBR sea level compared to present-day sea level (Lewis et al. 2013; Leonard et al. 2020; Hopley et al. 2007) it is left to the user to convert depth down core to an elevation relative to present-day sea level for fossil corals.

**Table 7. Geographic metadata for the CSV version of the GBRCD for record(s) describing the physical location of each coral archive sampling location.**

| Field name | Variable | Type | Description |
| --- | --- | --- | --- |
| geo_latitude | Latitude | numeric | Latitude for the coral. All values are negative due to the GBR's location south of the equator. See geo_notes for considerations. |
| geo_longitude | Longitude | numeric | Longitude for coral. All values are positive due to the GBR's location east of the Prime Meridian. See geo_notes for considerations. |
| geo_siteName | Site | text | Structured location names. Format follows site name: [name] Reef/Island e.g. Abraham Reef or Havannah Island. |
| geo_siteName2 | Site | text | Specific location name (if available) as supplied in publications, e.g. Geoffrey Bay, Heron Island Reef. |
| geo_elevation | Elevation | numeric | Elevation, if known, of corals in meters (m). Values are negative to indicate coral material was collected below present-day sea level or positive to indicate coral material was collected above present-day sea level. Where a range of values is reported the elevation is the average of the range. |
| geo_notes | Elevation notes | text | Any notes on coral geographic information. This includes reported specifics of coral elevation, such as the depth range listed or relationship to water height, information on fossil coral depths from drill core matrices (if available) and considerations. |

### 2.7.3 Dating Metadata

The GBRCD includes 18 BCE records that have chronologies determined in part by U-Th or radiocarbon dating. The dating metadata (Table 8) provides a brief overview of information available in the related record publications and includes either the

corrected U-Th age or uncalibrated $^{14}$C age (± uncertainty) referenced to 1950 (yBP; years before present) and a summary of related information and/or assumptions.

The complete record (or all available) of essential information for reporting U-Th and radiocarbon dating can be found in the original publications. Dating information published for the first time in this study is supplied in Appendix Table D1.





**Table 8. Dating metadata for the CSV version of the GBRCD. Basic information for dated samples.**

| Field name | Variable | Type | Description |
|---|---|---|---|
| dating_UThDate | U-Th Date | numeric | Uranium-thorium (U-Th) age corrected for initial thorium. If more than one age was measured, this field lists the youngest age, or the age used in the original record chronology. Units: yBP. |
| dating_UThDateUncertainty | U-Th Date Uncertainty | numeric | Uranium-thorium (U-Th) age (corrected) uncertainty. If more than one age was measured, this field lists the uncertainty for the youngest age, or the age used in the original record chronology. Units: yBP. |
| dating_14CDate | $^{14}$C Date | numeric | Uncalibrated radiocarbon ($^{14}$C) age. If more than one age was measured, this field lists the youngest age, or the age used in the original record chronology.  Units: yBP. |
| dating_14CDateUncertainty | $^{14}$C Date Uncertainty | numeric | Uncalibrated radiocarbon ($^{14}$C) age uncertainty. If more than one age was measured, this field lists the uncertainty for the youngest age, or the age used in the original record chronology. Units: yBP. |
| dating_notes | Additional dating information | text | Any additional information relating to dating of corals such as the local delta-R reservoir correction used as well as assumptions used for database. |

### 2.7.4 Analysis Metadata

Analysis fields provide information about, or obtained, during measurement of coral variables (Table 9). It includes the list of proxies measured, information about sampling, analytical methods (acronyms given in Table 10), data resolution (as contained in the database record; Table 11) and analytical precision of measurements (annotations given in Table 12). It also includes information such as whether the international coral reference JCp-1 (Okai et al. 2002; Hathorne et al. 2013) was measured and information published about screening for diagenetic alteration of the corals.

Coral data resolution is described as the minimum, maximum, mean and median number of data points per year for each record. The measurements per year were calculated for complete calendar years based on all unique age dates for each record, as records were filtered assuming duplicate dates represent replicate samples. A nominal data resolution label was assigned to each record based on the calculated median resolution described in Table 11 and is a key field for filtering records in the database. The term nominal resolution suffix '_uneven' has been adopted from the CoralHydro2k database (Walter et al. 2023) to describe corals with a variable resolution (i.e. where the record's min, max, mean, and median resolution are not the same). A notes field (meths_methodNotes; Table 9) is also included for information on the measurement method(s) including the sampling method, and any additional methodology notes, as well as an identifier ("REEs measured") for every record where any of the REEs (as listed in Sect. 2.6 above) were included in the record.





**Table 9. Analysis metadata for the CSV version of the GBRCD. Description of fields relating to the measurement of coral data, including variables measured, analysis method and the age model used. Note 'X' is used to represent a list of variables as stated in the description.**

| Field name | Variable | Type | Description |
|---|---|---|---|
| meths_primaryVariablesList | Proxy variables | text | List of proxy variables measured from coral archive. See Table 4 for complete list of variables. |
| meths_additionalVariablesList | Additional variables | text | List of variables that are not standard climate or environment proxy variables or are variables calculated from the measurement of proxy variables. See Table 4 for complete list of variables. |
| meths_XMethod | Method of X measurement | text | Method used for data measurement. (X = TE, Isotope, Lumin) e.g. ICP-MS. Where multiple TE or Isotope methods are used across variables, this field lists the Sr/Ca or $\delta^{18}O$ method and other methods in meths_altMethodInfo. See Table 10 for the complete list. |
| meths_XMethodMachine | Machine for X data measurement | text | Description of machine used for data measurement e.g. Finnigan MAT 251. ([X] = TE, Isotope, Lumin) |
| meths_altMethodInfo | Alternate proxy measurement information | text | List of proxy, method and machine used (if available) for data measurements not included in other method and machine information fields. Items are separated by ";". |
| meths_methodNotes | Sampling notes | text | Any notes on methods including a brief description of sampling method, and other record measurement notes that do not fit in other fields. |
| meths_hasResolutionNominal | Nominal resolution | text | Nominal temporal resolution of the coral record. Nominal choice is based on median frequency of coral data. See Table 11 for term definitions. |
| meths_resolutionMin | Minimum resolution | numeric | Minimum temporal resolution of the record, expressed as data points per year. |
| meths_resolutionMax | Maximum resolution | numeric | Maximum temporal resolution of the record, expressed as data points per year. |
| meths_resolutionMean | Mean resolution | numeric | Mean temporal resolution of the proxy record, expressed as data points per year. |
| meths_resolutionMedian | Median resolution | numeric | Median temporal resolution of the record, expressed as data points per year. |
| meths_isAnomaly | Anomaly data flag | logic | Indicates whether [variable] values are considered anomaly data. Anomaly definition and therefore calculation method may vary by publication. |





| Field name | Variable | Type | Description |
|---|---|---|---|
| meths_chronologyNotes | Notes on age model | text | Brief notes on method used to determine coral data age model. Includes target SST dataset if supplied. SST datasets include AIMS (Australian Institute of Marine Science), Extended Reconstructed SST (ERSST), Comprehensive Ocean-Atmosphere Data Set (COADS), Integrated global ocean station system (IGOSS), Hadley Centre Sea Ice and SST (HadISST), NOAA/Reyn-Smith Optimum Interpolation SST (OISST). Also includes methods for determining the age model for fossil corals where the age model incorporates the absolute age (Table 8). |
| meths_coralExtensionRate | Extension rate | numeric | Average coral extension rate in mm/year. If a range was given in the publication, 'Extension rate' is the average of the range.[a] |
| meths_coralExtensionRateNotes | Extension rate notes | text | Coral extension rate given in the publication or determined from data. This field includes the units, uncertainty, or ranges in values which are not included in the previous field. |
| meths_tissueThickness | Tissue thickness | numeric | Average coral tissue thickness in millimetres (mm). |
| meths_jcpUsed | Flag for measurement of JCp-1 | logic | Indicates whether the JCp-1 standard was measured. |
| meths_jcpSrCaValue | Measured value of JCp-1 | numeric | Sr/Ca value measured for JCp-1 standard. Units: mmol/mol. |
| meths_XAnalyticalPrecision | Measured X analytical precision | numeric | Analytical error for measured proxy values based on a standard (X = SrCa, UCa or d18O). |
| meths_XAnalyticalPrecisionUnits | X analytical precision units | text | Units of analytical error. (X = SrCa, UCa or d18O). Units are relative standard deviation (RSD; %), 2 relative standard deviation (2s %), permille (permil), or 2 standard deviations permille (2s permil). |
| meths_altPrecisionList | Alternate proxies with analytical precision data | text | List of proxies with supplied analytical precision in addition to Sr/Ca, U/Ca or $\delta^{18}$O. See Table 12. Provided to enable comparing precision between studies and assess use of records for research. Items are separated by ";". |





| Field name | Variable | Type | Description |
|---|---|---|---|
| meths_altAnalyticalPrecision | Measured analytical precision | text | List of analytical error value(s) based on a standard for alternate proxies. Values are numerical and are separated by ";" with the order matching meths_altPrecisionList. See Table 12 for the list. |
| meths_altAnalyticalPrecisionUnits | X analytical precision units | text | List of units for alternate proxy analytical error(s). Units are separated by ";" with the order matching meths_altPrecisionList. See Table 12 for the list of units. |
| meths_archiveDiagenesisCheck | Diagenesis screening information | text | Information about diagenesis screening as reported in publications, including the method(s) used to check diagenesis. Where screening information was supplied it is reported as 'Yes; [method(s)]'. Methods used to check include X-Ray Diffraction (XRD), petrographic thin section (TS), scanning electron microscopy (SEM), ultraviolet light assessment (UV). |

a – Descriptions based on Walter et al. (2023) CoralHydro2k database fields.

**Table 10. Description of abbreviations of methods used for the meths_XMethod field given in Table 9.**

| Method Acronym | Method |
|---|---|
| AMS | Accelerator Mass Spectrometry |
| GC | Gas counting |
| ICP-AES | Inductively Coupled Plasma Atomic Emission Spectrometry |
| ICP-OES | Inductively Coupled Plasma Optical Emission Spectrometry |
| ICP-MS | Inductively Coupled Plasma Mass Spectrometry |
| ID-TIMS | Isotope Dilution Thermal Ionisation Mass Spectrometry |
| IRMS | Isotope Ratio Mass Spectrometry |
| LA-ICP-MS | Laser Ablation Inductively Coupled Plasma Mass Spectrometry |
| LSA | Liquid Scintillation Analysis |
| MC-ICP-MS | Multicollector Inductively Coupled Plasma Mass Spectrometry |
| MS | Mass Spectrometry |
| PTIMS | Positive Thermal Ionisation Mass Spectrometry |
| Q-ICP-MS | Quadrupole Inductively Coupled Plasma Mass Spectrometry |
| SSAMS | Single Stage Accelerator Mass Spectrometry |
| TIMS | Thermal Ionisation Mass Spectrometry |



**Table 11. Description of terms for nominal resolution**

| Nominal resolution | Data points per year |
|---|---|
| weekly; weekly_uneven | >= 52 data points per year; "_uneven" was added to records with variable resolutions that typically have over 52 data points per year. |
| fortnightly; fortnightly_uneven | 26 data points per year; "_uneven" was added to records with variable resolutions that typically have 26–51 data points per year. |
| monthly; monthly_uneven | 12 data points per year; "_uneven" was added to records with variable resolutions that typically have 12–25 data points per year. |
| bimonthly; bimonthly_uneven | 6 data points per year; "_uneven" was added to records with variable resolutions that typically have 6–11 data points per year.[a] |
| quarterly; quarterly_uneven | 4 data points per year; "_uneven" was added to records with variable resolutions that typically have 4–5 data points per year.[a] |
| biannual; biannual_uneven | 2 data points per year; "_uneven" was added to records with variable resolutions that typically have 2–3 data points per year.[a] |
| annual; annual_uneven | 1 data point per year; "_uneven" was added to records with variable resolutions that typically have 1 data point per year, and are missing years of data. |
| >annual | Less than 1 data point per year.[a] |

a – Descriptions based on Walter et al. (2023) CoralHydro2k database fields.

**Table 12. List of isotopes, trace elements and trace element ratios with analytical precision values supplied in meths_altPrecisionList. The list of variables is formatted as per meths_altPrecisionList. Units are as provided by original publication(s) or as supplied.**

| Type | List | Units |
|---|---|---|
| Isotope | d11B, d13C, d14C, d15N | permille (permil) or 2 standard deviations permille (2s permil) |
| Trace Element | Ba, Mn, Th, Y | RSD (%) or 2 RSD (2s %) |
| Trace Element Ratio | BaCa, Ba138Ca, BCa, BMg, LiMg, MgCa, MnCa, REEsCa, Sr88Ca[a], YCa | RSD (%), mmol/mol, or 2 RSD (2s %) |

a – Sr88Ca = Sr$^{88}$/Ca as per D'Olivo & McCulloch (2022).

### 2.7.5 Calibration Metadata

Calibration metadata (Table 13) provides information about the creation of proxy-environment equations. Due to the ongoing
focus and broader regional significance of proxy-SST calibrations, detailed information has been provided for proxy-SST
calibrations (intercept, slope, slope uncertainty and r$^2$) for 'raw' (non-centred) data in associated fields for Sr/Ca, U/Ca and
$\delta^{18}$O. Raw data are the data values after measurement processing, while centred (or anomalised) data are the raw data minus
the data mean (which can be the mean of the whole record or part thereof). The calibration information for these SST proxies
was included due to their historical use and the body of published research on their feasibility and/or application for
reconstructing SSTs in the GBR. Where a centred calibration was preferred or the only one published, the calibration
information is separate from the more frequently used non-centred equations. As there were additional proxy-environmental
variable calibrations published for the database records, other calibration equations are provided in calibration notes where
applicable. Availability of additional proxy-environment calibrations that are not provided in the database are also flagged.





**Table 13. Calibration metadata for the CSV version of the GBRCD. Available information on the calibration of coral records to environmental variables, specifically SST. Note 'X' is used to represent a list of variables as stated in the description.**

| Field name | Variable | Type | Description |
| --- | --- | --- | --- |
| calib_isSSTCalibrationCoral | SST calibration coral flag | logic | Indicates whether the coral was used in creating an SST calibration in the database. Includes records that have been used to create a composite record calibration. |
| calib_useSSTCalibration | SST calibration flag | logic | Indicates whether there is SST-Sr/Ca, SST-U/Ca, or SST-$\delta^{18}$O calibration information in the database. Includes records for corals used to create calibration equations and where an SST calibration was applied (e.g. fossil corals). |
| calib_hasAlternateCalibration | Alternate calibration information flag | logic | Indicates whether there are additional or alternate calibration equation(s) available for the record that is not included in the database. Alternate calibration equations can include non-SST calibrations (e.g. luminescence-rainfall calibration equation), and where applicable, may be included in calibration notes. |
| calib_notes | Calibration notes | text | Unstructured description of calibration. Includes information about proxy-SST calibrations and proxy-other calibrations if relevant. |
| calib_isComposite | Composite calibration flag | logic | Indicates whether calibration is a composite from more than one record. |
| calib_compositeList | Composite calibration record list | text | List of coral IDs (if known) used for composite calibration. |
| calib_SSTdata | SST product | text | Description of SST dataset(s) used for proxy-SST calibration, and treatment where applicable. SST datasets include AIMS (Australian Institute of Marine Science), Extended Reconstructed SST (ERSST), Comprehensive Ocean-Atmosphere Data Set (COADS), Integrated global ocean station system (IGOSS), Great Barrier Reef Marine Park (GBRMPA), Hadley Centre Sea Ice and SST (HadISST), NOAA/Reyn-Smith Optimum Interpolation SST (OISST). |
| calib_fitPeriod | Calibration fit period | text | Time period used to fit coral proxy-SST calibration if known. Takes the form of [min year]–[max year]. |
| calib_hasCentredEquation | Centred equation flag | logic | Indicates whether there is an equation based on centred (data minus data mean for whole or specified period of data). |
| calib_centredEquationNotes | Centred equation information | text | Provides information on centred equation including method, equation (as provided in publication) and $r^2$ for relevant proxy-SST calibrations. |



| Field name | Variable | Type | Description |
|---|---|---|---|
| calib_method | Regression method | text | Regression method used for the proxy-SST regression(s) in the publication. If >1 method used, the field lists the methods used for the regression values in the following fields (i.e. calibX_equationIntercept, etc.). Regression methods include Ordinary Least Squares (OLS), Reduced Major Axis (RMA), and Weighted Least Squares (WLS). |
| calibX_equationIntercept | Proxy-SST intercept | numeric | The proxy-SST calibration intercept published for the coral record. Calibration equations take the form proxy = slope*SST + intercept. (Units: [data units]/°C) [a] (X = 1 (Sr/Ca), 2 (U/Ca), 3 ($\delta^{18}$O)) |
| calibX_equationSlope | Proxy-SST slope | numeric | The proxy-SST calibration slope published for the coral record. Calibration equations take the form proxy = slope*SST + intercept. (Units: [data units]/°C) [a] (X = 1 (Sr/Ca), 2 (U/Ca), 3 ($\delta^{18}$O)) |
| calibX_equationSlopeUncertainty | Proxy-SST slope uncertainty | numeric | The proxy-SST calibration slope uncertainty published for the coral record. Calibration equations take the form proxy = slope*SST + intercept. (Units: [data units]/°C)[a] (X = 1 (Sr/Ca), 2 (U/Ca), 3 ($\delta^{18}$O)) |
| calibX_equationR2 | Proxy-SST calibration $r^2$ | numeric | The proxy-SST calibration $r^2$ value published for the coral record.[a] (X = 1 (Sr/Ca), 2 (U/Ca), 3 ($\delta^{18}$O)) |

a – Descriptions based on Walter et al. (2023) CoralHydro2k database fields.

### 2.7.6 Publication Metadata

The GBRCD includes metadata for relevant publications relating to the coral records. The bibliographic information is supplied
for up to three related publications with relevant bibliographic information separated into separate fields for ease of filtering and includes (where applicable) the DOI for the publication and the archived data (Table 14). Another field for an alternate data URL related to the coral record is also provided to enable referencing the original data location when it has been provided in article supplements or similar. If a record (or part thereof) was featured in multiple publications, the order of the bibliographic information was determined by the primary publication source of the data archived, and then the publications
are generally in chronological order (oldest to newest) with preference for the most complete dataset or archive and methods description for the data.





**Table 14. Publication metadata. Publication details for one to three publications linked with the coral record. Note 'X' is used to represent number from the list of variables as stated in description.**

| Field name | Variable | Type | Description |
|---|---|---|---|
| pubX_firstauthor | First author | text | First author of publication (X = 1, 2, 3).[a] |
| pubX_year | Publication year | numeric | Year of publication (X = 1, 2, 3).[a] |
| pubX_doi | DOI | text | DOI of publication (X = 1, 2, 3).[a] |
| pubX_citation | Full citation | text | Complete citation of publication (X = 1, 2, 3).[a] Form is based on the Harvard reference style. |
| pubX_authors | Full author list | text | Full list of authors (in the order published) from each publication (X = 1, 2, 3), including the first listed author. |
| pubX_title | Title | text | Title of publication (X = 1, 2, 3).[a] |
| pubX_journal | Journal | text | Journal of publication (X = 1, 2, 3).[a] |
| pubX_doiData | Data DOI | text | DOI for dataset(s) (X = 1,2,3). |
| pubX_altDataURL | Alternate Data URL | text | Alternate/additional DOI or links to published data related to the coral record. |
| pub_isCoreIDOtherStudy | Core used in other studies | logic | Indicates if coral core ID was known to be used in other studies. Other studies may or may not be included in the database as the study data might not have been available or might be data that does not meet the GBRCD selection criteria (e.g. coral growth studies). If the core ID supplied was ambiguous (i.e. was the site location name), then it was not linked with other studies. Flag is to best of knowledge and may not be definitive. |
| pub_coreIDOtherStudyList | Publications of alternate studies of core | text | List of references where core has been used. Reference is in short form and full list is in the Appendix Table A1. List is to best of knowledge and may not be definitive. |

a – Descriptions based on Walter et al. (2023) CoralHydro2k database fields.

## 3 Database use and citation

The GBRCD is a comprehensive compilation of coral records from the GBR to date that span -5885 to 2017 CE. The database enables the investigation of multiple environmental factors via various coral proxy systems for the GBR, northeastern Australia and potentially the broader Indo-Pacific. Researchers can select from a list of 78 measured or calculated variables to investigate the GBR environment or determine new proxy-environment relationships.

The GBRCD is freely available on the NOAA NCEI World Data Service for Paleoclimatology at https://doi.org/10.25921/hqxk-8h74 (Arzey et al. 2024)

It is up to the user to determine applicable proxies for their investigation and filter records as needed. Best practice (Fig. 2) should begin with filtering by proxy or proxies of interest. Next the CSV dataset(s) should be joined with the metadata fields to enable further filtering of the database. It is then suggested to subsequently filter the data by other metadata of interest such as record coverage, resolution(s), and archive species. Users should read the field cdata_coralNotes for all records of interest as this includes relevant information on caveats and assumptions (if any) that were noted for each record.

Sample code is available for R and Python.



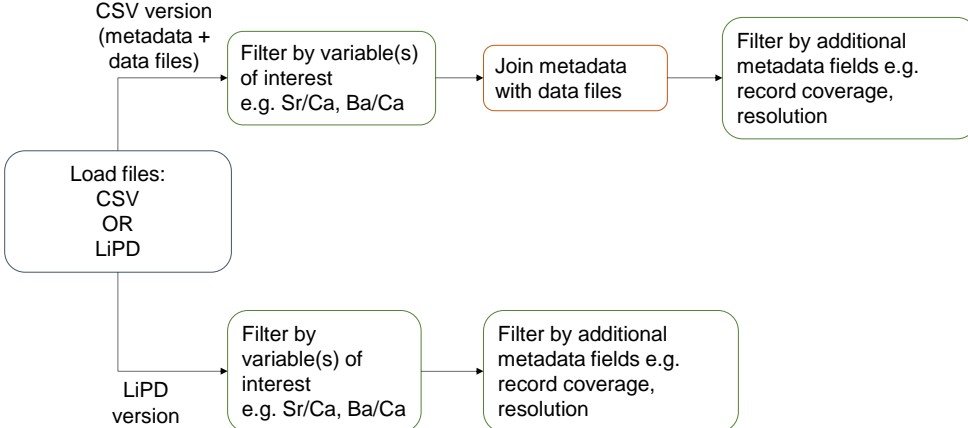

**Figure 2. Flow chart of best practice handling of the two database file versions.**

As previously noted, care is advised for the use of all records in the GBRCD as relevance of the measured variables may vary by location or resolution and so forth. Records have been collated for the GBRCD and basic quality checks have been completed on the data to enable archiving for this study.

If researchers use the GBRCD they should cite this study and the most recent version description of the GBRCD. It is also best practice to cite the original publications of the data (Table A1) alongside the GBRCD database. Citation information and

reference links to the original publications and data archive are included in the metadata (i.e. Table 14).

# 4 Characterising the Database

## 4.1 Record coverage

The GBRCD compiles 208 individual records published since the early 1990s from 58 named reefs and islands (Fig. 1). There are 193 (92.8 %) records from *Porites* spp., eight (3.8 %) from *Isopora* spp., six (2.9 %) from *Acropora* spp., and one (0.5 %)

from a *Cyphastrea* sp. coral. The coral records include 78 variables, of which 72 are measured and 6 are calculated. The majority (116) of records are from the central GBR region (17 to 20°S), 74 from the southern GBR (20 to 24°S), and 18 records from the northern GBR (11 to 17°S).

Most records (58.2 %) have a maximum ('core top') age between 2000 CE and 2017 CE, and 73.1 % between 1990–2017 CE. Peak density of records occurs in the 1980s with ≥105 records in each year between 1980–1984 and nearly as many (≥100) in

the early 1990s (1990–1993) (Fig. 2). Only 59 (28.4 %) of the records' minimum age occurs before 1900 CE, and 18 of those are fossil coral records that range from ~2296 to 5885 years BCE. Of these fossil records, 16 (88.9 %) are from the central GBR and two are from the southern GBR.

Of the 140 records measuring trace elements in the GBRCD, 49.3 % of records are a result of the development of laser ablation methods. There have been 69 LA-ICP-MS trace element records published and 71 trace element records analysed by other

mass spectrometry methods. Many (54) of the non-LA-ICP-MS records are from studies that assess or use SST proxies, likely due to the expectation that trace element records from laser ablation methods have a poorer fit with SST than TE records derived from solution methods. For example, Wu et al. (2021a) LA-ICP-MS study found a notable poor periodicity and Sr/Ca-SST relationship in two out of six records, which led them to suggest the usefulness of LA-ICP-MS for Sr/Ca-SST analysis is potentially limited by the relatively larger LA-ICP-MS analytical error than other mass spectrometry methods.

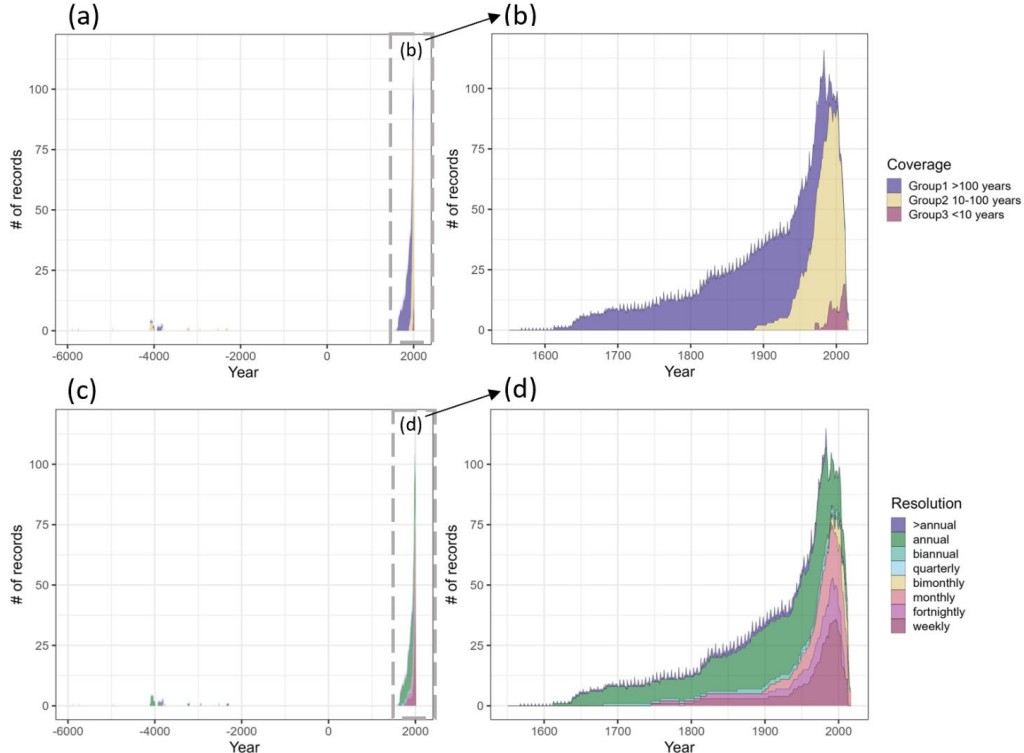

**Figure 3. Temporal coverage of records in the GBRCD colour coded by (a) record coverage Group 1–3 as per Table 6 (1 – records >100 years; 2 – records >10 – <100 years; 3 – records <10 years), B) subset of (a) showing only records since 1550 CE, (c) nominal resolution as per Table 11 (see figure legend) (records with "_uneven" nominal resolution were combined with even resolution records), and (d) subset of (c) showing only records since 1550 CE. The 'spikiness' visible in (b) Group 1 & (d) >annual resolution records is due to the combination of biennial and quinquennial resolution records.**

The majority of records in the database include Sr/Ca measurements (118 of 208; Fig. 4), and 63 of these records are used in SST calibrations. The other variables measured most frequently include B/Ca (59 records), Ba/Ca (78 records. 4), Mg/Ca (73 records; Fig. 4), U/Ca (77 records; Fig. 4), and luminescence (57 records; Fig. 5). There are 51 isotope records, of which 27 include $\delta^{18}O$ records (Fig. 5), 17 $\delta^{13}C$, 15 $\delta^{11}B$, 6 $\delta^{14}C$, 5 $\delta^{15}N$ (Fig. 5), 1 $\delta^{66}Zn$ and 1 $\delta^{98}Mo$. Additionally, there are 18 records measuring REEs (Fig. 5), noting this number excludes records that only measure yttrium and no REEs.

The number and spatial distribution of Sr/Ca records (Fig. 4C) reflects the dual utility of the ratio as an SST proxy and to determine the coral age model. Similarly, U/Ca (Fig. 4D) is also used as an SST proxy and to determine coral age models. When combined with other proxies (e.g. Sr/Ca), U/Ca may improve SST predictions (Wu et al. 2021a; DeCarlo et al. 2016). The B/Ca proxy is commonly measured in addition to target proxies rather than being the focus of the study. However, B/Ca has more recently been paired with $\delta^{11}B$ in studies examining pH, to determine the dissolved inorganic carbon in the calcifying fluid. Mg/Ca has been measured relatively frequently (Fig. 4B), and while known as an SST proxy in other genera, the Mg/Ca-SST relationship in *Porites* spp. is weak or results are highly variable (e.g. Sadler et al. 2014; Wu et al. 2021a), possibly due to the coral actively discriminating against $Mg^{2+}$ during biomineralisation (Marchitto et al. 2018). Ba/Ca (Fig. 4A) is frequently measured in the GBR, primarily to assess terrigenous inputs and river flow, and is mainly measured by LA-ICP-MS (58 of 78 Ba/Ca records).

Despite the widespread analysis of coral $\delta^{18}O$ across the tropical Pacific (~100 $\delta^{18}O$-only and paired Sr/Ca- $\delta^{18}O$ records (Walter et al. 2023)), there are few (only 27) $\delta^{18}O$ records available for the GBR (Fig. 5A), especially considering the length of time $\delta^{18}O$ has been used as an SST or hydrological proxy. This is potentially due to the paired forcing (SST and hydrology)





that determines the coral δ¹⁸O for nearshore corals, as well as the development and use of coral luminescence for assessing

freshwater inputs into the GBR. Records including luminescence measurements make up 27.4 % of the GBRCD (Fig. 5B).

**Figure 4. Location map of GBRCD records for common major trace element ratios; (a) Ba/Ca, (b), Mg/Ca, (c) Sr/Ca and (d) U/Ca. The size of the point indicates the number of records, and is specific to each map.**



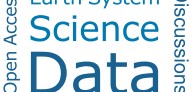

**Figure 5. Location map of GBRCD records for (a) $\delta^{18}$O, (b) luminescence (Lumin), (c) $\delta^{15}$N and (d) Rare Earth Elements (REEs). The size of the point indicates the number of records, and is specific to each map. Note that Lumin\* includes all the methods included in the GBRCD for measuring luminescence.**



## 5 Discussion

There has been a push for coral records to include a range of details to meet minimum reporting standards for palaeo-archive
records (such as MARPA (Dassié et al. 2017) and PaCTS (Khider et al. 2019)). These community-led publications report the
ideal metadata that should be supplied with the publication of each record and provide a useful guide for best practice. The
call for minimum reporting standards for data and metadata is echoed here. A particular challenge in producing the GBRCD
was that metadata was often spread across multiple publications. To improve traceability between publications, relevant
metadata could be included in the supplement and in a publicly available data repository.

Another area for improvement is the reporting of screening corals for diagenesis. Diagenesis is a known source of error in
geochemical analysis as it can remove the primary coral proxy signal (Hendy et al. 2007; Sayani et al. 2011; McGregor &
Gagan 2003; Nothdurft et al. 2007). Few publications (37 of 208 records in the GBRCD) report on diagenetic alteration and
fail to mention the screening method(s) or outcomes (see meths_archiveDiagenesisCheck). There are no established best
practice guidelines for screening of coral for diagenesis at present, though it is suggested that it is ideal to use petrographic
analysis with a combination of methods such as XRD, and densitometry (McGregor & Abram 2008). However, due to the
strengths and weaknesses of the various methods as noted by Nothdurft & Webb (2009), examination by petrographic analysis
(preferably using scanning electron microscopy) is required to confirm the type and level of diagenetic alteration (if any)
present in a sample.

X-radiographs showing sampling paths should also be routinely provided. Pioneering work on corals from the GBR showed
that it is essential for corals to be sampled along the maximum growth axis. Providing this information was previously common
(Alibert & McCulloch 1997; Calvo et al. 2007; Pelejero et al. 2005; Chakraborty et al. 2000; DeLong et al. 2007), but x-
radiographs have not been published consistently in recent years. Computed tomography (CT) scans of coral cores are
sometimes used instead of, or in addition to, X-radiographs. Unlike X-radiographs, CT scans can be run on uncut coral cores
and drill core matrices and similar to X-radiographs, can provide information about extension rates, density and calcification
(Bosscher 1993; Mollica et al. 2018; Crook et al. 2013), as well as growth morphologies and bioerosion (Prouty et al. 2017;
DeCarlo et al. 2015; Li et al. 2021). As with X-radiographs, the CT scan images providing relevant information should be
included with the paper.

The GBRCD lends itself to meeting some of the suggestions put forward by Lough (2004) to improve the use of corals for
palaeoclimatology. While not absolute replication, the GBRCD facilitates using multiple records to determine common
environmental signals and identify non-climatic artifacts in those records. For example, the database enables combining
multiple records to assess trends, similar to methods used by Hendy et al. (2002).

The long and comprehensive history of coral research on the GBR can make it difficult to find relevant research and records
to understand the current state of coral palaeoclimate and palaeoenvironment studies on the GBR and thus identify gaps in
existing research. The GBRCD can be used to discern where future coral proxy records are needed. Additionally, the database
includes 67 duplicate core names, which generally indicates different types of analyses on the same coral head and may provide
an opportunity for comparisons between methods.

The GBRCD can be used for analysis and examination of trends at local to regional scales. Records from the GBR database
can be selectively used alongside other datasets to assess the past environment of Australia and/or the Pacific and identify
trends and periods of note. It enables comparisons of different periods, such as the modern corals and older Holocene coral
records, making it possible to identify notable changes between the time periods. Furthermore, records from the GBRCD can
be combined with new and existing models of the palaeoenvironment.



### 5.1 Database Availability and its Future

This is stage 1 of the GBRCD.

The database is archived on the NOAA NCEI World Data Service for Paleoclimatology. To access the database, visit
https://doi.org/10.25921/hqxk-8h74 (Arzey et al. 2024) Example scripts to help users access the database, in either format, are available on the NOAA study page. The example scripts and database information are also available on the GBRCD GitHub (https://github.com/arzeyak/GBR-Coral-Skeletal-Proxy-Records-Database; GBRCD, 2024).

Future updates to the database are expected to occur on an annual basis. We welcome submission of historic GBR coral record data for inclusion in the GBRCD, including existing unpublished datasets, supporting FAIR data principles (Wilkinson et al.
2016). Submitted records should meet general criteria outlined above in 'Selection criteria for inclusion in GBRCD'. For researchers that want to submit records (published or unpublished) to the GBRCD, please check the GBRCD GitHub for details on how to submit your records for the database.

## 6 Conclusions

The GBRCD is a comprehensive database of coral records from the GBR produced and published since the early 1990s, as
well as seven new records that are first published here. The database archives 208 coral records that are publicly available at NOAA WDS.

Records have been standardised and converted to a machine-readable format that is easy to use to investigate the database, and to select records of interest for analysis. With detailed metadata in the GBRCD there are many avenues for investigating the past and present GBR environment.

There is scope in the future for improving the archiving of research datasets by building on the GBRCD and including the minimum recommended information for traceability and use.



## Appendix A – GBRCD Records

Table A1. Reference data for publications cited in the GBRCD. Primary and secondary variables are the variable name as per Table
4. Citations in the Record Publication(s) columns are listed in the order presented in the database (pub1; pub2; pub3). Citations in
the Additional Core Publication(s) column are listed in alphanumerical order.

| Dataset ID | Core Name | Latitude | Longitude | Location | Primary Variables (Secondary Variables) | Record Publication(s) | Additional Core Publication(s) |
|---|---|---|---|---|---|---|---|
| AL03DAV01_1 | Davies 2 (side) | -18.8 | 147.7 | Davies Reef | Sr/Ca (Distance) | Alibert et al. (2003); Alibert & McCulloch (1997) | Fallon (2000); Fallon et al. (2003) |
| AL03DAV01_2 | Davies 2 (side) | -18.8 | 147.7 | Davies Reef | Ba; Ba/Ca; Cd; Mn; Mn/Ca; P; Y (Distance) | Alibert et al. (2003) | Alibert & McCulloch (1997); Fallon (2000); Fallon et al. (2003) |
| AL03PAN01_1 | PAN 98-2 (B3) | -18.8 | 146.4 | Pandora Reef | Sr/Ca (Distance) | Alibert et al. (2003) | |
| AL03PAN01_2 | PAN 98-2 (B3) | -18.8 | 146.4 | Pandora Reef | Ba/Ca; Mn; P (Distance) | Alibert et al. (2003) | |
| AL03PAN01_3 | PAN 98-2 (B1) | -18.8 | 146.4 | Pandora Reef | Ba/Ca; Cd; Mn; P; Y (Distance) | Alibert et al. (2003) | |
| AL03PAN01_4 | PAN 98-2 (B1) | -18.8 | 146.4 | Pandora Reef | Ba/Ca; Cd; Mn; Mn/Ca; P; Y (Distance) | Alibert et al. (2003) | |
| AR24DIP01 | Dip-04 | -18.414 | 147.448 | Dip Reef | Ba/Ca; Mg/Ca; Sr/Ca; U/Ca; Y/Ca (Distance) | This study | |
| AR24OTI01 | GUT12 | -23.5053 | 152.0923 | One Tree Island | Ba/Ca; Sr/Ca (Distance) | This study | |
| AR24OTI02 | OTI 1-3 | -23.499 | 152.072 | One Tree Island | Ba/Ca; B/Ca; Ca/Ca; Mg/Ca; Sr/Ca; U/Ca (Distance) | This study | |
| AR24OTI03 | OTI 3-1 | -23.499 | 152.072 | One Tree Island | Ba/Ca; B/Ca; Ca/Ca; Mg/Ca; Sr/Ca; U/Ca (Distance) | This study | |
| AR24OTI04 | OTI4_6C (OTI-4-19-6C) | -23.499 | 152.072 | One Tree Island | Ba/Ca; B/Ca; Ca/Ca; Mg/Ca; Sr/Ca; U/Ca (Distance) | This study | |



| Dataset ID | Core Name | Latitude | Longitude | Location | Primary Variables (Secondary Variables) | Record Publication(s) | Additional Core Publication(s) |
|---|---|---|---|---|---|---|---|
| AR24OTI05 | OTI4_6D (OTI-4-22-6D) | -23.499 | 152.072 | One Tree Island | Ba/Ca; B/Ca; Ca/Ca; Mg/Ca; Sr/Ca; U/Ca (Distance) | This study | |
| AR24SLY01 | SR | -19.25 | 149.1167 | Stanley Reef | $\delta^{13}C$; $\Delta^{14}C$; $\delta^{18}O$ | This study | Chakraborty (1993); Chakraborty et al. (2000) |
| BR17HER01 | GBR-637 | -23.442 | 151.914 | Heron Island | $\delta^{18}O$; Sr/Ca | Brenner et al. (2017) | |
| BR17HER02 | GBR-641A | -23.442 | 151.914 | Heron Island | $\delta^{18}O$; Sr/Ca | Brenner et al. (2017) | |
| BR17HER03 | GBR-538 | -23.442 | 151.914 | Heron Island | $\delta^{18}O$; Sr/Ca | Brenner et al. (2017) | |
| BR17HER04 | HER-1F (HER-OR) | -23.442 | 151.914 | Heron Island | $\delta^{18}O$; Sr/Ca | Brenner et al. (2017) | |
| BR17HER05 | HER-13 (HER12-13) | -23.442 | 151.914 | Heron Island | $\delta^{18}O$; Sr/Ca | Brenner et al. (2017) | Linsley et al. (2019) |
| CA07FLI01 | Flinders (FLI02A) | -17.73 | 148.43 | Flinders Reef | $\delta^{18}O$; Sr/Ca | Calvo et al. (2007) | Pelejero et al. (2005) |
| CH93SLY01 | SR | -19.25 | 148.1167 | Stanley Reef | $\delta^{13}C$; $\delta^{18}O$ (Density; Distance) | Chakraborty (1993); Chakraborty et al. (2000) | |
| CH93SLY02 | SR | -19.25 | 148.1167 | Stanley Reef | $\delta^{13}C$; $\delta^{18}O$ (Density; Distance) | Chakraborty (1993); Chakraborty et al. (2000) | |
| DE14ARL01 | 10AR1 | -16.6381 | 146.1036 | Arlington Reef | $\delta^{13}C$; $\delta^{18}O$; Mg/Ca; Sr/Ca ($\delta^{18}O_{seawater}$; Extension) | Deng et al. (2014) | |
| DE14ARL02 | 10AR2 | -16.6381 | 146.1036 | Arlington Reef | $\delta^{13}C$; $\delta^{18}O$; Mg/Ca; Sr/Ca ($\delta^{18}O_{seawater}$; Extension) | Deng et al. (2014); Wang et al. (2019) | Chen et al. (2021); Xiao et al. (2020) |
| DO15GBR01 | 1709_6 | -17.876 | 146.729 | Reef 17-065 | $\delta^{11}B$ | D'Olivo et al. (2015) | D'Olivo et al. (2018) |





| Dataset ID | Core Name | Latitude | Longitude | Location | Primary Variables (Secondary Variables) | Record Publication(s) | Additional Core Publication(s) |
|---|---|---|---|---|---|---|---|
| DO15HAV01 | HAV09_3 | -18.836 | 146.5259 | Havannah Island | $\delta^{11}B$ | D'Olivo et al. (2015) | D'Olivo & McCulloch (2017); D'Olivo et al. (2013); D'Olivo et al. (2018) |
| DO15HAV02 | HAV09_3 | -18.836 | 146.5259 | Havannah Island | $\delta^{11}B$ | D'Olivo et al. (2015); Chen et al. (2021) | D'Olivo & McCulloch (2017); D'Olivo et al. (2013); D'Olivo et al. (2018) |
| DO15HAV03 | HAV06A | -18.84 | 146.54 | Havannah Island | $\delta^{11}B$ | D'Olivo et al. (2015); Chen et al. (2021) | D'Olivo et al. (2013) |
| DO15PAN01 | PAN02 (PAN02A) | -18.81 | 146.43 | Pandora Reef | $\delta^{11}B$ | D'Olivo et al. (2015); Chen et al. (2021) | D'Olivo et al. (2013) |
| DO15RIB09 | RIB09_3 | -18.48 | 146.87 | Rib Reef | $\delta^{11}B$ | D'Olivo et al. (2015) | D'Olivo et al. (2013); D'Olivo et al. (2018) |
| DO17HAV01 | HAV09_3 (Path A) | -18.836 | 146.5259 | Havannah Island | $\delta^{11}B$ (Extension) | D'Olivo & McCulloch (2017) | D'Olivo et al. (2015); D'Olivo et al. (2018) |
| DO17HAV02 | HAV09_3 (Path B) | -18.836 | 146.5259 | Havannah Island | B/Ca; $\delta^{11}B$; Li/Ca; Li/Mg; Mg/Ca; Sr/Ca; U/Ca (Extension) | D'Olivo & McCulloch (2017) | D'Olivo et al. (2015); D'Olivo et al. (2018) |
| DO17HAV03 | HAV09_3 (Path C) | -18.836 | 146.5259 | Havannah Island | B/Ca; $\delta^{11}B$; Li/Ca; Li/Mg; Mg/Ca; Sr/Ca; U/Ca (Extension) | D'Olivo & McCulloch (2017) | D'Olivo et al. (2015); D'Olivo et al. (2018) |
| DO17HAV04 | HAV09_3 (Path D) | -18.836 | 146.5259 | Havannah Island | B/Ca; $\delta^{11}B$; Li/Ca; Li/Mg; Mg/Ca; Sr/Ca; U/Ca (Extension) | D'Olivo & McCulloch (2017) | D'Olivo et al. (2015); D'Olivo et al. (2018) |
| DO18DAV01 | DAV-1 (DAV09_2) | -18.83 | 147.63 | Davies Reef | B/Ca; Li/Ca; Li/Mg; Mg/Ca; Sr/Ca; U/Ca | D'Olivo et al. (2018) | |





| Dataset ID | Core Name | Latitude | Longitude | Location | Primary Variables (Secondary Variables) | Record Publication(s) | Additional Core Publication(s) |
|---|---|---|---|---|---|---|---|
| DO18DAV02 | DAV-2 (DAV13_2) | -18.8 | 147.63 | Davies Reef | B/Ca; Li/Ca; Li/Mg; Mg/Ca; Sr/Ca; U/Ca | D'Olivo et al. (2018) | McCulloch et al. (2017); Thompson et al. (2022) |
| DO18DAV03 | DAV-3 (DAV13_3) | -18.8 | 147.63 | Davies Reef | B/Ca; Li/Ca; Li/Mg; Mg/Ca; Sr/Ca; U/Ca | D'Olivo et al. (2018) | McCulloch et al. (2017); Thompson et al. (2022) |
| DO18FIT01 | FITZ (FZ04B) | -16.923 | 145.993 | Fitzroy Island | B/Ca; Li/Ca; Li/Mg; Mg/Ca; Sr/Ca; U/Ca | D'Olivo et al. (2018) | D'Olivo & McCulloch (2022) |
| DO18GBR01 | CORE-17 (1709_6) | -17.876 | 146.729 | Reef 17-065 | B/Ca; Li/Ca; Li/Mg; Mg/Ca; Sr/Ca; U/Ca | D'Olivo et al. (2018) | |
| DO18HAV01 | HAV (HAV09_3) | -18.836 | 146.5259 | Havannah Island | B/Ca; Li/Ca; Li/Mg; Mg/Ca; Sr/Ca; U/Ca | D'Olivo et al. (2018) | D'Olivo & McCulloch (2017) |
| DO18LIZ01 | LIZ (LIZ13_1) | -14.69 | 145.44 | Lizard Island | B/Ca; Li/Ca; Li/Mg; Mg/Ca; Sr/Ca; U/Ca | D'Olivo et al. (2018) | D'Olivo et al. (2015) |
| DO18MYR01 | MYR-1 (MYR5_271) | -18.27 | 147.38 | Myrmidon Reef | B/Ca; Li/Ca; Li/Mg; Mg/Ca; Sr/Ca; U/Ca | D'Olivo et al. (2018) | |
| DO18MYR02 | MYR-2 (MYR13_2) | -18.27 | 147.38 | Myrmidon Reef | B/Ca; Li/Ca; Li/Mg; Mg/Ca; Sr/Ca; U/Ca | D'Olivo et al. (2018) | |
| DO18MYR03 | MYR-3 (MYR13_5) | -18.27 | 147.38 | Myrmidon Reef | B/Ca; Li/Ca; Li/Mg; Mg/Ca; Sr/Ca; U/Ca | D'Olivo et al. (2018) | |
| DO18MYR04 | MYR-4 (MYR13_7) | -18.27 | 147.38 | Myrmidon Reef | B/Ca; Li/Ca; Li/Mg; Mg/Ca; Sr/Ca; U/Ca | D'Olivo et al. (2018) | |
| DO18RIB01 | RIB (RIB09_3) | -18.48 | 146.87 | Rib Reef | Ba/Ca; B/Ca; B/Mg; Li/Ca; Li/Mg; Mg/Ca; Sr/Ca; U/Ca | D'Olivo et al. (2018) | D'Olivo et al. (2015) |
| DO22ARL01 | AER04-1 | -16.68 | 146.03 | Arlington Reef | Ba$^{138}$/Ca; Ba/Ca; Sr/Ca; U/Ca | D'Olivo & McCulloch (2022) | |
| DO22BRT01 | BRT01 | -18.14 | 146.73 | Britomart Reef | Ba/Ca; Sr/Ca; U/Ca | D'Olivo & McCulloch (2022) | |





| Dataset ID | Core Name | Latitude | Longitude | Location | Primary Variables (Secondary Variables) | Record Publication(s) | Additional Core Publication(s) |
|---|---|---|---|---|---|---|---|
| DO22FIT01 | FZ04B-1 | -16.93 | 145.99 | Fitzroy Island | $Ba^{138}/Ca$; Ba/Ca; Sr/Ca; U/Ca | D'Olivo & McCulloch (2022) | D'Olivo et al. (2018) |
| DO22FIT02 | FZ04-2 | -16.93 | 145.99 | Fitzroy Island | $Ba^{138}/Ca$; Ba/Ca; Sr/Ca; U/Ca | D'Olivo & McCulloch (2022) | |
| DO22HIG01 | HI03-1 | -17.17 | 146 | High Island | $Ba^{138}/Ca$; Ba/Ca; Sr/Ca; U/Ca | D'Olivo & McCulloch (2022) | |
| DO22HMP01 | HMP01B | -23.2 | 151 | Humpy Island | Ba/Ca; Sr/Ca; U/Ca | D'Olivo & McCulloch (2022) | Lough (2007); Lough (2011a); Lough (2011b); |
| DO22MAG01 | MAG01D | -19.15 | 146.87 | Geoffrey Bay, Magnetic Island | $Ba^{138}/Ca$; Ba/Ca; Sr/Ca; U/Ca | D'Olivo & McCulloch (2022) | Barnes & Lough (1992); De'ath et al. (2009); Erler et al. (2016); Lewis (2005); Lewis et al. (2007); Lough (2007); Lough (2011a); Lough (2011b); Lough & Barnes (1997) |
| DO22OYS01 | OY04-3 | -16.62 | 145.91 | Oyster Reef | $Ba^{138}/Ca$; Ba/Ca; Sr/Ca; U/Ca | D'Olivo & McCulloch (2022) | |
| DO22PAN01 | PAN08-1 | -18.82 | 146.43 | Pandora Reef | $Ba^{138}/Ca$; Ba/Ca; Sr/Ca; U/Ca | D'Olivo & McCulloch (2022) | D'Olivo et al. (2013) |
| DO22PAN02 | PAN08-2 | -18.82 | 146.43 | Pandora Reef | $Ba^{138}/Ca$; Ba/Ca; Sr/Ca; U/Ca | D'Olivo & McCulloch (2022) | D'Olivo et al. (2013) |
| DR95LMI01 | Lady Musgrave | -23.9 | 152.38 | Lady Musgrave Island | $\Delta14C$ | Druffel & Griffin (1995) | |
| DR99ABR01_1 | Abraham-1 (99aust02a) | -22.1 | 152.48 | Abraham Reef | $\delta^{13}C$; $\delta^{18}O$ | Druffel & Griffin (1999); Druffel & Griffin (1995); Druffel & Griffin (1993) | Linsley et al. (2019) |

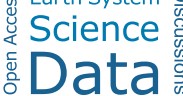



| Dataset ID | Core Name | Latitude | Longitude | Location | Primary Variables (Secondary Variables) | Record Publication(s) | Additional Core Publication(s) |
|---|---|---|---|---|---|---|---|
| DR99ABR01_2 | Abraham-1 (99aust02a) | -22.1 | 152.48 | Abraham Reef | $\Delta14C$ | Druffel & Griffin (1999); Druffel & Griffin (1995); Druffel & Griffin (1993) | Linsley et al. (2019) |
| DR99ABR02 | Abraham-2 (99aust02b) | -22.1 | 152.48 | Abraham Reef | $\Delta14C$ | Druffel & Griffin (1999); Druffel & Griffin (1995); Druffel & Griffin (1993) | |
| DR99HER01 | Heron (99aust03a) | -23.44 | 151.98 | Heron Island | $\Delta14C$ | Druffel & Griffin (1999); Druffel & Griffin (1995) | |
| EL19OTI01 | PET12 | -23.4873 | 152.0814 | Pete's Bay, One Tree Island | $Ba^{138}/Ca$; Ba/Ca; B/Ca; Cd/Ca; Fe/Ca; Li/Ca; Mg/Ca; Mn/Ca; Pb/Ca; P/Ca; Sr/Ca; U/Ca; Y/Ca | Ellis et al. (2019) | |
| ER16MAG01 | MAG01D | -19.15 | 146.87 | Geoffrey Bay, Magnetic Island | $\delta^{15}N$; $\delta^{18}O$ (Calcification; Density; Extension) | Erler et al. (2016); Erler et al. (2020) | Barnes & Lough (1992); D'Olivo & McCulloch (2022); De'ath et al. (2009); Lewis (2005); Lewis et al. (2007); Lough (2011a); Lough (2011b); Lough & Barnes (1997) |




| Dataset ID | Core Name | Latitude | Longitude | Location | Primary Variables (Secondary Variables) | Record Publication(s) | Additional Core Publication(s) |
|---|---|---|---|---|---|---|---|
| ER20HAV01 | HAV01A | -18.837 | 146.548 | Havannah Island | $\delta^{15}N$ | Erler et al. (2020) | D'Olivo & McCulloch (2022); De'ath et al. (2009); Hendy et al. (2002); Hendy et al. (2003a); Hendy et al. (2003b); Hendy et al. (2007); Hendy et al. (2012); Isdale et al. (1998); Lough (2007); Lough (2011a); Lough (2011b); Lough & Barnes (1997); Lough et al. (2015); McCulloch et al. (2003); Palmer et al. (2015) |
| ER20HAV02 | Hav33a (Hav33b) | -18.837 | 146.548 | Havannah Island | $\delta^{15}N$ | Erler et al. (2020) | |
| ER20PAN01 | Pan04b | -18.815 | 146.436 | Pandora Reef | $\delta^{15}N$ | Erler et al. (2020) | De'ath et al. (2009); Lough (2007); Lough & Barnes (1997) |
| ER20PAN02 | Pan22b | -18.815 | 146.436 | Pandora Reef | $\delta^{15}N$ | Erler et al. (2020) | Cantin & Lough (2014); De'ath et al. (2009) |
| FA00PAN01 | Pandora 1-98 (b) | -18.8 | 146.4 | Pandora Reef | Ba/Ca; B/Ca; Mg/Ca; Sr/Ca; U/Ca (Distance) | Fallon (2000) | |
| FA03DAV01 | Davies 2 | -18.8 | 147.7 | Davies Reef | Ba/Ca; B/Ca; Mg/Ca; Mn; Sr/Ca; U/Ca (Distance) | Fallon et al. (2003); Fallon (2000) | Alibert & McCulloch (1997); Alibert et al. (2003) |
| FA03DAV02 | Davies 8 | -18.8 | 147.7 | Davies Reef | Ba/Ca; B/Ca; Mg/Ca; Mn; Sr/Ca; U/Ca (Distance) | Fallon et al. (2003); Fallon (2000) | Alibert & McCulloch (1997) |



| Dataset ID | Core Name | Latitude | Longitude | Location | Primary Variables (Secondary Variables) | Record Publication(s) | Additional Core Publication(s) |
|---|---|---|---|---|---|---|---|
| FA03HAV01 | Havannah Island | -18.843 | 146.537 | Havannah Island | Ba/Ca; B/Ca; Mg/Ca; Mn; Sr/Ca; U/Ca (Distance) | Fallon et al. (2003); Fallon (2000) | |
| FA03MYR01 | Myrmidon 2 (Myr-2) | -18.266 | 147.383 | Myrmidon Reef | Ba/Ca; B/Ca; Mg/Ca; Mn; Sr/Ca; U/Ca (Distance) | Fallon et al. (2003); Fallon (2000) | Marshall & McCulloch (2002) |
| FA03ORP01 | Orpheus Island | -18.634 | 146.495 | Orpheus Island | Ba/Ca; B/Ca; Mg/Ca; Mn; Sr/Ca; U/Ca (Distance) | Fallon et al. (2003); Fallon (2000) | |
| FA03PAN01 | Pandora 1-98 (a) | -18.8 | 146.4 | Pandora Reef | Ba/Ca; B/Ca; Mg/Ca; Mn; Sr/Ca; U/Ca (Distance) | Fallon et al. (2003); Fallon (2000) | |
| FA03WHE01 | Wheeler Reef | -18.799 | 147.528 | Wheeler Reef | Ba/Ca; B/Ca; Mg/Ca; Sr/Ca; U/Ca (Distance) | Fallon et al. (2003); Fallon (2000) | Alibert & McCulloch (1997) |
| GA98ORP01 | OR-1 | -18.567 | 146.483 | Iris Point, Orpheus Island | $\delta^{18}O$; Sr/Ca ($\delta^{18}O_{seawater}$) | Gagan et al. (1998); Gagan et al. (2002); Gagan et al. (1994) | Gagan et al. (2012) |
| HE02GBR01 | NA | -17.78 | 146.13 | Central GBR | $\delta^{18}O$; Sr/Ca; U/Ca ($\delta^{18}O_{seawater}$) | Hendy et al. (2002); Hendy et al. (2007) | |
| HE03GBR01 | NA | -17.78 | 146.13 | Central GBR | Luminescence visual indices | Hendy et al. (2003a) | |
| HE03GPI01 | GPI-02A-01 (GPI02A) | -18.733 | 146.567 | Great Palm Island | Luminescence visual indices | Hendy et al. (2003a); Hendy et al. (2007); Leonard et al. (2016) | De'ath et al. (2009); Hendy et al. (2002); Hendy et al. (2003b); Leonard et al. (2016); Lough (2007); Lough (2011a); Lough (2011b); Lough & Barnes (1997) |



| Dataset ID | Core Name | Latitude | Longitude | Location | Primary Variables (Secondary Variables) | Record Publication(s) | Additional Core Publication(s) |
|---|---|---|---|---|---|---|---|
| IS98HAV02 | Havannah (HAV01A) | -18.85 | 146.55 | Havannah Island | Luminescence Sum | Isdale et al. (1998) | D'Olivo & McCulloch (2022); De'ath et al. (2009); Erler et al. (2020); Hendy et al. (2002); Hendy et al. (2003a); Hendy et al. (2003b); Hendy et al. (2007); Hendy et al. (2012); Lough (2007); Lough (2011a); Lough (2011b); Lough & Barnes (1997); Lough et al. (2015); McCulloch et al. (2003); Palmer et al. (2015) |
| IS98PAN01 | Pandora (PAN08B) | -18.82 | 146.43 | Pandora Reef | Luminescence Sum | Isdale et al. (1998) | Hendy et al. (2002); Hendy et al. (2003a); Hendy et al. (2003b); Hendy et al. (2007); Hendy et al. (2012) |
| JU08KES01a | KIB | -20.9177 | 149.4178 | Keswick Island | Luminescence | Jupiter et al. (2008) | |
| JU08KES01b | KIA | -20.9177 | 149.4186 | Keswick Island | Ce; Dy; Er; Eu; Gd; Ho; La; Lu; Nd; Pr; Sm; Tb; Tm; Y; Yb; Y/Ho | Jupiter (2008); Jupiter (2006) | |
| JU08KES02 | KIC | -20.9177 | 149.4178 | Keswick Island | Luminescence | Jupiter et al. (2008) | Marion et al. (2021) |
| JU08KES03 | KIE | -20.9319 | 149.4267 | Keswick Island | Ba/Ca; Luminescence | Jupiter et al. (2008) | |
| JU08RTI01a | RTC | -21.1755 | 149.2639 | Round Top Island | Luminescence | Jupiter et al. (2008) | Marion et al. (2021) |




| Dataset ID | Core Name | Latitude | Longitude | Location | Primary Variables (Secondary Variables) | Record Publication(s) | Additional Core Publication(s) |
|---|---|---|---|---|---|---|---|
| JU08RTI01b | RTF | -21.1708 | 149.2644 | Round Top Island | Ce; Dy; Er; Eu; Gd; Ho; La; Lu; Nd; Pr; Sm; Tb; Tm; Y; Yb; Y/Ho | Jupiter (2008); Jupiter (2006) | Marion et al. (2021) |
| JU08RTI02 | RTH | -21.172 | 149.2639 | Round Top Island | Ba/Ca; Luminescence | Jupiter et al. (2008) | Marion et al. (2021) |
| JU08RTI03 | RTI | -21.1717 | 149.2634 | Round Top Island | Luminescence | Jupiter et al. (2008) | |
| JU08SCA01 | SCA | -20.8522 | 149.6021 | Scawfell Island | Luminescence | Jupiter et al. (2008) | |
| JU08SCA02 | SCB | -20.8522 | 149.6021 | Scawfell Island | Luminescence | Jupiter et al. (2008) | |
| JU08SCA03 | SCC | -20.8522 | 149.6021 | Scawfell Island | Ba/Ca; Luminescence | Jupiter et al. (2008) | Marion et al. (2021) |
| LE05MAG01 | MAG01D | -19.15 | 146.858 | Geoffrey Bay, Magnetic Island | $\delta^{13}C$; $\delta^{18}O$ ($\delta^{18}O_{seawater}$) | Lewis (2005) | Barnes & Lough (1992); D'Olivo & McCulloch (2022); De'ath et al. (2009); Erler et al. (2016); Lewis et al. (2007); Lough (2007); Lough (2011a); Lough (2011b); Lough & Barnes (1997) |
| LE05NEL01 | NEL01D | -19 | 147 | Nelly Bay, Magnetic Island | $\delta^{13}C$; $\delta^{18}O$ ($\delta^{18}O_{seawater}$) | Lewis (2005) | Lewis et al. (2007); Lough et al. (2014) |
| LE05NEL02 | NEL03D | -19 | 147 | Nelly Bay, Magnetic Island | $\delta^{13}C$; $\delta^{18}O$ ($\delta^{18}O_{seawater}$) | Lewis (2005) | Lough et al. (2014) |


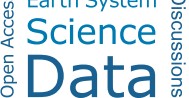

| Dataset ID | Core Name | Latitude | Longitude | Location | Primary Variables (Secondary Variables) | Record Publication(s) | Additional Core Publication(s) |
|---|---|---|---|---|---|---|---|
| LE07MAG01 | MAG01D | -19.15 | 146.87 | Geoffrey Bay, Magnetic Island | Ba; Ba/Ca; /Ca; Mg/Ca; Mn; Pr; Sm; Sr/Ca; Th; U/Ca; Y; Y/Ho (Distance) | Lewis et al. (2007); Lewis (2005) | Barnes & Lough (1992); D'Olivo & McCulloch (2022); De'ath et al. (2009); Erler et al. (2016); Lewis (2005); Lough (2007); Lough (2011a); Lough (2011b); Lough & Barnes (1997) |
| LE07NEL01 | NEL01D | -19 | 147 | Nelly Bay, Magnetic Island | Ba; Ba/Ca; /Ca; Mg/Ca; Mn; Pr; Sm; Sr/Ca; Th; U/Ca; Y; Y/Ho (Distance) | Lewis et al. (2007); Lewis (2005) | Lough et al. (2014) |
| LE12CID01 | CID-01A | -20.3 | 148.9 | Cid Harbour Island | Ba$^{138}$/Ca; Ba/Ca; B/Ca; Ca/Ca; Cd; Cd/Ca; Li/Ca; Mg/Ca; Mn; Mn/Ca; P; Pb/Ca; P/Ca; Sr/Ca; U/Ca; Y; Y/Ca (Distance) | Lewis et al. (2012) | De'ath et al. (2009); Lough (2007); Lough (2011a); Lough (2011b); Lough & Barnes (1997) |
| LE12CID02 | CID-71B | -20.254 | 148.923 | Cid Harbour Island | Ba$^{138}$/Ca; Ba/Ca; B/Ca; Ca/Ca; Cd; Cd/Ca; Li/Ca; Mg/Ca; Mn; Mn/Ca; P; Pb/Ca; P/Ca; Sr/Ca; U/Ca; Y; Y/Ca (Distance) | Lewis et al. (2012) | |





| Dataset ID | Core Name | Latitude | Longitude | Location | Primary Variables (Secondary Variables) | Record Publication(s) | Additional Core Publication(s) |
|---|---|---|---|---|---|---|---|
| LE12CID03 | CID-73B | -20.253 | 148.92 | Cid Harbour Island | Ba$^{138}$/Ca; Ba/Ca; B/Ca; Cd; Cd/Ca; Ce; Ce/Ca; Eu; Eu/Ca; Gd; Gd/Ca; La; La/Ca; Li/Ca; Lu; Lu/Ca; Mg/Ca; Mn; Mn/Ca; P; Pb/Ca; P/Ca; Pr; Pr/Ca; Sm; Sm/Ca; Sr/Ca; U/Ca; Y; Yb; Yb/Ca; Y/Ca; Zn/Ca | Lewis et al. (2012) | |
| LE12COB01 | COB-71A | -19.396 | 148.806 | Cobham Reef | Ba$^{138}$/Ca; Ba/Ca; B/Ca; Cd/Ca; Li/Ca; Mg/Ca; Mn/Ca; Pb/Ca; P/Ca; Sr/Ca; U/Ca; Y/Ca (Distance) | Lewis et al. (2012) | |
| LE12HAS01 | HWD-73B | -20.26 | 149.096 | Haslewood Island | Ba$^{138}$/Ca; Ba/Ca; B/Ca; Cd; Cd/Ca; Ce; Ce/Ca; Eu; Eu/Ca; Gd; Gd/Ca; La; La/Ca; Li/Ca; Lu; Lu/Ca; Mg/Ca; Mn; Mn/Ca; P; Pb/Ca; P/Ca; Pr; Pr/Ca; Sm; Sm/Ca; Sr/Ca; U/Ca; Y; Yb; Yb/Ca; Y/Ca; Zn/Ca (Distance) | Lewis et al. (2012) | |



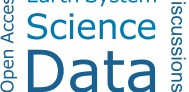

| Dataset ID | Core Name | Latitude | Longitude | Location | Primary Variables (Secondary Variables) | Record Publication(s) | Additional Core Publication(s) |
|---|---|---|---|---|---|---|---|
| LE12HKO01 | SNH-73A | -20.089 | 148.907 | Stonehaven Bay, Hook Island | Ba$^{138}$/Ca; Ba/Ca; B/Ca; Cd; Cd/Ca; Ce; Ce/Ca; Eu; Eu/Ca; Gd; Gd/Ca; La; La/Ca; Li/Ca; Lu; Lu/Ca; Mg/Ca; Mn; Mn/Ca; P; Pb/Ca; P/Ca; Pr; Pr/Ca; Sm; Sm/Ca; Sr/Ca; U/Ca; Y; Yb; Yb/Ca; Y/Ca; Zn/Ca (Distance) | Lewis et al. (2012) | |
| LE12NMI01 | NMI-73A | -20.22 | 148.811 | North Molle Island | Ba$^{138}$/Ca; Ba/Ca; B/Ca; Cd; Cd/Ca; Ce; Ce/Ca; Eu; Eu/Ca; Gd; Gd/Ca; La; La/Ca; Li/Ca; Lu; Lu/Ca; Mg/Ca; Mn; Mn/Ca; P; Pb/Ca; P/Ca; Pr; Pr/Ca; Sm; Sm/Ca; Sr/Ca; U/Ca; Y; Yb; Yb/Ca; Y/Ca; Zn/Ca (Distance) | Lewis et al. (2012) | |
| LE12REP01 | REP-71A | -20.618 | 148.865 | Repulse Island | Ba$^{138}$/Ca; Ba/Ca; B/Ca; Cd; Cd/Ca; Ce; Ce/Ca; La; La/Ca; Li/Ca; Lu; Lu/Ca; Mg/Ca; Mn; Mn/Ca; P; Pb/Ca; P/Ca; Pr; Pr/Ca; Sr/Ca; U/Ca; Y; Yb; Yb/Ca; Y/Ca; Zn/Ca (Distance) | Lewis et al. (2012) | |





| Dataset ID | Core Name | Latitude | Longitude | Location | Primary Variables (Secondary Variables) | Record Publication(s) | Additional Core Publication(s) |
|---|---|---|---|---|---|---|---|
| LE12REP02 | REP-72A | -20.617 | 148.864 | Repulse Island | Ba$^{138}$/Ca; Ba/Ca; B/Ca; Cd; Cd/Ca; Ce; Ce/Ca; La; La/Ca; Li/Ca; Lu; Lu/Ca; Mg/Ca; Mn; Mn/Ca; P; Pb/Ca; P/Ca; Pr; Pr/Ca; Sr/Ca; U/Ca; Y; Yb; Yb/Ca; Y/Ca; Zn/Ca (Distance) | Lewis et al. (2012) | |
| LE12REP03 | REP-72B | -20.617 | 148.864 | Repulse Island | Ba$^{138}$/Ca; Ba/Ca; B/Ca; Ca/Ca; Cd; Cd/Ca; Li/Ca; Mg/Ca; Mn; Mn/Ca; P; Pb/Ca; P/Ca; Sr/Ca; U/Ca; Y; Y/Ca (Distance) | Lewis et al. (2012) | |
| LE12SHW01 | SHW-82A | -20.48 | 149.069 | Shaw Island | Ba$^{138}$/Ca; Ba/Ca; B/Ca; Cd; Cd/Ca; Li/Ca; Mg/Ca; Mn; Mn/Ca; P; Pb/Ca; P/Ca; Sr/Ca; U/Ca; Y; Y/Ca (Distance) | Lewis et al. (2012) | |
| LE12SHW02 | SHW-83B | -20.48 | 149.069 | Shaw Island | Ba$^{138}$/Ca; Ba/Ca; B/Ca; Cd; Cd/Ca; Li/Ca; Mg/Ca; Mn; Mn/Ca; P; Pb/Ca; P/Ca; Sr/Ca; U/Ca; Y; Y/Ca (Distance) | Lewis et al. (2012) | |
| LE12WHI01 | WHI-34A | -20.169 | 148.957 | Whitsunday Island | Ba$^{138}$/Ca; Ba/Ca; B/Ca; Cd; Cd/Ca; Ce; Ce/Ca; La; La/Ca; Li/Ca; Mg/Ca; Mn; Mn/Ca; P; Pb/Ca; P/Ca; Sr/Ca; U/Ca; Y; Y/Ca (Distance) | Lewis et al. (2012) | De'ath et al. (2009) |





| Dataset ID | Core Name | Latitude | Longitude | Location | Primary Variables (Secondary Variables) | Record Publication(s) | Additional Core Publication(s) |
|---|---|---|---|---|---|---|---|
| LE16GPI01 | PAM2.0 | -18.733 | 146.567 | Great Palm Island | Luminescence visual indices | Leonard et al. (2016) | |
| LE16GPI02 | PAM3.1 | -18.733 | 146.567 | Great Palm Island | Luminescence visual indices | Leonard et al. (2016) | |
| LE16GPI03 | PAM5.0 | -18.733 | 146.567 | Great Palm Island | Luminescence visual indices | Leonard et al. (2016) | |
| LE18GFB01 | GFB33B | -19.154 | 146.866 | Geoffrey Bay, Magnetic Island | Ba$^{138}$/Ca; Ba/Ca; B/Ca; Cd/Ca; Li/Ca; Mg/Ca; Mn/Ca; Pb/Ca; P/Ca; Sr/Ca; U/Ca; Y/Ca (Distance) | Lewis et al. (2018); D'Olivo & McCulloch (2022) | Saha et al. (2019b); Saha et al. (2021) |
| LE18GFB02 | GFB34A | -19.164 | 146.853 | Geoffrey Bay, Magnetic Island | Ba$^{138}$/Ca; Ba/Ca; B/Ca; Cd/Ca; Li/Ca; Mg/Ca; Mn/Ca; Pb/Ca; P/Ca; Sr/Ca; U/Ca; Y/Ca (Distance) | Lewis et al. (2018); D'Olivo & McCulloch (2022) | |
| LE18HAV01 | HAV32A | -18.835 | 146.544 | Havannah Island | Ba$^{138}$/Ca; Ba/Ca; B/Ca; Cd/Ca; Li/Ca; Mg/Ca; Mn/Ca; Pb/Ca; P/Ca; Sr/Ca; U/Ca; Y/Ca (Distance) | Lewis et al. (2018); D'Olivo & McCulloch (2022) | Lough et al. (2015) |
| LE18HAV02 | HAV34A | -18.837 | 146.548 | Havannah Island | Ba$^{138}$/Ca; Ba/Ca; B/Ca; Cd/Ca; Li/Ca; Mg/Ca; Mn/Ca; Pb/Ca; P/Ca; Sr/Ca; U/Ca; Y/Ca (Distance) | Lewis et al. (2018); D'Olivo & McCulloch (2022) | Lough et al. (2015) |
| LE18NEL01 | NEL29A | -19.165 | 146.854 | Nelly Bay, Magnetic Island | Ba$^{138}$/Ca; Ba/Ca; B/Ca; Cd/Ca; Li/Ca; Mg/Ca; Mn/Ca; Pb/Ca; P/Ca; Sr/Ca; U/Ca; Y/Ca (Distance) | Lewis et al. (2018); D'Olivo & McCulloch (2022) | Cantin & Lough (2014); De'ath et al. (2009); Lough et al. (2014) |




| Dataset ID | Core Name | Latitude | Longitude | Location | Primary Variables (Secondary Variables) | Record Publication(s) | Additional Core Publication(s) |
|---|---|---|---|---|---|---|---|
| LE18NEL02 | NEL39A | -19.169 | 146.85 | Nelly Bay, Magnetic Island | Ba$^{138}$/Ca; Ba/Ca; B/Ca; Cd/Ca; Li/Ca; Mg/Ca; Mn/Ca; Pb/Ca; P/Ca; Sr/Ca; U/Ca; Y/Ca (Distance) | Lewis et al. (2018); D'Olivo & McCulloch (2022) | Cantin & Lough (2014); De'ath et al. (2009) |
| LE18PAN01 | PAN31B | -18.813 | 146.427 | Pandora Reef | Ba$^{138}$/Ca; Ba/Ca; B/Ca; Cd/Ca; Li/Ca; Mg/Ca; Mn/Ca; Pb/Ca; P/Ca; Sr/Ca; U/Ca; Y/Ca (Distance) | Lewis et al. (2018); D'Olivo & McCulloch (2022) | Cantin & Lough (2014); De'ath et al. (2009) |
| LE18PAN02 | PAN36B | -18.813 | 146.427 | Pandora Reef | Ba$^{138}$/Ca; Ba/Ca; B/Ca; Cd/Ca; Li/Ca; Mg/Ca; Mn/Ca; Pb/Ca; P/Ca; Sr/Ca; U/Ca; Y/Ca (Distance) | Lewis et al. (2018); D'Olivo & McCulloch (2022) | Cantin & Lough (2014); De'ath et al. (2009) |
| LE18PEL01 | PEL30A | -18.538 | 146.49 | Pelorus Island | Ba$^{138}$/Ca; Ba/Ca; B/Ca; Cd/Ca; Li/Ca; Mg/Ca; Mn/Ca; Pb/Ca; P/Ca; Sr/Ca; U/Ca; Y/Ca (Distance) | Lewis et al. (2018) | |
| LE18PEL02 | PEL33B | -18.538 | 146.491 | Pelorus Island | Ba$^{138}$/Ca; Ba/Ca; B/Ca; Cd/Ca; Li/Ca; Mg/Ca; Mn/Ca; Pb/Ca; P/Ca; Sr/Ca; U/Ca; Y/Ca (Distance) | Lewis et al. (2018) | |
| LE19HIG01 | HI 12.1 | -17.161 | 146.007 | High Island | Ba/Ca; Ce; Dy; Er; Eu; Gd; Ho; La; Lu; Nd; Pr; Sm; Tb; Tm; Y; Yb; Zr | Leonard et al. (2019) | |





| Dataset ID | Core Name | Latitude | Longitude | Location | Primary Variables (Secondary Variables) | Record Publication(s) | Additional Core Publication(s) |
|---|---|---|---|---|---|---|---|
| LE19RUS01 | FRI 12.1 | -17.224 | 146.09 | Russell Island | Ba/Ca; Ce; Dy; Er; Eu; Gd; Ho; La; Lu; Nd; Pr; Sm; Tb; Tm; Y; Yb; Zr | Leonard et al. (2019) | |
| LE19RUS02 | FRI 12.3 | -17.228 | 146.091 | Russell Island | Ba/Ca; Ce; Dy; Er; Eu; Gd; Ho; La; Lu; Nd; Pr; Sm; Tb; Tm; Y; Yb; Zr | Leonard et al. (2019) | |
| LE19SUD01 | SUD 12.1 | -17.026 | 146.211 | Sudbury Cay | Ba/Ca; Ce; Dy; Er; Eu; Gd; Ho; La; Lu; Nd; Pr; Sm; Tb; Tm; Y; Yb; Zr | Leonard et al. (2019) | |
| LO11ACH01 | ACH01A | -18.95 | 146.65 | Acheron Island | Luminescence Range | Lough (2011b); Lough (2011a) | Barnes & Lough (1992) |
| LO11BRO01 | BRO01A | -18.15 | 146.28 | Brook Island | Luminescence Range | Lough (2011a); Lough (2011b); D'Olivo & McCulloch (2022) | Barnes & Lough (1992); De'ath et al. (2009); Hendy et al. (2002); Hendy et al. (2003a); Hendy et al. (2003b); Hendy et al. (2007); Hendy et al. (2012); Lough (2007); Lough & Barnes (1997) |
| LO11CID01 | CID01A | -20.27 | 148.93 | Cid Harbour Island | Luminescence Range | Lough (2011a); Lough (2011b) | De'ath et al. (2009); Lewis et al. (2012); Lough (2007); Lough & Barnes (1997) |
| LO11CNR01 | CNR01B | -15.13 | 145.32 | Conical Rock Reef | Luminescence Range | Lough (2011a) | De'ath et al. (2009); Lough & Barnes (1997) |
| LO11COO01 | COO01E | -18.03 | 146.18 | Coombe Island | Luminescence Range | Lough (2011a); Lough (2011b) | Barnes & Lough (1992); De'ath et al. (2009); Lough (2007); Lough & Barnes (1997) |
| LO11DUN01 | DUN02A | -17.95 | 146.17 | Dunk Island | Luminescence Range | Lough (2011a); Lough (2011b) | |





| Dataset ID | Core Name | Latitude | Longitude | Location | Primary Variables (Secondary Variables) | Record Publication(s) | Additional Core Publication(s) |
|---|---|---|---|---|---|---|---|
| LO11GPI01 | GPI02A | -18.68 | 146.58 | Great Palm Island | Luminescence Range | Lough (2011a); Lough (2011b) | De'ath et al. (2009); Hendy et al. (2002); Hendy et al. (2003a); Hendy et al. (2003b); Hendy et al. (2007); Lough (2007); Lough & Barnes (1997) |
| LO11HAV01 | HAV01A | -18.85 | 146.55 | Havannah Island | Luminescence Range | Lough (2011a); Lough (2011b); Palmer et al. (2015) | D'Olivo & McCulloch (2022); De'ath et al. (2009); Erler et al. (2020); Hendy et al. (2002); Hendy et al. (2003a); Hendy et al. (2003b); Hendy et al. (2007); Hendy et al. (2012); Isdale et al. (1998); Lough (2007); Lough & Barnes (1997); Lough et al. (2015); McCulloch et al. (2003) |
| LO11HKO01 | HKO01B | -20.07 | 148.95 | Hook Island | Luminescence Range | Lough (2011a); Lough (2011b) | Barnes & Lough (1992); De'ath et al. (2009); Lough (2007); Lough & Barnes (1997) |
| LO11HKO02 | SNH01A | -20.1 | 148.9 | Stonehaven Bay, Hook Island | Luminescence Range | Lough (2011a); Lough (2011b) | De'ath et al. (2009); Lough & Barnes (1997) |
| LO11HMP01 | HMP01B | -23.2 | 151 | Humpy Island | Luminescence Range | Lough (2011a); Lough (2011b); D'Olivo & McCulloch (2022) | Lough (2007); |



| Dataset ID | Core Name | Latitude | Longitude | Location | Primary Variables (Secondary Variables) | Record Publication(s) | Additional Core Publication(s) |
|---|---|---|---|---|---|---|---|
| LO11LUP01 | LUP01C | -20.28 | 149.12 | Lupton Island | Luminescence Range | Lough (2011a); Lough (2011b) | Barnes & Lough (1992); De'ath et al. (2009); Lough (2007); Lough & Barnes (1997) |
| LO11MAG01 | MAG01D | -19.15 | 146.87 | Geoffrey Bay, Magnetic Island | Luminescence Range | Lough (2011a); Lough (2011b); D'Olivo & McCulloch (2022) | Barnes & Lough (1992); De'ath et al. (2009); Erler et al. (2016); Lewis (2005); Lewis et al. (2007); Lough (2007); Lough & Barnes (1997) |
| LO11NMI01 | NMI01B | -20.23 | 148.8 | North Molle Island | Luminescence Range | Lough (2011a); Lough (2011b) | Barnes & Lough (1992); De'ath et al. (2009); Lough (2007); Lough & Barnes (1997) |
| LO11NOR01 | NOR01B | -17.2 | 146.1 | Normanby Island | Luminescence Range | Lough (2011a); Lough (2011b); D'Olivo & McCulloch (2022) | Barnes & Lough (1992); De'ath et al. (2009); Lough (2007) |
| LO11PAN01 | PAN07B | -18.82 | 146.43 | Pandora Reef | Luminescence Range | Lough (2011a) | Barnes & Lough (1992); De'ath et al. (2009); Lough (2007) |
| LO11SMI01 | SMI01C | -20.27 | 148.83 | South Molle Island | Luminescence Range | Lough (2011a); Lough (2011b) | Barnes & Lough (1992); De'ath et al. (2009); Lough (2007); Lough & Barnes (1997) |
| LO14NEL01 | NEL03A | -19 | 147 | Nelly Bay, Magnetic Island | Luminescence Range | Lough et al. (2014) | |
| LO14NEL02 | NEL03D | -19 | 147 | Nelly Bay, Magnetic Island | Luminescence Range | Lough et al. (2014) | Lewis (2005) |
| LO14NEL03 | NEL07B | -19 | 147 | Nelly Bay, Magnetic Island | Luminescence Range | Lough et al. (2014) | |





| Dataset ID | Core Name | Latitude | Longitude | Location | Primary Variables (Secondary Variables) | Record Publication(s) | Additional Core Publication(s) |
|---|---|---|---|---|---|---|---|
| LO14NEL04 | NEL07C | -19 | 147 | Nelly Bay, Magnetic Island | Luminescence Range | Lough et al. (2014) | |
| LO14NEL05 | NEL01D | -19 | 147 | Nelly Bay, Magnetic Island | Luminescence Range | Lough et al. (2014) | Lewis (2005); Lewis et al. (2007) |
| LO14NEL06 | NEL29A | -19.1646 | 146.8658 | Nelly Bay, Magnetic Island | Luminescence Range | Lough et al. (2014) | Cantin & Lough (2014); De'ath et al. (2009); Lewis et al. (2018) |
| LO14NEL07 | NEL35A | -19.164 | 146.854 | Nelly Bay, Magnetic Island | Luminescence Range | Lough et al. (2014) | Cantin & Lough (2014); De'ath et al. (2009) |
| LO14NEL08 | NEL39B | -19.169 | 146.8502 | Nelly Bay, Magnetic Island | Luminescence Range | Lough et al. (2014) | Cantin & Lough (2014); De'ath et al. (2009) |
| LO15HAV01 | HAV01A | -18.85 | 146.55 | Havannah Island | Luminescence Range | Lough et al. (2015) | D'Olivo & McCulloch (2022); De'ath et al. (2009); Erler et al. (2020); Hendy et al. (2002); Hendy et al. (2003a); Hendy et al. (2003b); Hendy et al. (2007); Hendy et al. (2012); Isdale et al. (1998); Lough (2007); Lough (2011a); Lough (2011b); Lough & Barnes (1997); McCulloch et al. (2003); Palmer et al. (2015) |
| LO15HAV02 | HAV31B | -18.837 | 146.5483 | Havannah Island | Luminescence Range | Lough et al. (2015) | |
| LO15HAV03 | HAV32A | -18.8351 | 146.544 | Havannah Island | Luminescence Range | Lough et al. (2015) | Lewis et al. (2018) |
| LO15HAV04 | HAV33A | -18.837 | 146.5483 | Havannah Island | Luminescence Range | Lough et al. (2015) | |



| Dataset ID | Core Name | Latitude | Longitude | Location | Primary Variables (Secondary Variables) | Record Publication(s) | Additional Core Publication(s) |
|---|---|---|---|---|---|---|---|
| LO15HAV05 | HAV34A | -18.837 | 146.5483 | Havannah Island | Luminescence Range | Lough et al. (2015) | Lewis et al. (2018) |
| MA00MYR01 | Myr-2 | -18.266 | 147.383 | Myrmidon Reef | Sr/Ca (Distance) | Marshall (2000); Marshall & McCulloch (2002) | Fallon (2000); Fallon et al. (2003) |
| MA00MYR02 | Myr2-16D | -18.266 | 147.383 | Myrmidon Reef | $\delta^{13}$C; $\delta^{18}$O (Distance) | Marshall (2000) | |
| MA00MYR03 | Myr2-17D | -18.266 | 147.383 | Myrmidon Reef | Ba/Ca; Mg/Ca; Sr/Ca; U/Ca (Distance) | Marshall (2000) | |
| MA00SLY01 | St-1 | -19.2892 | 148.0519 | Stanley Reef | Sr/Ca (Distance) | Marshall (2000); Marshall & McCulloch (2002) | |
| MC03HAV01 | Havannah | -18.843 | 146.537 | Havannah Island | Ba/Ca | McCulloch et al. (2003); Lewis et al. (2007); Walther et al. (2013) | |
| MC03PAN01 | Pandora | -18.8 | 146.4 | Pandora Reef | Ba/Ca | McCulloch et al. (2003); D'Olivo & McCulloch (2022) | |
| MC17DAV01 | Davies 13-2 (D-2) | -18.8 | 147.63 | Davies Reef | Ba/Ca; B/Ca; B/Mg; $\delta^{11}$B; Li/Ca; Li/Mg; Mg/Ca; Sr/Ca; U/B; U/Ca; U/Sr ($\delta^{11}$B pH$_{cf}$) | McCulloch et al. (2017); Thompson et al. (2022) | D'Olivo et al. (2018) |
| MC17DAV02 | Davies 13-3 (D-3) | -18.8 | 147.63 | Davies Reef | Ba/Ca; B/Ca; B/Mg; $\delta^{11}$B; Li/Ca; Li/Mg; Mg/Ca; Sr/Ca; U/B; U/Ca; U/Sr ($\delta^{11}$B pH$_{cf}$) | McCulloch et al. (2017); Thompson et al. (2022) | D'Olivo et al. (2018) |
| PE05FLI01 | FL02A (FLI02A) | -17.73 | 148.43 | Flinders Reef | $\delta^{11}$B; $\delta^{13}$C ($\Omega arag$; Calcification; Extension; $\delta^{11}$B pH) | Pelejero et al. (2005) | Calvo et al. (2007) |



| Dataset ID | Core Name | Latitude | Longitude | Location | Primary Variables (Secondary Variables) | Record Publication(s) | Additional Core Publication(s) |
|---|---|---|---|---|---|---|---|
| RA17MYR01 | myra-30 | -18.278 | 147.379 | Myrmidon Reef | Ba/Ca; Mg/Ca; Sr/Ca; U/Ca; Y/Ca | Razak et al. (2017) | |
| RA17MYR02 | myra-31 | -18.278 | 147.379 | Myrmidon Reef | Ba/Ca; Mg/Ca; Sr/Ca; U/Ca; Y/Ca | Razak et al. (2017) | |
| RE19CLK01 | CLK (CLK06A) | -11.97 | 143.28 | Clerke Reef | Luminescence; Sr/Ca (Density; Distance) | Reed et al. (2019) | |
| RE19EEL01 | EEL (EELB10) | -12.5 | 143.52 | Eel Reef | $\delta^{13}C$; $\delta^{18}O$; Luminescence; Sr/Ca (Density; Distance) | Reed et al. (2019) | Barnes & Lough (1992); Lough & Barnes (2000); Lough et al. (2002) |
| RE19GBR01 | 13-050 (050B06) | -13.33 | 143.95 | Reef 13-050 | $\delta^{13}C$; $\delta^{18}O$; Luminescence; Sr/Ca (Density; Distance) | Reed et al. (2019) | Barnes & Lough (1992); Lough & Barnes (1992); De'ath et al. (2009); Lough et al. (2002); Madin et al. (2016) |
| RE19NOM01 | NOM (NOM05A) | -12.09 | 143.29 | Nomad Reef | Luminescence; Sr/Ca (Density; Distance) | Reed et al. (2019) | |
| RE19POR01 | POR (PORB03) | -12.591 | 143.408 | Portland Roads | $\delta^{13}C$; $\delta^{18}O$; Luminescence; Sr/Ca (Density; Distance) | Reed et al. (2019) | Barnes & Lough (1992); Lough & Barnes (1992); De'ath et al. (2009); Lough et al. (2002); Madin et al. (2016) |
| RO14GKI01 | GK2 | -23.1505 | 150.9739 | Great Keppel Island | Luminescence G/B | Rodriguez-Ramirez et al. (2014); Grove et al. (2015) | Saha et al. (2018a); Saha et al. (2019a); Saha et al. (2021) |
| RO14GKI02 | GK3 | -23.192 | 150.9627 | Great Keppel Island | Luminescence G/B | Rodriguez-Ramirez et al. (2014) | |
| RO14KIN01 | KR-MMA-1 | -17.767 | 146.133 | King Reef | $\delta^{18}O$; Sr/Ca ($\delta^{18}O_{seawater}$) | Roche et al. (2014) | |
| RO14KIN02 | KR-AMA-2 | -17.767 | 146.133 | King Reef | $\delta^{18}O$; Sr/Ca ($\delta^{18}O_{seawater}$) | Roche et al. (2014) | |



| Dataset ID | Core Name | Latitude | Longitude | Location | Primary Variables (Secondary Variables) | Record Publication(s) | Additional Core Publication(s) |
|---|---|---|---|---|---|---|---|
| RO14MIA01 | MI1 | -23.155 | 150.9035 | Miall Island | Luminescence G/B | Rodriguez-Ramirez et al. (2014); Grove et al. (2015) | |
| RO14MIA02 | MI2 | -23.1554 | 150.9034 | Miall Island | Luminescence G/B | Rodriguez-Ramirez et al. (2014) | |
| RO14SQR01 | SQ1 | -23.0997 | 150.8862 | Square Rocks | Luminescence G/B | Rodriguez-Ramirez et al. (2014); Grove et al. (2015) | |
| RO14SQR02 | SQ2 | -23.0997 | 150.8862 | Square Rocks | Luminescence G/B | Rodriguez-Ramirez et al. (2014); Grove et al. (2015) | |
| SA16HER01a | AR HL 3 | -23.451 | 151.93 | Heron Island | Sr/Ca | Sadler et al. (2016a) | |
| SA16HER01b | P HS 1 | -23.451 | 151.93 | Heron Island | Sr/Ca | Sadler et al. (2016b) | |
| SA16HER02a | AR HL 4 | -23.451 | 151.93 | Heron Island | Sr/Ca | Sadler et al. (2016a) | |
| SA16HER02b | P HS 3 | -23.451 | 151.93 | Heron Island | Sr/Ca | Sadler et al. (2016b) | |
| SA16HER03a | AR HL 6 | -23.451 | 151.93 | Heron Island | Sr/Ca | Sadler et al. (2016a) | |
| SA16HER03b | P HS 4 | -23.451 | 151.93 | Heron Island | Sr/Ca | Sadler et al. (2016b) | |
| SA16HER04a | AR HS 3 | -23.451 | 151.93 | Heron Island | Sr/Ca | Sadler et al. (2016a) | |
| SA16HER04b | P HS 7 | -23.451 | 151.93 | Heron Island | Sr/Ca | Sadler et al. (2016b) | |
| SA16HER05a | AR HS 5 | -23.451 | 151.93 | Heron Island | Sr/Ca | Sadler et al. (2016a) | |
| SA16HER05b | R9C (track 1) | -23.451 | 151.93 | Heron Island | Sr/Ca | Sadler et al. (2016b) | |
| SA16HER06a | AR HS 6 | -23.451 | 151.93 | Heron Island | Sr/Ca | Sadler et al. (2016a) | |
| SA16HER06b | R15B3 | -23.451 | 151.93 | Heron Island | Sr/Ca | Sadler et al. (2016b) | |





| Dataset ID | Core Name | Latitude | Longitude | Location | Primary Variables (Secondary Variables) | Record Publication(s) | Additional Core Publication(s) |
|---|---|---|---|---|---|---|---|
| SA21GFB01 | GFB33B | -19.1543 | 146.8656 | Geoffrey Bay, Magnetic Island | Ba/Ca; Ce; Dy; Eu; Gd; Ho; La; Lu; Mn/Ca; Nd; Nd/Yb; Pr; Sm; Sr/Ca; Tb; Tm; Y; Yb; Y/Ca (Ce/Ce*; ΣREE/Ca) | Saha et al. (2021); Saha et al. (2019b) | Lewis et al. (2018) |
| SA21GKI01 | GK2 | -23.17 | 150.98 | Great Keppel Island | Ba/Ca; Ce; Dy; Eu; Gd; Ho; La; Lu; Mg/Ca; Mn/Ca; Nd; Pr; Sm; Sr/Ca; Tb; Tm; U/Ca; V/Ca; Y; Yb; Y/Ca | Saha et al. (2021); Saha et al. (2018a); Saha et al. (2019a) | Rodriguez-Ramirez et al. (2014) |
| SA21RAT01 | RI2 | -23.7663 | 151.3178 | Rat Island | Ba/Ca; Ce; Dy; Eu; Gd; Ho; La; Lu; Mn/Ca; Nd; Pr; Sm; Sr/Ca; Tb; Tm; Y; Yb; Y/Ca | Saha et al. (2021); Saha et al. (2018b) | |
| WA13HAV01 | Havannah (Hav08_2) | -18.8396 | 146.5502 | Havannah Island | Ba/Ca | Walther et al. (2013); D'Olivo & McCulloch (2022) | |
| WA13MYR01 | Myrmidon (MYR_S5) | -18.2614 | 147.3765 | Myrmidon Reef | Ba/Ca | Walther et al. (2013) | |
| WA19ARL01 | 10AR2 | -16.6381 | 146.1036 | Arlington Reef | $\delta^{98}$Mo; Fe; Mn; Ti | Wang et al. (2019); Deng et al. (2014) | Chen et al. (2021); Xiao et al. (2020) |
| WE09ARL01_1 | AREO 4 | -16.667 | 146.109 | Arlington Reef | Ba/Ca; $\delta^{11}$B; $\delta^{13}$C; $\delta^{18}$O; Mg/Ca; Sr/Ca ($\delta^{11}$B pH) | Wei et al. (2009); D'Olivo & McCulloch (2017); Chen et al. (2021) | Wei et al. (2015) |
| WE09ARL01_2 | AREO 4 | -16.674 | 146.109 | Arlington Reef | Ba/Ca; $\delta^{11}$B; $\delta^{13}$C; $\delta^{18}$O; Mg/Ca; Sr/Ca ($\delta^{11}$B pH) | Wei et al. (2009); D'Olivo & McCulloch (2017); Chen et al. (2021) | Wei et al. (2015) |
| WU21CUR01 | SEN01C | -23.7547 | 151.3169 | South End, Curtis Island | B/Ca; Li/Ca; Mg/Ca; Sr/Ca; U/Ca | Wu et al. (2021a) | Cantin et al. (2018) |



| Dataset ID | Core Name | Latitude | Longitude | Location | Primary Variables (Secondary Variables) | Record Publication(s) | Additional Core Publication(s) |
|---|---|---|---|---|---|---|---|
| WU21HUM01 | HUM03A | -23.4035 | 151.1472 | Hummocky Island | B/Ca; Li/Ca; Mg/Ca; Sr/Ca; U/Ca | Wu et al. (2021a) | Cantin et al. (2018) |
| WU21HUM02 | HUM04B | -23.4024 | 151.146 | Hummocky Island | B/Ca; Li/Ca; Mg/Ca; Sr/Ca; U/Ca | Wu et al. (2021a) | |
| WU21MAS01a | MAS02A | -23.532 | 151.7403 | Mast Head Island | B/Ca; Li/Ca; Mg/Ca; Sr/Ca; U/Ca | Wu et al. (2021a) | Cantin et al. (2018) |
| WU21MAS01b | MAS01E | -23.5325 | 151.746 | Mast Head Island | $\Delta14C$ | Wu et al. (2021b); Cantin et al. (2018) | |
| WU21SHW01 | SHW82C | -20.4796 | 149.0691 | Shaw Island | B/Ca; Li/Ca; Mg/Ca; Sr/Ca; U/Ca | Wu et al. (2021a) | |
| WU21SMI01 | SMI81A | -20.2588 | 148.8283 | South Molle Island | B/Ca; Li/Ca; Mg/Ca; Sr/Ca; U/Ca | Wu et al. (2021a) | |
| XI20ARL01 | 10AR2 | -16.6381 | 146.1036 | Arlington Reef | $\delta^{13}C$; $\delta^{18}O$; $\delta^{66}Zn$; Sr/Ca; Zn | Xiao et al. (2020); | Chen et al. (2021); Deng et al. (2014); Wang et al. (2019) |

Note: Additional publication referenced in in GBRCD includes:

- Chakraborty & Ramesh (1993, 1997) for CH93SLY01 & CH93SLY02.
- Davies & Hopley (1983) for MA00MYR02 & MA00MYR03.
- Kamber et al. (2005) for JU08KES01b & JU08RTI01b
- Min et al. (1995) for HE02GBR01, LE07MAG01 & LE07NEL01.





## Appendix B – GBRCD to LiPD Field Translation

**Table B1. GBRCD to LiPD time series (TS) field translation dictionary**

| GBRCD Field | LiPD Field | Notes |
|---|---|---|
| Age | year | LiPD standard age unit is yr AD so this description is used for the LiPD version rather than yr CE which is used for the CSV version. yr CE and yr AD are identical values. |
| cdata_datasetID | dataSetName | |
| cdata_coreName | paleoData_core | |
| cdata_altCoreName | NA | Joined with cdata_coreName, separated by ";". |
| cdata_collectTime | collectionYear | |
| cdata_minYear | minYear | Values for LiPD extracted from GBRCD record 'Age' field. |
| cdata_maxYear | maxYear | |
| cdata_archiveSpecies | paleoData_sensorSpecies | |
| cdata_isDatabaseDuplicate | paleoData_gbrIsDatabaseDuplicate | |
| cdata_dataCoverageGroup | paleoData_gbrDataCoverageGroup | |
| cdata_coralNotes | notes | |
| geo_latitude | geo_latitude | |
| geo_longitude | geo_longitude | |
| geo_siteName | geo_siteName | |
| geo_siteName2 | geo_location | |
| geo_elevation | geo_elevation | |
| geo_elevationNotes | geo_notes | |
| dating_UThDate | paleoData_gbrUThDate | |
| Dating_UThDateError | paleoData_gbrUThDateUncertainty | |
| dating_14CDate | paleoData_gbr14CDate | |
| Dating_14CDateError | paleoData_gbr14CDateUncertainty | |
| dating_notes | paleoData_gbrDatingNotes | |
| meths_primaryVariablesList | NA | List of variables intrinsic to LiPD and therefore unnecessary to translate as a separate field in LiPD format. |
| meths_additionalVariablesList | NA | |
| meths_TEMethod | paleoData_measurementMethod | GBRCD CSV fields are split by measurement type whereas for the LiPD TS a single field name is used, and the information is grouped with the variable. Where |
| meths_TEMethodMachine | paleoData_measurementInstrument | |
| meths_isotopeMethod | paleoData_measurementMethod | |
| meths_isotopeMethodMachine | paleoData_measurementInstrument | |
| meths_luminMethod | paleoData_measurementMethod | |



| GBRCD Field | LiPD Field | Notes |
|---|---|---|
| meths_luminMethodMachine | paleoData_measurementInstrument | multiple methods and/or instruments have been used for one proxy both are available in the relevant field separated by ';'. |
| meths_altMethodInfo | paleoData_measurementMethod | |
| meths_methodNotes | paleoData_notes | |
| meths_hasResolutionNominal | paleoData_samplingResolution | |
| meths_resolutionMin | NA | |
| meths_resolutionMax | NA | Values intrinsic to LiPD format. |
| meths_resolutionMean | NA | |
| meths_resolutionMedian | NA | |
| meths_isAnomaly | paleoData_gbrIsAnomaly | |
| meths_chronologyNotes | paleoData_gbrChronologyNotes | |
| meths_coralExtensionRate | paleoData_coralExtensionRate | |
| meths_coralExtensionRateNotes | paleoData_coralExtensionRateNotes | |
| meths_tissueThickness | paleoData_coralTissueThickness | |
| meths_jcpMeasured | paleoData_jcpUsed | |
| meths_jcpSrCaValue | paleoData_jcpMeasured | |
| meths_SrCaAnalyticalPrecision | paleoData_uncertaintyAnalytical | |
| meths_SrCaAnalyticalPrecisionUnits | paleoData_uncertaintyAnalyticalUnits | GBRCD CSV fields are split by variables whereas for the LiPD TS a single field name is used (for the uncertainty and for the uncertainty units) and the information is grouped with the variable. |
| meths_UCaAnalyticalPrecision | paleoData_uncertaintyAnalytical | |
| meths_UCaAnalyticalPrecisionUnits | paleoData_uncertaintyAnalyticalUnits | |
| meths_d18OAnalyticalPrecision | paleoData_uncertaintyAnalytical | |
| meths_d18OAnalyticalPrecisionUnits | paleoData_uncertaintyAnalyticalUnits | |
| meths_altPrecisionList | N/A | |
| meths_altAnalyticalPrecision | paleoData_analyticalUncertainty | |
| meths_altAnalyticalPrecisionUnits | paleoData_uncertaintyAnalyticalUnits | |
| meths_archiveDiagenesisCheck | paleoData_measurementMaterialScreening | |
| calib_isSSTCalibration | gbrIsSstCalibration | |
| calib_useSSTCalibration | gbrUseSstCalibration | |
| calib_hasAlternateCalibration | gbrHasAlternateCalibration | |
| calib_notes | calibration_notes | |
| calib_isComposite | gbrIsCompositeCalibration | |
| calib_compositeList | gbrCompositeCalibrationList | |
| calib_SSTdata | calibration_targetDataset | |
| calib_fitPeriod | calibration_datasetRange | |
| calib_hasCentredEquation | N/A | The GBRCD field is a flag for information occurring in calib_centredEquationNotes and is not used in the LiPD format. |
| calib_centredEquationNotes | calibration_gbrCentredEquationNotes | |
| calib_method | calibration_method | |



| GBRCD Field | LiPD Field | Notes |
|---|---|---|
| calib1_equationIntercept | calibration_equationIntercept | |
| calib1_equationSlope | calibration_equationSlope | |
| calib1_equationSlopeUncertainty | calibration_equationSlopeUncertainty | GBRCD CSV fields are split by |
| calib1_equationR2 | calibration_equationR2 | measurement type whereas for |
| calib2_equationIntercept | calibration_equationIntercept | the LiPD TS a single field name |
| calib2_equationSlope | calibration_equationSlope | is used (for the intercept, slope, |
| calib2_equationSlopeUncertainty | calibration_equationSlopeUncertainty | slope uncertainty and $r^2$) and the |
| calib2_equationR2 | calibration_equationR2 | information is grouped with the |
| calib3_equationIntercept | calibration_equationIntercept | variable. |
| calib3_equationSlope | calibration_equationSlope | |
| calib3_equationSlopeUncertainty | calibration_equationSlopeUncertainty | |
| calib3_equationR2 | calibration_equationR2 | |
| pub1_firstauthor | pub1_firstAuthor | |
| pub1_year | pub1_year | |
| pub1_doi | pub1_doi | |
| pub1_citation | pub1_citation | |
| pub1_authors | pub1_author | |
| pub1_title | pub1_title | |
| pub1_journal | pub1_journal | |
| pub1_doiData | pub1_doiData | |
| pub1_altDataURL | pub1_altDataURL | |
| pub2_firstauthor | pub2_firstAuthor | |
| pub2_year | pub2_year | |
| pub2_doi | pub2_doi | |
| pub2_citation | pub2_citation | |
| pub2_authors | pub2_author | |
| pub2_title | pub2_title | |
| pub2_journal | pub2_journal | |
| pub2_altDataURL | pub2_altDataURL | |
| pub3_firstauthor | pub3_firstAuthor | |
| pub3_year | pub3_year | |
| pub3_doi | pub3_doi | |
| pub3_citation | pub3_citation | |
| pub3_authors | pub3_author | |
| pub3_title | pub3_title | |
| pub3_journal | pub3_journal | |
| pub3_altDataUrl | pub3_altDataUrl | |
| pub_isCoreIDOtherStudy | gbrIsCoreIDOtherStudy | |
| pub_coreIDOtherStudyList | gbrCoreIDOtherStudyList | |




## Appendix C – GBRCD Additional Coral Information

Several of the coral records were measured in specific contexts, had additional processing to recreate the original published age model in the GBR database or are published in the GBR database for the first time. This contextual and methodological information is summarised here and is included in the relevant notes field(s) in the GBR database metadata table.

### C1 Split Datasets

In the GBRCD four corals and their related records from three publications were given split IDs due to differences in the data, such as resolution and/or age model. The Alibert et al. (2003) (Davies-2; GBRCD ID: AL03DAV01_1 & AL03DAV01_2. PAN 98-2; AL03PAN01_1 from AL03PAN01_2, AL03PAN01_3, & AL03PAN01_4), Druffel & Griffin (1999) (Abraham-1; DR99ABR01_1 & DR99ABR01_2) and Wei et al. (2009) Arlington Reef (AREO 4; WE09ARL01_1 & WE09ARL01_2) coral records were split due to differences in the resolution of the data. AREO4 (WE09ARL01_1 & WE09ARL01_2) was split where the resolution changed from quinquennial (1800–1939) to annual (1940–2004) for all measured variables. The Abraham-1 (DR99ABR01_1 & DR99ABR01_2) coral record was previously archived separately ($\delta^{13}$C and $\delta^{18}$O is ~annual, and the $\Delta^{14}$C is ~biennial), and this was maintained in the database. Davies-2 (AL03DAV01_1 & AL03DAV01_2) and PAN 98-2 (AL03PAN01_1 from AL03PAN01_2, AL03PAN01_3 & AL03PAN01_4) records were split due to differences in resolutions arising from two methods of measuring trace elements (thermal ionisation mass spectrometer (TIMS) vs laser ablation inductively coupled mass spectrometer (LA-ICP-MS)). Additionally, the LA-ICP-MS measurements from Alibert et al. (2003) were not combined into a single record as there were three distinct age models for the data and each of these was presented in the original publication.

### C2 Additions, modifications and PhD theses

The GBRCD includes 15 records from five publications where the age model was created for archiving in the GBRCD. As described in the main manuscript, a base level of information was required to archive each record in the GBRCD, notably each dataset should have an age model with an age assigned to each data point. A brief description of the method(s) used to create the age model for these records, as well any change(s) in the record compared to the previously published version, is described below.

Additionally, as PhD theses can be more difficult to access than published journal articles, a brief description of relevant information is supplied for six records from three PhD theses.

#### C2.1 Chakraborty (1993) – PhD thesis

SR coral records (track 1 and track 2) (CH93SLY01 & CH93SLY02) were collected from Stanley Reef in December 1986 (Chakraborty et al. 2000). The coral was prepared and analysed for stable isotopes along two tracks, track 1 close to the central growth axis and track 2 ~20º off the axis, as described in Chakraborty et al. (2000). Methods were as described in Chakraborty & Ramesh (1993, 1997). The original coral age model was based on Comprehensive Ocean-Atmosphere Data Set (COADS) SST data, however, since COADS is missing months of data for the Stanley Reef region, the HadISST1.1 data (grid centred on 19.5º S, 148.5º E; Raynor et al. 2003) was used for archiving in the GBRCD. A new age model was created to obtain ages based on linear interpolation using QAnalySeries to align $\delta^{18}$O minima (maxima) with SST maxima (minima) for quarterly averaged HadISST data. The age model was based on available information; however, the records published in Chakraborty et al. (2000) have additional $\delta^{18}$O measurements not available to be archived in the GBRCD. There was no change in the SR track 1 age model, but for SR track 2, ~two years of data (summer 1978 to winter 1980) do not have sufficient data points to pair with the SST data and so these years were not included in the database.



**C2.2 Ellis et al. (2019)**

PET12 (EL19OTI01; Ellis et al. 2019) was harvested by J. Mallela in May 2012 from Pete's Bay in the One Tree Island lagoon. The original PET12 chronology published in Ellis et al. (2019) was based on Sr/Ca maxima and minima, and was applied to 3 pieces of PET-12 (PET12-1, 2 and 3) after data smoothing with a 10-point running mean and then interpolated to a monthly time series. LA-ICP-MS data from an additional piece of PET-12 (PET-12-4) was combined with the data from the other pieces (PET12-1, 2 and 3) (Fig. C1). To align the entire PET12 LA-ICP-MS dataset with the standard procedure used by other

LA-ICP-MS datasets in the GBRCD, the original data smoothed with a 10-point running mean was further reduced with a 10-point mean. A new chronology was then assigned to the dataset using QAnalySeries to tie Sr/Ca maxima (minima) with SST minima (maxima) using a weekly averaged NOAA Reyn_SmithOIv2 weekly SST dataset (Reynolds et al. 2002) centred on 23.5° S, 152.5° E. Scanning electron microscopy (SEM) analysis of an alternate slice of the PET12 coral by A. Arzey indicated variable and patchy diagenetic alteration with evidence of micro-borers and secondary aragonite crystals generally 0-10 µm in

length, but up to 20 µm length observed in some sections (Fig. C2).

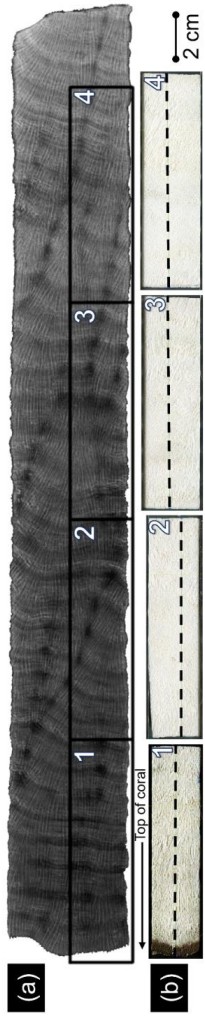

**Figure C1. PET12 (EL19OTI01) (a) X-radiograph (negative) of PET12 (EL19OTI01) coral. Black boxes indicate the 4 pieces analysed and are labelled with the piece number (1–4 in the top of the box). (b) Photos of PET12 pieces 1–4 and dashed black lines indicate the track analysed by LA-ICP-MS for each section of the record included in EL19OTI01. X-radiograph and photos supplied**
**by J. Mallela.**

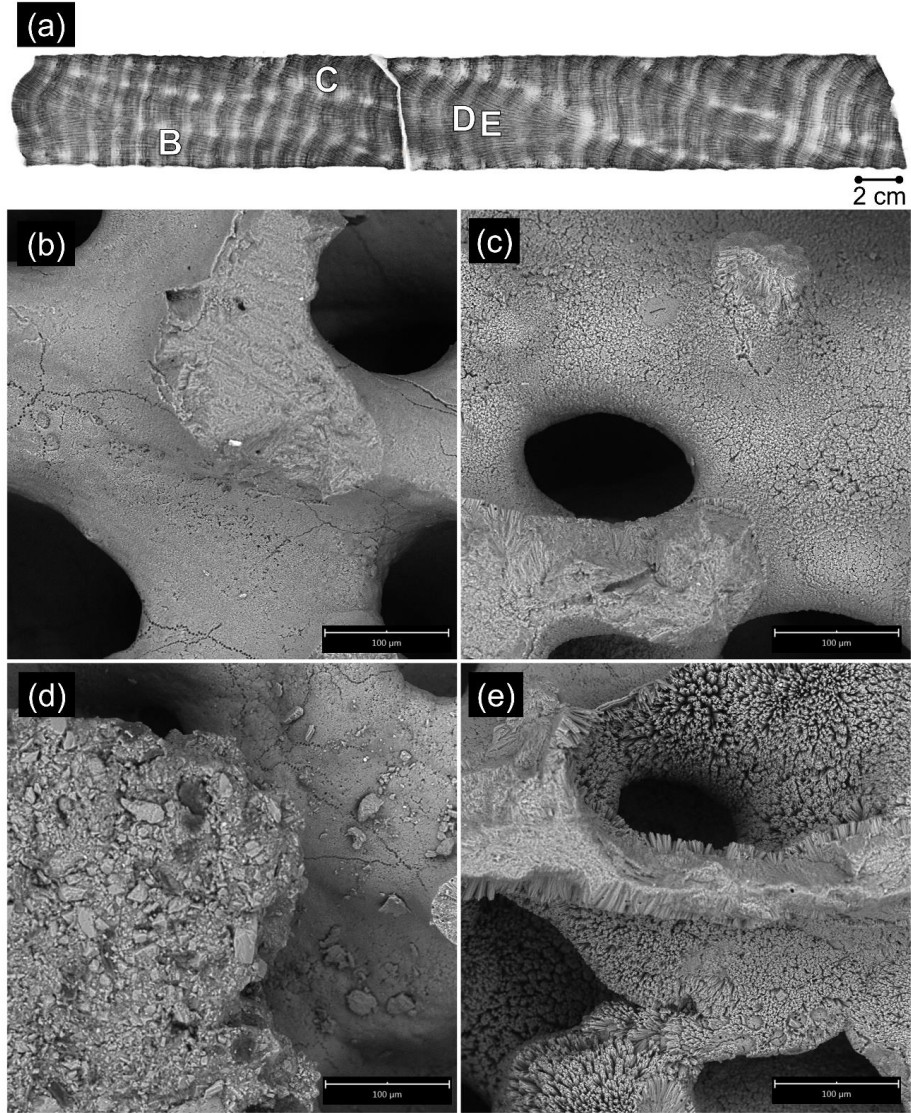

**Figure C2. PET12 (EL18OTI01) SEM images from alternate slice of PET12 coral. (a) Approximate locations of SEM images labelled B–E on coral X-radiograph. X-radiograph by PRP Diagnostic Imaging Wollongong. (b) to (e) Select SEM images from PET12 (EL19OTI01); (b) No secondary aragonite. Evidence of micro-borers, (c) Low-level diagenetic alteration with 100% cover of secondary aragonite needles ≤5 µm and evidence of micro-borers, (d) Bioclastic cement plug above relatively pristine coral surface with cement chip(e) 90% cover of secondary aragonite needles of 5–20 µm. Scale bar indicates 100 µm (b–e).**

**C2.3 Fallon (2000) – PhD thesis**

Pan 1-98b (FA00PAN01) is a *Porites* sp. coral collected from Pandora Reef in October 1998, and is an additional core from the same coral as Pan 1-98a (related record published in Fallon et al. (2003); FA03PAN01). Pan 1-98b was dead on top when collected and likely stopped recording environmental information in March/April 1998, as described in Fallon (2000). A 46 mm section from Pan 1-98b covering the period 1996–1998 was analysed by LA-ICP-MS as described in Fallon et al. (2003), the method used an ArF excimer laser (193 nm) that was masked to illuminate a 50x500 µm rectangle on the coral surface and a laser pulsed at 5 Hz using a 50 mJ power setting.



### C2.4 Jupiter (2008)

KIA (JU08KES01b) and RTF (JU08RTI01b) are *Porites* spp. coral cores collected in 2004 from Keswick Island and Round Top Island, respectively and were previously described in Jupiter (2006) (PhD thesis). Both corals' annual growth bands were bulk sampled and 3–5 consecutive years per decade were analysed for rare earth elements and Yttrium (REYs) for the period 1950–2002. Samples were analysed using a Thermo X-Series inductively coupled mass spectrometer (ICP-MS) at the University of Queensland.

### C2.5 Lewis (2005) – PhD thesis

MAG01D (LE05MAG01) is a modern *Porites* sp. coral collected from Geoffrey Bay, Magnetic Island by the Australian Institute of Marine Science (AIMS) in 1987, while NEL01D (LE05NEL01) and NEL03D (LE05NEL02) are fossil *Porites* spp. corals collected from Nelly Bay Harbour, Magnetic Island in 2001, and were previously described in Lewis (2005). Top and bottom growth bands for NEL01D and NEL03D were U-Th dated at the ACQUIRE laboratory at the University of 620 Queensland. Fragments (2 x 2 cm samples) of the bottom and top of MAG01D, NEL01D and NEL03D were platinum coated and examined for diagenesis using Scanning Electron Microscopy and additional thin sections were also examined with a Leica IM50 microscope. Analysis for diagenetic alteration of the corals noted no distinct differences between the modern and fossil corals. The corals were sampled by homogenising 5-yearly increments and were split into two batches. One batch was pre-treated with $H_2O_2$ and was analysed for Ba/Ca, Mg/Ca, Sr/Ca, and Mn at the Queensland Health Scientific Services (QHSS) 625 laboratory. The other batch was not pre-treated and was analysed for $\delta^{13}C$ and $\delta^{18}O$ with a Micromass PRISM III stable isotope mass spectrometer at the University of Wollongong.

### C2.6 Marshall (2000) – PhD thesis

Myr-2 (MA00MYR01) is a *Porites* sp. coral collected from Myrmidon Reef in July 1996 (Marshall & McCulloch 2002). As 630 the data was archived in the PhD thesis appendix without an age model, a new coral age model was created for archiving in the GBRCD. The new coral age model was based on published information from Marshall & McCulloch (2002) and Marshall (2000). The original data chronology and calibration was based on Australian Institute of Marine Science (AIMS) weekly instrumental Myrmidon Reef SST data. However, the AIMS Myrmidon loggers' SST data only covered the period from 1988 to 1996, whereas the Myr-2 Sr/Ca record extends back to 1973, and is at approximately monthly resolution (<20 measurements 635 per year). A new chronology was assigned to the data using QAnalySeries to align Sr/Ca minima (maxima) with SST maxima (minima) for HadISST grid square centred on 18.5° S, 147.5° E.

Myr2-16D (MA00MYR02) & Myr2-17D (MA00MYR03) are *Porites* sp. coral collected from the Myrmidon Hole 2 drill core matrix (Davies & Hopley 1983; Marshall 2000). Myr2-16D was analysed by Marshall (2000) for $\delta^{13}C$ and $\delta^{18}O$ at The Australian National University using a Finnigan MAT 251 mass spectrometer coupled with a Kiel device to assess the 640 fluorescent banding in the coral and its potential link with precipitation/river flow. The $\delta^{18}O$ data presented in Marshall (2000) have a correction of 0.011 ‰ per metre applied for ice volume that assumed sea level was 15–20 m below present-day sea level. The correction is not applied to the $\delta^{18}O$ data archived in the GBRCD. Myr2-17D was analysed for B, Mg, Ca, Sr, Ba and U by LA-ICP-MS using a 193 nm λ ArF excimer laser connected to a VG Elemental Plasmaquad PQ2+ instrument with a 50 x 500 µm slit as described in Marshall (2000). TIMS U-series dates were available for both coral pieces, however the data 645 for both corals did not include an age model. An age model for Myr-2-16D and Myr2-17D was created for archiving using QAnalySeries to match $\delta^{18}O$ or Sr/Ca maxima with generalised monthly SST minima (minimum SST occurring in July).

The Myr2-17D U-Th calibrated age published in Marshall (2000) was 7880 ± 60 years [ago]. However, no date for the U-series analysis is supplied. Thus, to determine the age relative to the Common Era for archiving it is assumed that it was dated in the same year as publication (i.e. 2000), thus transforming Myr2-17D age to -5881 ± 60 CE (i.e. 5881 ± 60 BCE).





Myr2-16D was in 3 parts and each section was previously U-Th dated separately giving a calibrated age range from 7646 ± 60 years [ago] for the basal section to 7968 ± 100 years [ago] for the top section (with a middle section dated as 7846 ± 30 years [ago]) (Marshall 2000). Density band counting of an alternate slice suggests a maximum of 20 years between the top and bottom of the coral piece. The U-series dates are not within error and/or suggest an age reversal in the Myrmidon Hole 2 drill core when considering the date of the top (youngest) section of Myr2-16D is older than Myr2-17D. In 2022 a piece of the middle section (~5–6 cm from the top of Myr2-16D) of the alternate slice was U-Th dated and used as the age of Myr2-16D for the GBRCD. The 2022 U-series date was measured at the Radiogenic Isotope Facility, University of Queensland and gave a calibrated age of -5755 ± 13 CE (i.e. 5755 ± 13 BCE; Table D1).

St-1 is a *Porites* sp. coral collected from Stanley Reef in January 1999 (Marshall 2000; Marshall & McCulloch 2002). The age model was created based on published information from Marshall & McCulloch (2002) and Marshall (2000). A chronology was assigned to the data for archiving in the GBRCD using QAnalySeries to align Sr/Ca minima (maxima) with SST maxima (minima) for fortnightly averaged AIMS in situ logger data from Davies Reef and Hardy Reef. The original publications only used the Davies Reef SST data, which due to missing data can only create an age model for the period of August 1996 to January 1999. The AIMS Hardy Reef SST data closely resembles the Davies Reef SST data, and both are approximately equidistant from Stanley Reef making them a suitable dataset to estimate the regional SST seasonality. Using both SST datasets enables the St-1 age model to be extended to cover the period from February 1995 to January 1999.

### C2.7 Razak et al. (2017)

Myra-30 (RA17MYR01) and myra-31 (RA17MYR02) are both *Isopora palifera* corals harvested from Myrmidon Reef in May 2013 (Razak et al. 2017). The age models were created for archiving in the GBRCD based on published information from Razak et al. (2017). A chronology was assigned using QAnalySeries to align Sr/Ca minima (maxima) with Myrmidon SST maxima (minima) for fortnightly averaged SSTs from a composite of 10 AIMS in situ loggers. Outliers were removed from the Sr/Ca records to match values reported in Razak et al. (2017). No outliers were removed from other measured trace element ratios.

### C2.8 Sadler et al. (2016b)

All modern and Holocene coral records from Heron Reef published in Sadler et al. (2016b) (SA16HER01b–SA16HER06b; modern: P HS 1, P HS 3, P HS 4 & P HS 7. Holocene: R9C, R15B3) were publicly archived with time series for all data points expressed as calendar year integers (i.e. all data from 2014 had an age listed as 2014). Therefore, to recreate the seasonal cycle (as presented in Sadler et al. 2016b) the chronology was assigned by tying Sr/Ca minima (maxima) to bimonthly averaged SST maxima (minima) using QAnalySeries. The original coral chronologies and Sr/Ca-SST calibration for the modern records were created using the Commonwealth Scientific and Industrial Research Organisation/Pacific Marine Environmental Laboratory Moored Autonomous pCO2 (MAPCO 2) project in the Wistari Channel (Sadler et al. 2016b), but this was not available for the GBRCD. For the GBRCD, modern SST data was averaged from AIMS in situ temperature loggers. The AIMS data loggers from the region have patchy temporal coverage, so the bimonthly SST dataset is an average of data logger data from Erskine Island (x1), Halftide Rocks (x1), Halfway Island (x2), Heron Reef (x16), Square Rocks (x2), and Tryon Island (x2). For the Holocene corals the SST data used to reconstruct the age model was a generalised seasonal cycle based on the AIMS bimonthly averaged SST data for the Southern GBR region (SST minimum occurs in July–August). Holocene corals were dated at the Radiogenic Isotope Facility, University of Queensland as described in Sadler et al. (2016b). However, no measurement date for U-Th analysis was published. To determine the coral ages relative to the Common Era the measurement date was determined using the Radiogenic Isotope Facility records (June 2015; Pers. comm. T. Clark).



### C2.9 Xiao et al. 2020

10AR2 (XI20ARL01) is a *Porites* sp. coral harvested from Arlington Reef in April 2010 (Deng et al. 2014). The age model was created for archiving in the GBRCD based on information published in Xiao et al. (2020). A chronology was assigned using QAnalySeries to align Sr/Ca minima with the start of each year (i.e. January) to obtain a record with 10–16 data points per year.

### C3 Unpublished Records

**C3.1 AR24DIP01**

Dip-04 (AR24DIP01) is an *Isopora palifera* coral that was harvested from Dip Reef in January 1988, which was in an existing collection held at the Australian Institute of Marine Science (AIMS). The data were measured by T. Razak but have not been published previously. As per the methods described in Razak et al. (2017), ~500 µg of coral powder was milled at 1 mm increments with a 1–2 mm depth using a hand-held drill with a 1 mm drill bit (Fig. C3). Trace elements (Ba/Ca, Mg/Ca, Sr/Ca,

U/Ca, and Y/Ca) were measured using a Thermo X-Series II quadruple ICP-MS at the University of Queensland Radiogenic Isotope Facility following a modified method described in Nguyen et al. (2013). The age model for this study was created by aligning Sr/Ca minima (maxima) with SST maxima (minima) using QAnalySeries. The seasonal cycle pattern was cross-checked with U/Ca data. As no nearby in situ instrumental measurements of SST exist for the period included in the coral record, chronology was established using the HadISST dataset (grid square centred on 18.5º S, 147.5º E). No outliers were

removed from the dataset.

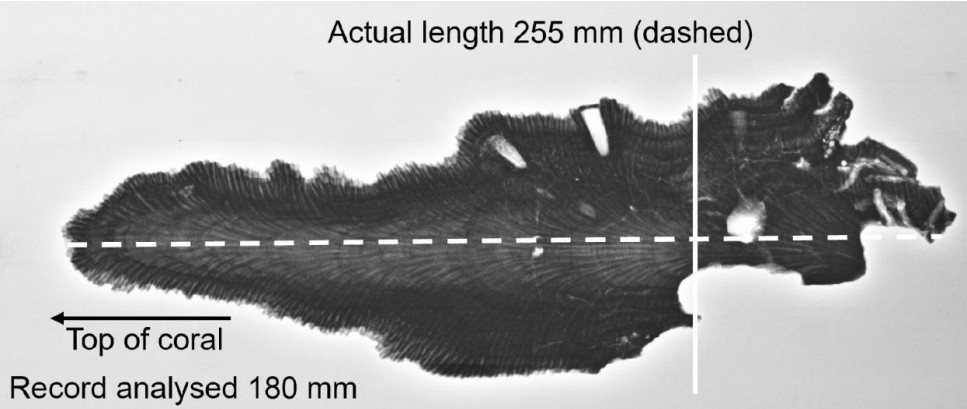

**Figure C3. X-radiograph (positive) of Dip-04 (AR24DIP01) coral. White dashed line indicates the track analysed with white solid line the end of record section. X-radiograph supplied by T. Razak.**


**C3.2 AR24OTI01**

GUT12 (AR24OTI01) is a *Porites* sp. coral that was harvested in May 2012 by J. Mallela from the Gutter in the One Tree Island lagoon. The data have not been published previously. GUT12 was previously sliced into ~7 mm thick slabs, and a section of GUT12 ~15 cm in length was selected for trace element analysis by A. Arzey (Fig. C4). This section was measured

using the AIMS densitometer and examined for diagenetic alteration using a Phenom XL benchtop SEM at the University of Wollongong by A. Arzey. Micro-borer holes and secondary aragonite crystals were present along most of the analysis track with patches of trace amounts of secondary aragonite crystal (<2 µm) increasing along the measurement track until 100% of skeleton was covered in secondary aragonite crystals generally between 0–10 µm length (Fig. C5). Overall diagenetic alteration





is low level, and the coral skeleton preservation is fair to good. Coral powder was drilled continuously from a ledge at 0.3–0.6

mm increments using a 2 mm TiN-coated end mill to give a resolution of ~17-34 (median 24) samples per year. Mg/Ca and Sr/Ca ratios were measured by inductively coupled atomic emission spectrometry (ICP-AES) at the Australian Nuclear Science and Technology Organisation (ANSTO) with a Thermo Scientific iCAP 7600 series ICP-AES. Data was standardised using measured JCp-1. The coral age model was created using QAnalySeries to align Sr/Ca minima (maxima) with SST maxima (minima) using the fortnightly averaged NOAA Reyn_SmithOIv2 weekly SST dataset, grid centred on 23.5º S, 152.5º E.

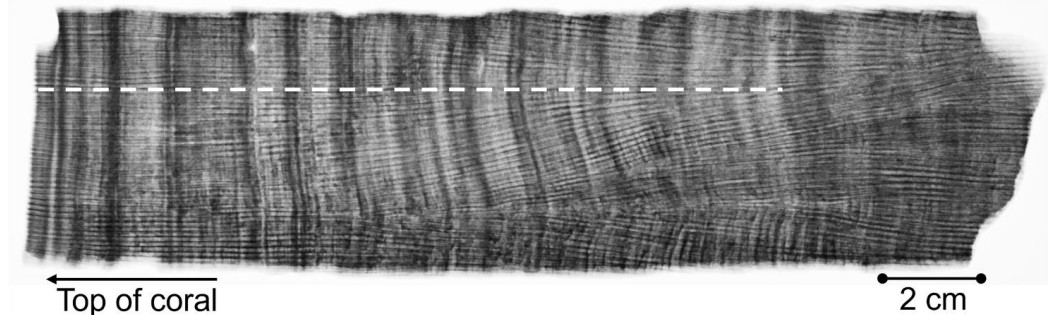


**Figure C4. X-radiograph (positive) of GUT12 (AR24OTI01) coral. Dashed white line indicates the track analysed by ICP-AES for the record included in AR24OTI01. X-radiographs by PRP Diagnostic Imaging Wollongong.**

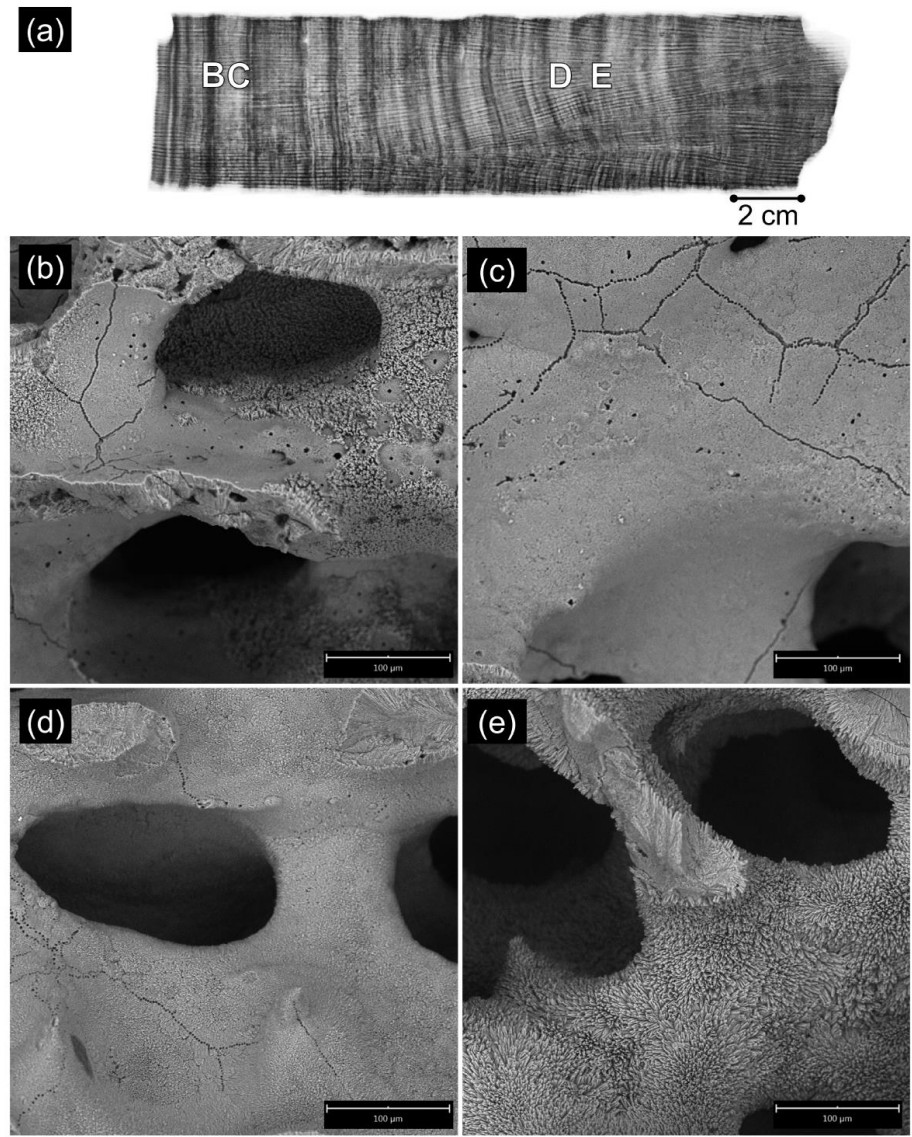

**Figure C5. GUT12 (AR24OTI01) SEM images (a) Approximate locations of SEM images labelled B–E on coral X-radiograph. (b) to (e) Select SEM images from GUT12; (b) Secondary aragonite needles <5 µm patchy cover of coral surface (~50%), (c) No secondary aragonite needles. Evidence of micro-borers, (d) low-level diagenetic alteration with 100% cover of secondary aragonite needles <5 µm, (e) low-level diagenetic alteration with 100% cover of secondary aragonite needles of 5-15 µm. Scale bar indicates 100 µm (b-e).**

### C3.3 AR24OTI02 & AR24OTI03

OTI 1-3 (AR24OTI02) and OTI 3-1 (AR24OTI03) are *Porites* spp. corals that were harvested in 2004 from the One Tree Island lagoon by G. Marion. The corals were part of the Enrichment of Nutrients Coral Reef Experiment (ENCORE) (Koop et al. 2001; Steven & Atkinson 2003), and were transplanted from the reef slope onto the reef flat on 17/01/1995 and stained with Alizarin Red S on 23/11/1995. OTI 1-3 (Fig. C6) was collected from the reference treatment patch reef at ambient levels of nutrients, while OTI 3-1 (Fig. C7) was from the enriched Nitrogen treatment patch reef. Both corals were measured by LA-





ICP-MS as per methods described in Lewis et al. (2012); corals were analysed using a Varian 820 ICP-MS, with output data smoothed with a 10-point running mean followed by a 10-point mean. The original age model supplied by S. Lewis (where chronology assigned with Sr/Ca minima (maxima) matched to February 8th (August 8th)) was adjusted to improve correlation with SST by fitting data to match the annual maxima and minima of a fortnightly averaged NOAA Reyn_SmithOIv2 weekly

SST dataset centred on 23.5º S, 152.5º E grid. Each piece was examined for diagenetic alteration by A. Arzey, although due to the absence of information for both pieces, the original LA-ICP-MS track cannot be directly matched to the area examined by the SEM. OTI 1-3 and OTI 3-1 show highly variable diagenetic alteration (Fig. C8 & Fig. C9). Both corals show above average levels of diagenetic alteration compared to other corals examined, which may be due to their handling for, or the altered nutrient exposure from, the ENCORE experiment. There is scope for future research to determine the cause of the

abnormal level of diagenetic alteration. Caution should be exercised if using OTI 1-3 and OTI 3-1 for climate assessment, due to the high levels of diagenesis and the corals' use in the ENCORE experiments.

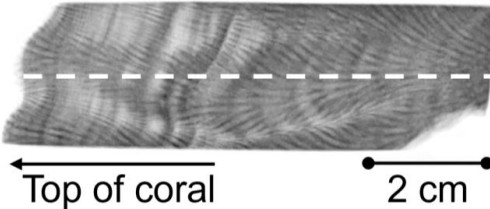

**Figure C6. X-radiograph (positive) of alternate slice of OTI 1-3 (AR24OTI02) coral. Dashed white line indicates the estimated track analysed by LA-ICP-MS for the record included in AR24OTI02. X-radiographs by PRP Diagnostic Imaging Wollongong.**


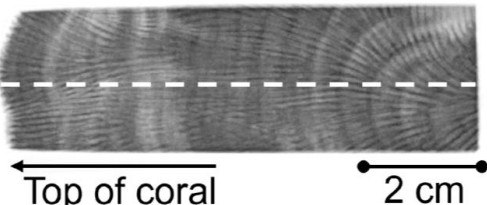

**Figure C7. X-radiograph (positive) of alternate slice of OTI 3-1 (AR24OTI03) coral. Dashed white line indicates the estimated track analysed by LA-ICP-MS for the record included in AR24OTI03. X-radiographs by PRP Diagnostic Imaging Wollongong.**

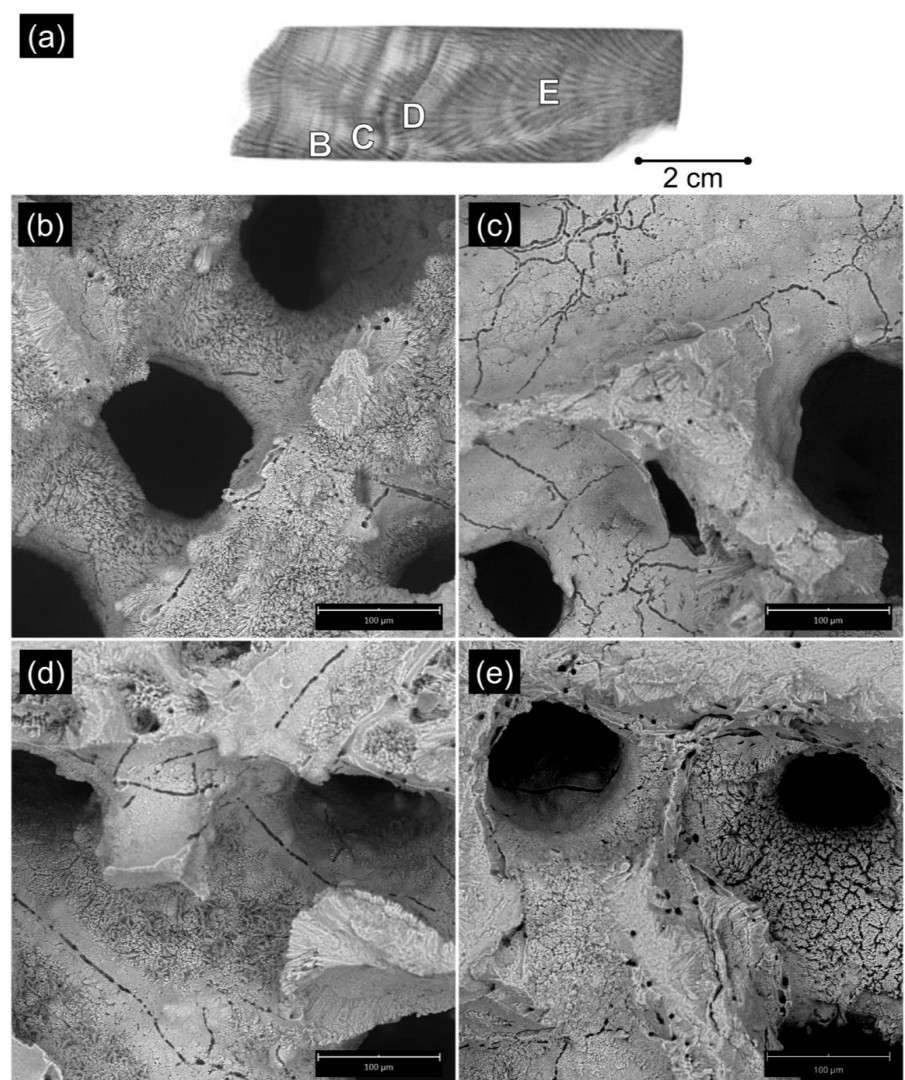


**Figure C8. OTI 1-3 (AR24OTI02) SEM images. (a)** Approximate locations of SEM images labelled B–E on coral X-radiograph. **(b) to (e)** Select SEM images from OTI 1-3; **(b)** low-level diagenetic alteration with 100% cover of secondary aragonite needles of 5–15 µm and micro-borer tracks, **(c)** No secondary aragonite needles, with evidence of dissolution and micro-borers tracks, **(d)** Evidence of micro-borer tracks and possibly dissolution. Secondary aragonite needles and calcite crystals (0–10 µm), **(e)** 90% surface covered
by secondary aragonite needles, left side of coral needles are ≤5 µm and right side of coral needles are ~5–15 µm. Evidence of micro-borers visible. Scale bar indicates 100 µm (b–e).





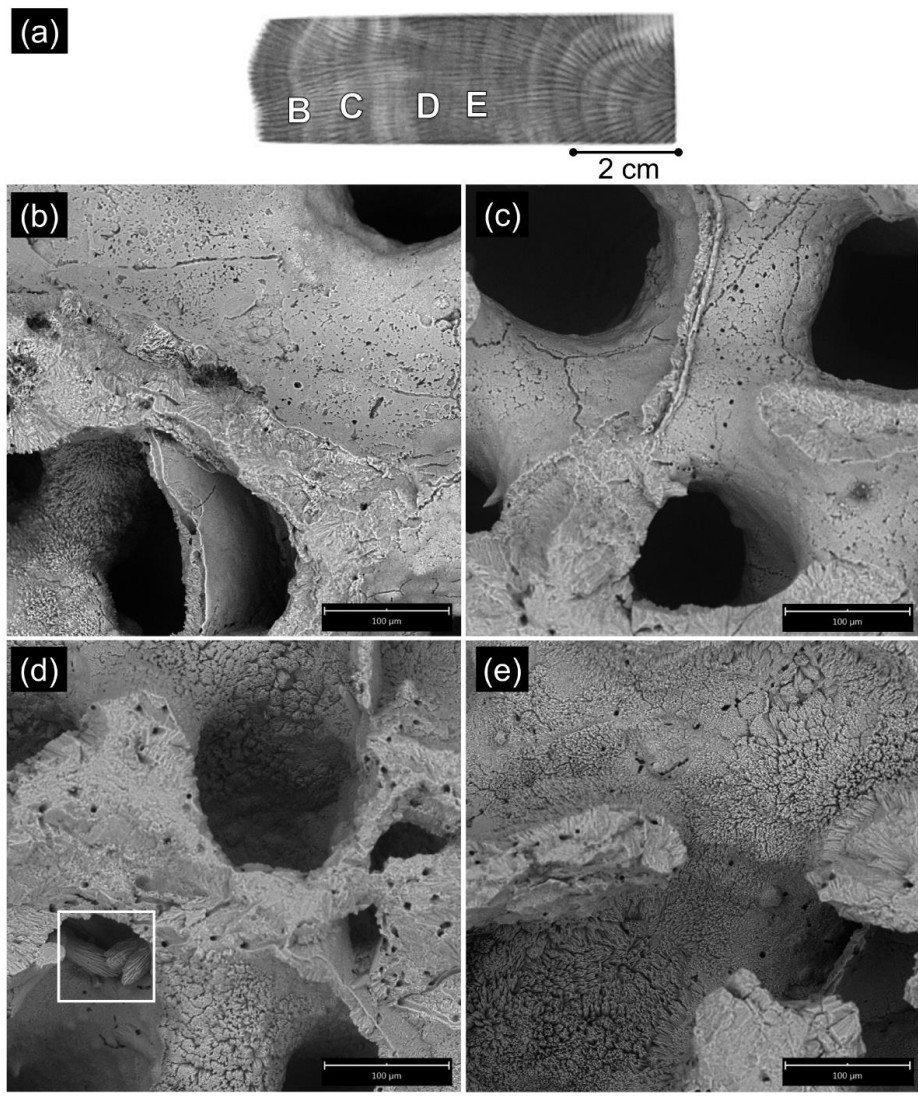

**Figure C9. OTI 3-1 (AR24OTI03) SEM images. (a) Approximate locations of SEM images labelled B–E on coral X-ray. (b) to (e)**
**Select SEM images from OTI 3-1; (b) Secondary aragonite needles (0–10 μm) in lower left of image and possible low Mg-calcite cement on coral surface in upper right of image. Evidence of micro-borers, (c) No secondary aragonite. Evidence of micro-borers and possible minor dissolution of coral surface, (d) 100% cover with secondary aragonite needles (0–15 μm) with evidence of micro-borers. High Mg-calcite splays occur in pore space; marked by white box, (e) 100% cover with secondary aragonite needles (~5–15 μm) with evidence of micro-borers. Scale bar indicates 100 μm (b–e).**

### C3.4 AR24OTI04 & AR24OTI05

OTI4_6C (OTI-4-19; AR24OTI04) & OTI4_6D (OTI-4-22; AR24OTI05) were collected from the One Tree Island hole 4 reef core matrix (Davies & Hopley 1983) at a depth listed as between 5.72–7.47 m (OTI Hole 4 section 6) and estimated to be 6.2 m and 6.5 m depth down core, respectively. Both corals were measured by LA-ICP-MS as per methods described in Lewis et al. (2012) and as per AR24OTI02 & AR24OTI03 above. An age model was created by matching the Sr/Ca maxima with SST
minima of a generalised SST time series (minimum SST occurs in August). The age model was further refined by adjusting for best fit of U/Ca maxima to SST minima. An off-cut of AR24OTI04 (Fig. C10) and alternate slice of AR24OTI05 were dated in July 2022 at the Radiogenic Isotope Facility, University of Queensland to give an age of -3803 ± 13 CE & -3772 ±





11 CE respectively (Table D1). An alternate slice of AR24OTI04 was also dated in 2022, but the age used for AR24OTI04 is based on the off-cut of the slice that was measured with LA-ICP-MS as the relationship with the section dated for the alternate

slice is not certain. SEM screening of AR24OTI04 coral off-cut indicated 100% cover of coral surface with secondary aragonite needles generally between 0–10µm, although small areas with crystals up to 20 µm were observed (Fig. C11).

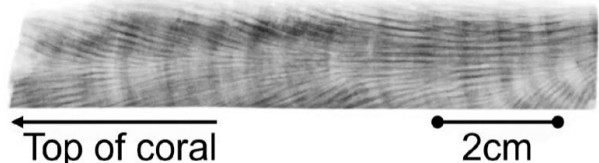

**Figure C10. X-radiograph (positive) of OTI4_6C (OTI-4-19; AR24OTI04) coral. Piece is off-cut of the coral piece measured by LA-ICP-MS for the AR24OTI04 record. X-radiographs by PRP Diagnostic Imaging Wollongong.**




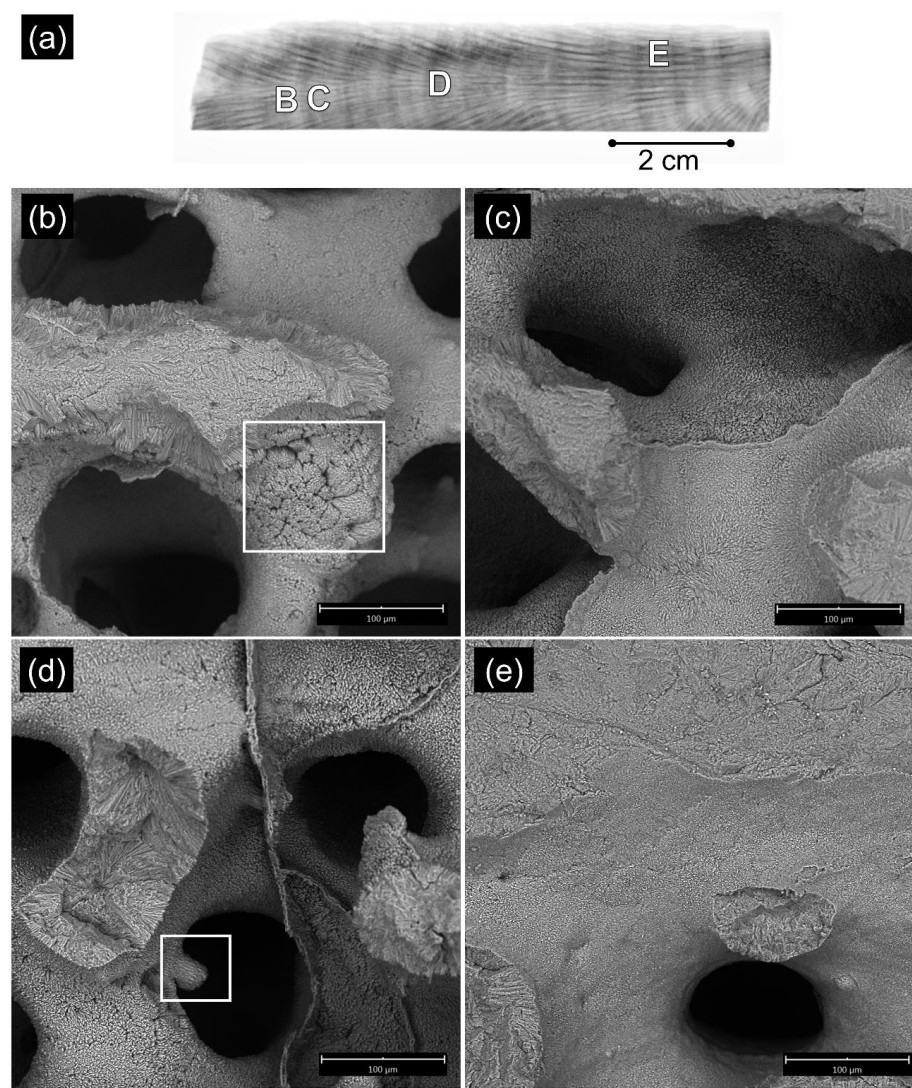

**Figure C11. OTI4_6C (OTI-4-19; AR24OTI04) SEM images. (a) Approximate locations of SEM images labelled B–E on coral X-ray. (b) to (e) Select SEM images from OTI4_6C (AR24OTI04); (b) 100% cover of secondary aragonite needles generally <10 μm, but section with uneven surface (marked by white box) indicates ~10–20 μm crystals, (c) 100% cover of secondary aragonite needles**
**of <10 μm, (d) 100% cover of secondary aragonite needles of 5–10 μm. Includes secondary aragonite covering likely fungal growth or part of borer tube that occurred when coral was alive (marked by white box), (e) 80% cover of trace (<5 μm) crystals of secondary aragonite. Scale bar indicates 100 μm (b–e).**

### C3.5 AR24SLY01

The SR coral (AR24SLY01) was measured for radiocarbon by S. Chakraborty and R. Bhushan in 1991 at the Physical Research
Laboratory in Ahmedabad, India. Annual bands from the SR coral were analysed as per methods described in Chakraborty (1993) for other coral samples. Briefly, the annual samples were cut from the slice using a <1 mm thickness small diamond wheel operated by an electric hand drill. The cut bands were dried by heating overnight at 80–90° C and 15g of powder used for the radiocarbon analysis. β-counting was carried out using a Packard Tri-Carb Liquid Scintillator Analyzer in low level count mode. The coral X-ray was published in Chakraborty et al. (2000).

## Appendix D – Uranium-Thorium Dating Information

**Table D1. Uranium-thorium age data obtained using a Nu Plasma II Multicollector Inductively Coupled Plasma Mass Spectrometer of fossil corals from the Great Barrier Reef, Australia. Note: AA47_OTI-4-19_4_Th and AA_OTI4_6C_1 samples are alternate slices of the same coral piece linked to the GBRCD record AR24OTI04. (See next page)**

Notes on Table D1:

Values in parentheses are activity ratios calculated from atomic ratios using decay constants of Cheng et al. (2000). All values have been corrected for laboratory procedural blanks. All errors reported as 2-sigma. Uncorrected $^{230}$Th age (ka) was calculated using Isoplot/EX 3.75 program (Ludwig 2003), where ka denotes thousand years.

To account for non-radiogenic or initial $^{230}$Th$_0$, two correction factors were applied:

a) A bulk-Earth correction assuming a non-radiogenic $^{230}$Th/$^{232}$Th$_0$ activity value = 0.825+/-50%, with $^{238}$U, $^{234}$U, $^{232}$Th and $^{230}$Th in secular equilibrium.

b) A two-component mixing correction value determined using the equation of Clark et al. (2014) that accounts for both detrital and hydrogenous $^{230}$Th/$^{232}$Th$_0$ shown in Eq. (D1):

$$\left(\frac{^{230}Th}{^{232}Th}\right)_{mix} = \left(\left(\frac{^{232}Th_{live}}{^{232}Th_{dead}}\right) \times \left(\frac{^{230}Th}{^{232}Th}\right)_{live}\right) + \left(\left(\frac{^{232}Th_{dead} - ^{232}Th_{live}}{^{232}Th_{dead}}\right) \times \left(\frac{^{230}Th}{^{232}Th}\right)_{sed}\right) \quad \text{(D1)}$$

Where $^{232}$Th$_{dead}$ is the measured $^{232}$Th value (ppb) in the individual dead coral sample. $^{232}$Th$_{live}$ is 0.77 ppb, being the mean $^{232}$Th value of 43 live *Porites* coral samples from the inshore Great Barrier Reef with a corresponding $^{230}$Th/$^{232}$Th$_{live}$ activity ratio of $1.066 \pm 0.063$ (20%) (Clark et al. 2012). $^{230}$Th/$^{232}$Th$_{sed}$ activity ratio representative of the insoluble Th component incorporated post-mortem or as particulates during growth is assumed to be $0.61 \pm 0.12$ (20%). This value is determined from the average y-intercept values of five $^{230}$Th/$^{232}$Th versus $^{232}$Th/$^{238}$U isochrons obtained from dead *Porites* corals sampled from the Palm Islands region (Clark et al. 2014).

The values in Table D1 are those determined after calculating the two-component correction. The 'Bulk-Earth corr. age (ka)' is supplied to indicate the difference in age calculated between the two corrections (~4–6 years).




| GBRCD ID | Lab Sample ID (Sample ID) | Cleaning Method | Date of chemistry | U (ppm) | 232Th (ppb) | (230Th/232Th) | (230Th/238U) | δ234U[a] | Uncorr. Age (ka) | Two-component corr. age (ka) | Bulk-Earth corr. age (ka) | Corr. Initial (234U/238U) | Two-component corr. age (CE) |
|---|---|---|---|---|---|---|---|---|---|---|---|---|---|
| AR24OTI04 | AA47_OTI-4-19_4_Th (OTI-4-19_4) | H$_2$O$_2$ | 4/7/2022 | 2.6683 ±0.0014 | 0.2241 ±0.00023 | 2153.51 ±3.93 | 0.05961 ±0.000010 | 1.1453 ±0.0016 | 5.8306 ±0.013 | 5.82445 ±0.013 | 5.8203 ±0.0013 | 1.1477 ±0.0017 | -3803 ±13 |
| AR24OTI04 | AA45_OTI4_6C_1_C (OTI4_6C_1) | H$_2$O$_2$ | 4/7/2022 | 2.6979 ±0.0013 | 0.04085 ±0.00012 | 11965.67 ±36.99 | 0.05971 ±0.000083 | 1.1457 ±0.0012 | 5.8386 ±0.010 | 5.8339 ±0.010 | 5.8301 ±0.0010 | 1.1481 ±0.0012 | -3812 ±10 |
| AR24OTI04 | AA73_OTI4_6C_1_U (OTI4_6C_1) | 2 x H$_2$O$_2$ | 5/7/2022 | 2.7023 ±0.0014 | 0.03442 ±0.000069 | 14267.90 ±34.66 | 0.05989 ±0.000088 | 1.1462 ±0.0013 | 5.8537 ±0.011 | 5.8491 ±0.011 | 5.8453 ±0.0011 | 1.1487 ±0.0013 | -3828 ±11 |
| AR24OTI04 | AA88_OTI4_6C_1_D (OTI4_6C_1) | Water | 5/7/2022 | 2.7092 ± 0.00085 | 0.03468 ±0.000086 | 14211.60 ±41.59 | 0.05996 ±0.000096 | 1.1456 ±0.0010 | 5.8564 ±0.011 | 5.8596 ±0.011 | 5.8556 ±0.0011 | 1.1481 ±0.0010 | -3838 ±11 |
| AR24OTI5 | AA46_OTI4_6D_1_Th (OTI4_6D_1) | H$_2$O$_2$ | 4/7/2022 | 2.6694 ±0.0016 | 0.03175 ±0.00010 | 15130.62 ±52.18 | 0.05931 ±0.00094 | 1.1456 ±0.00099 | 5.7985 ±0.011 | 5.7939 ±0.011 | 5.7901 ±0.011 | 1.1480 ±0.0010 | -3772 ±11 |
| MA00MYR02 | AA61_MYR2_16D_1 (MYR2_16D_1) | H$_2$O$_2$ | 5/7/2022 | 2.4849 ±0.0012 | 0.07492 ±0.000079 | 7932.69 ±12.56 | 0.07882 ±0.00010 | 1.1444 ±0.0010 | 7.7815 ±0.013 | 7.7762 ±0.013 | 7.7698 ±0.013 | 1.1477 ±0.0011 | -5755 ±13 |

a – $\delta^{234}U \left(\frac{^{234}U}{^{238}U}\right) \times 1000$ (D2)



## Appendix E – GBRCD Data Resources

AIMS loggers: derived data based on Australian Institute of Marine Science data

https://data.aims.gov.au/aimsrtds/datatool.xhtml

Hadley Centre Sea Ice and Sea Surface Temperature (HadISST) (Rayner et al. 2003)

https://coastwatch.pfeg.noaa.gov/erddap/griddap/erdHadISST.html

NOAA REYN_SmithOIv2 (Reynolds et al. 2002)

http://iridl.ldeo.columbia.edu/SOURCES/.NOAA/.NCEP/.EMC/.CMB/.GLOBAL/.Reyn_SmithOIv2/

Reefs and shoals – Queensland: Queensland Department of Resources

© State of Queensland (Department of Resources) 2023

http://qldspatial.information.qld.gov.au/catalogue/custom/search.page?q=%22Reefs and shoals - Queensland%22

Drainage basins – Queensland

© State of Queensland (Department of Resources) 2023

http://qldspatial.information.qld.gov.au/catalogue/custom/search.page?q=%22Drainage basins - Queensland%22

Major watercourse lines – Queensland

© State of Queensland (Department of Resources) 2023

http://qldspatial.information.qld.gov.au/catalogue/custom/search.page?q=%22Major watercourse lines - Queensland%22

Populated places – Queensland

© State of Queensland (Department of Resources) 2023

http://qldspatial.information.qld.gov.au/catalogue/custom/search.page?q=%22Populated%20places%20-%20Queensland%22

Mainland – Queensland

© State of Queensland (Department of Resources) 2023

http://qldspatial.information.qld.gov.au/catalogue/custom/search.page?q=%22Mainland%20-%20Queensland%22



## Author Contributions

AKA & HVM conceived the study and designed the database. AKA built and curated the database, and database testing was conducted by AKA & HWF. TRC contributed data analysis and advice for U-Th dating. AKA, SEL, JM, JMW, SC & TBR provided coral data and associated technical information. NPM facilitated converting the database to LiPD format, and AKA & HWF provided example scripts for using the database in R and Python. AKA & HVM wrote the manuscript with input from TRC, JMW, SEL, JM, NPM, HWF, SC, TBR & MJF.

## Competing interests

The authors declare that they have no conflict of interest.

## Acknowledgements

We would like to thank all the researchers whose publicly archived data or whose data were archived in supplementary material were included in the GBRCD, as well as the researchers who supplied us with their data (Appendix A).

We are thankful for the software and data sources that made this project possible. Sea surface temperature data (HadISST1.1) were provided by Met Office and accessed from https://coastwatch.pfeg.noaa.gov/erddap/info/erdHadISST/index.html.

Figure 1 was generated using QGIS 3.22.11 (https://www.qgis.org) with GIS layers sourced from State of Queensland (Department of Resources) found at http://qldspatial.information.qld.gov.au/catalogue/. The full list of GIS layers is available in Appendix E.

Figures 3–5 were generated using R 4.3.1 (https://www.R-project.org/) and the ggplot2 3.4.3, sf 1.0-14 and ozmaps 0.4.5 R packages found at https://cran.r-project.org/web/packages/ggplot2/index.html, https://cran.r-project.org/web/packages/sf/index.html & https://cran.r-project.org/web/packages/ozmaps/index.html.

We would also like to thank the NOAA NCEI, especially Edward Gille, and LiPD teams for facilitating archiving and distribution of the GBRCD.

JM's fieldwork was undertaken with the permission of the Great Barrier Reef Marine Park Authority (GBRMPA Permit Number G12.35021.1) and One Tree Island Research Station.

## Financial Support

This research was funded by an Australian Government Research Training Program scholarship. This research was supported by an AINSE Ltd. Post Graduate Research Award (PGRA) (AKA). This research was funded by an Australian Research Council (ARC) Discovery Project DP200100206 to HVM, JMW and TRC. This work was supported by ARC SRIEAS Grant SR200100005 Securing Antarctica's Environmental Future and ARC Future Fellowship (FT140100286) to HVM. TRC was funded by an ARC Discovery Early Career Researcher Award (DECRA) DE180100017. JM was funded by an ARC DECRA. DE120101998.

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
