# Peer review of "Coral Skeletal Proxy Records Database for the Great Barrier Reef, Australia"

_Earth System Science Data, 2024_

## Author Comment (AC1)

Dear Editor and Reviewers,

We would like to thank you for your support and constructive review of our manuscript. We have carefully considered the comments and outline below our responses and manuscript revisions. To address these comments, we have updated the manuscript text and updated the GBRCD database to better convey that the database is open, and database fields have been created or amended to accommodate future updates. We have made minor edits to further improve the readability of the manuscript.

Following, please find our detailed responses to the reviewers' comments.

Authors' responses are shown in blue.

**Reviewer #1** – Niels de Winter**

Dear Sebastiaan van de Velde, dear authors,

As requested, I read and reviewed the manuscript titled "Coral Skeletal Proxy Records Database for the Great Barrier Reef, Australia" submitted by Ariella Arzey and colleagues for publication in Earth System Science Data. In their manuscript, the authors present and discuss the new online Great Barrier Reef Coral Skeletal Records Database (GBRCD) which compiles a large quantity of data gathered in studies into skeletal coral records in the Great Barrier Reef in Australia. The main aim of this study, and the database, is to compile this proxy data from various sources to make it available and machine readable for future research projects. Given the wide distribution of this type of data in different formats and in various repositories and supplements, this database presents an important effort to making the hard work of coral researchers more easily available for meta-analyses and comparison studies.

The authors provide a clear introduction explaining the need for a comprehensive database of coral records in the Great Barrier Reef and a thoughtful background section on the reef setting and the types of proxies contained in the dataset. A small point of feedback would be that it is not immediately clear that this is an open dataset and that more data can be added. However, this becomes clear at the end of the manuscript. The two formats of the database (LiPd + CSV) ensure ease of access of all the data both through manual downloading and data processing and by machine reading. Figures 3-5 give a nice sneak peek into the spread and location of geochemical records in GBR.

In conclusion, I believe this manuscript and the database it presents represent a valuable contribution to the field of (coral) sclerochronology and paleoclimatology and I would be happy to support its publication more or less as is.

Thank you for your positive comments on our manuscript. We greatly appreciate your feedback and your willingness to support its publication. We have clarified throughout the manuscript that the GBRCD is an open database with future updates planned. In addition, we have added a sentence to the abstract making this point:

*"The intention is to update the GBRCD annually, depending on the availability of relevant new GBR records or submission of legacy records to the GBRCD for archiving."*

Below, I detail a few minor textual comments I had while reading through the manuscript, but beside these I think the manuscript is in pretty good shape.

We have addressed the minor textual suggestions you provided as follows:

**Minor comments**

Line 152: "The GBRCD is along the lines of" should probably be rephrased to something like "The aim of the GBRCD is similar to"

This sentence has been amended to read:

*"The GBRCD is similar to the Coral Trait Database (Madin et al., 2016), but focuses on the GBR and coral proxy measurements (i.e. geochemical and luminescence measurements)*

Line 157: The past tense in this criterion and others suggests that the database is done and not a living product. Perhaps this can be rephrased.

References to past tense of criteria have been amended to:

> "▪ *Length. The records are continuous (multi-year) measurements from coral skeletons.*
>
> ▪ *Age model. Each record includes a relevant chronology (i.e. an estimate of age as the independent variable) or has a chronology that could be recreated relatively easily based on available data."*

Line 208: It is unclear to me how a Mg/Ca ratio can have a value <0, since it is a ratio of two concentrations. Perhaps the authors can use another example of "abnormal data".

To address the issue of the abnormal data example, it was decided to clarify the sentence rather than replace the example as it is indicative of clearly abnormal data across fields of coral geochemistry research and emphasises the following suggestion that it is "up to the user to determine appropriate screening procedures". The example has been amended to:

*"Potentially abnormal data (e.g. values <0 for Mg/Ca [after correcting against a method and/or lab standard]) were not removed from the datasets as they may be indicative of climate or environmental events, and it is up to the user to determine appropriate screening procedures when using the database."*

Best wishes,

Niels de Winter

---

## Author Comment (AC2)

Dear Editor and Reviewers,

We would like to thank you for your support and constructive review of our manuscript. We have carefully considered the comments and outline below our responses and manuscript revisions. To address these comments, we have updated the manuscript text and updated the GBRCD database to better convey that the database is open, and database fields have been created or amended to accommodate future updates. We have made minor edits to further improve the readability of the manuscript.

Following, please find our detailed responses to the reviewers' comments.

Authors' responses are shown in blue.

**Reviewer #2**

**General comments**

This manuscript provides details on a new coral geochemistry database for the Great Barrier Reef, the first for a particular location and the most studied reef in the field of coral-based paleoclimatology. The paper is well-written and comprehensive and the database is easy to use and well-documented. This database will be useful to coral and climate researchers, especially those working with climate models for many decades. The authors intend to update the database annually and provide a contact email on the GitHub page but not a person to contact. It would be useful to have this information in the manuscript and NOAA Paleo website. Additionally, please add details for continued maintenance at the institution (AIMS)? or PI level so data can be submitted 10 to 20 years from now. There should also be a mechanism for corrections to be database to be submitted to those maintaining the database.

Thank you for your detailed review of our manuscript. We appreciate your support for its publication and are pleased you found the database easy to use.

We opted for a general email address (GBRCDsubmissions[@]outlook) so that updates are not dependent on a single person, and that everyone in the GBRCD team can add any new submissions to the database. The general email address will facilitate including new team members, whether or not they are based at the University of Wollongong. We envisage that corrections to the database can also be submitted via the same email address. At present updates to the GBRCD will be facilitated by the team at the University of Wollongong, and the data are hosted at NOAA/WDS Paleoclimatology. We have added the following text to the discussion and database availability sections to clarify these points:

*"An email address for submissions to the GBRCD is provided at the GBRCD GitHub and NOAA/WDS Paleoclimatology study page to facilitate capture of legacy or new coral data."*

*"For researchers that want to submit records (published or unpublished) to the GBRCD, please check the GBRCD GitHub for details on how to submit your records for the database via the email address available on the GBRCD GitHub and NOAA/WDS Paleoclimatology study page."*

**Specific comments**

Why just coral records to the Holocene? What about the IODP coral records and other GBR coral records that go that further in time? (e.g., Felis 2022 doi:10.1029/2021GL096495, Brenner 2020 doi:10.1029/2020PA003962, Brenner 2017, doi:10.1002/2016PA002973, Felis 2014 doi:10.1038/ncomms5102). perhaps these could be added in the next update to the database.

Thank you for your suggestion of additional records to be included in the GBRCD for future updates. Brenner 2017, doi:10.1002/2016PA002973, records are from modern corals and are currently included in the database as 'BR17HER01' to 'BR17HER05'. Felis 2022, doi:10.1029/2021GL096495, Felis 2014 doi:10.1038/ncomms5102, and Brenner 2020 doi:10.1029/2020PA003962, do not meet the second criteria for inclusion in the GBRCD: *"Length. The records are continuous (multi-year) measurements from coral skeletons."*

However, these records could be considered in future updates of the GBRCD if revising/updating the GBRCD criteria.

The authors also included growth data (extension per year, density, calcification) in the database, please mention this in the introduction. These parameters help understand the coral geochemical data and should be archived together. A repository of X-radiographs from the corals and cores would also be useful, but this reviewer understands hosting the large file sizes is an issue but should be included as part of a permanent archive or a next update project.

We have added a reference to growth data, as suggested, in the introduction and updated section 2.1 Selection criteria for inclusion in GBRCD:

*"This study presents a comprehensive database of GBR coral geochemical and luminescence records, and associated variables such as growth characteristics (where available), compiling both published records (201), and records published here for the first time (7 records)."*

*"Where available, additional variables, such as coral growth data (density, extension, etc.), is included alongside the geochemical and luminescence data for records in the GBRCD where the additional variable data was at the same resolution as the geochemical or luminescence measurements."*

We agree that a repository of X-radiographs from the coral cores would be useful, however, except for the x-radiographs we have produced and included in the appendix, we do not have the physical/digital images to archive. The coral scans for records are either included as a figure or appendix in the original publication or are not available. We would like to encourage authors to archive their scans (x-radiographs, computed tomography, etc.) in publications' supplemental material. As noted in the manuscript discussion, as a first step it is essential that authors publish their scans alongside their publications when submitting for publication. We have revised our discussion sentence to be more specific on this point:

*"X-radiographs showing sampling paths should also be routinely published with the data, either as a figure in the paper, or as supplementary material."*

The authors note that we are losing valuable coral data as corals become increasingly more threatened, which is true. However, we are also losing coral data as researchers retire or computer systems age, and thus data and paper files are lost, unreadable on newer computers, or destroyed. Rescuing these data sets is also important and may require transcribing paper files to digital formats. I would encourage the authors to note this as well and may be used to persuade institutions to save these important files as well as funding agencies to support these efforts.

We agree that it is important to save these files, and it is one of the motivating factors for creating the GBRCD. In addition to our current statement in the introduction "research data are being lost to time", which covers the examples listed by the reviewer, we have added a discussion paragraph on the topic and included an additional statement in the conclusion:

*"With the continued passage of time, it is likely there will be further loss of existing coral records and associated data. Due to physical misplacement or destruction of data files, inability to read old computer files and/or retirement of researchers with existing records, the coral data underpinning early coral research may be lost forever. We urge researchers in possession of existing records to reach out to archival teams such as those sustaining existing databases (e.g. the GBRCD or CoralHydro2k database (Walter et al., 2023)), to ensure records are archived for future use. An email address for submissions to the GBRCD is provided at the GBRCD GitHub and NOAA/WDS Paleoclimatology study page to facilitate capture of legacy or new coral geochemical data."*

*"The GBRCD promotes efforts to ensure important research data is archived and that existing coral records are not lost to time."*

I would also encourage the authors to make physical coral cores and their location a part of the database or a separate database, with International Generic Sample Numbers (IGSN) (Dassie et al., 2017). The IGSN numbers should be included in the metadata as well.

We agree that future updates should include IGSNs if available. To facilitate this, we have added another metadata field "cdata_IGSN", included a description in Table 5, and updated section 2.7.1 Identifier Metadata and Table B1.

However, IGSNs were introduced after many records in the GBRCD were published and so there is no associated IGSN for these records. Furthermore, we do not have physical possession of the samples included in the database and so are unable to appropriately register the samples. We have added a sentence to the discussion encouraging authors to assign and report IGSNs for their samples:

*"The addition of IGSNs to capture the metadata associated with the physical samples would also improve the utility of the coral data (Dassié et al., 2017)."*

Regarding diagenesis in the discussion section. There are best practices for detecting diagenesis that are well rooted in geology including thin sections, SEM, and XRD as well as X-radiographs, and these techniques are taught at the undergraduate and graduate level in geology. Many studies have urged researchers to assess their corals for diagenesis, see the study of Quinn et al., 2006 suggested that "...practice of publishing coral geochemical records, especially those that depict abrupt and/or large changes in tropical SST, without additional documentation of the pristine nature of corals used in their study should be

avoided." See the recent paper of Weerabaddana et al., 2024 https://doi.org/10.1029/2023PA004730. Yes, this should be standard practice and researchers should be and are trained to detect diagenesis in their corals and reconstructions, and in this reviewer's experience, it has been done in the last 20 years. It may not be mentioned in the main text but is given in supplemental files. Some short-form journals (Nature, Science, GRL, Geology, etc.) do not give authors the space to add these details to the paper so they get left out. This does not mean the authors did not screen for diagenesis.

Detecting diagenesis should be routine and our authorship team have advocated for this (e.g. McGregor and Gagan 2003, McGregor and Abram 2008). It was part of our motivation for including diagenetic screening as a field in the database. However, only 40 of 208 records included in the GBRCD mention their diagenetic screening methods. This number was reached after detailed reading of publications' main text, supplemental material, as well as cross-referencing sample IDs between publications. Potentially more than 40 records were screened for diagenesis, however, this is not guaranteed, and given the consequences of diagenesis for palaeoclimate and palaeoenvironment reconstructions from coral records, we do not make any assumptions. By flagging the importance of screening for diagenesis in the GBRCD discussion, we aim to prompt future authors to ensure they include diagenetic screening details in their publications (either in the main text or supplemental materials).

Regarding the methods for detecting diagenesis, we have updated the section to include the words *"established standard"* as the practice utilised varies between researchers when screening their coral samples for diagenesis.

As a result of the reviewer's comments, we have revised the discussion paragraph as follows:

*"Another area for improvement is the reporting of screening corals for diagenesis. Diagenesis is a known source of error in geochemical analysis as it can remove the primary coral proxy signal (Hendy et al., 2007; Sayani et al., 2011; Mcgregor and Gagan, 2003; Nothdurft et al., 2007; Quinn and Taylor, 2006; Weerabaddana et al., 2024). Only 40 of 208 records in the GBRCD reported on diagenetic alteration. The majority fail to mention the screening method(s) or outcomes (see meths_archiveDiagenesisCheck) in the publication text, supplementary materials or could be determined from cross-referencing sample IDs in other publications. There is no established standard of best practice for screening corals for diagenesis at present, though it is suggested that it is ideal to use petrographic analysis with a combination of methods such as XRD, and densitometry (Mcgregor and Abram, 2008). Other emerging approaches (e.g. Murphy et al. 2017; Takada et al., 2017) may also be useful for diagenesis screening. However, due to the strengths and weaknesses of the various methods as noted by Nothdurft and Webb (2009), examination by petrographic analysis (preferably using scanning electron microscopy) is required to confirm the type and level of diagenetic alteration (if any) present in a sample. Given the consequences of diagenesis for palaeoclimate and palaeoenvironment reconstructions from coral records we strongly advocate for authors to report their diagenetic screening methods in their publications or supplements."*

This reviewer likes the comment in the discussion about the minimum metadata that should be included in a publication. I hope journals start to insist on these metadata for coral paleo papers. Additionally, this reviewer appreciates the additional information included in the Appendices.

The conclusions section is redundant and not needed, please remove it.

The conclusion is included as per ESSD manuscript requirements (https://www.earth-system-science-data.net/submission.html#manuscriptcomposition).

We have addressed the comments and technical corrections you have suggested as follows:

**Technical corrections, typing errors**

Figure 1 Are the selected rivers those studied by research in the database or just author preference? If they are studied rivers, perhaps rephrase "select" to "studied".

Rivers included in the figure are selected for reference and are not all the rivers that have been studied. We have therefore retained *"select"* in the caption.

Line 118 Revise "…as they incorporate trace elements, varying isotopes, and organic materials in proportion to environmental variations…" Each element has several isotopes and it is the ratio (e.g., $\delta^{18}O$ and $\delta^{13}C$) or absolute amount (e.g., $^{14}C$, $^{239+240}Pu$) that are used for climate and environmental proxies whereas other isotopes are used for dating ($^{230}Th$).

This line has been revised to:

*"Coral skeletons provide in situ records of the reef environment as they incorporate trace elements, a variety of element isotopes, and organic materials in proportion to climate and environmental variations in the marine environment."*

Line 128 Revise "…and were at the forefront of the development of new technology and methods to measure these coral properties."

Line has been amended to:

*"Beginning in the late 1960s, studies using GBR corals provided empirical evidence of a relationship between skeletal oxygen isotopes, luminescence, trace elements, radiocarbon and density and the coral's marine environment (Weber and Woodhead, 1969; Weber and Woodhead, 1970; Weber and Woodhead, 1972; Isdale, 1984; Druffel and Griffin, 1993; Lough and Barnes, 1990; Mcculloch et al., 2003) and were at the forefront of the development of new technology and methods to measure these coral properties (Isdale et al., 1998; Gagan et al., 1998; Sinclair et al., 1998; Barnes et al., 2003; D'olivo et al., 2018)."*

The data files for one Tree Island and Dip and Stanley Reefs on the NOAA Paleoclimatology website are tab-delimited, not CSV files as stated in the manuscript lines 254, 198, 191, 73, 34, etc.). CSV files use commas to separate values, not tabs. "# Data line format - tab-delimited text, variable short name as header"

The entire GBRCD is available as CSV and LiPD formatted files at the NOAA study page. Each version of the complete database is available as a zipped folder (.zip files). The reviewer described tab delimited files (supplied as .txt files on the NOAA GBRCD study page) are the standard NOAA files for archiving records, and are only available for the new

(not previously published) records on the associated GBRCD study page. Those tab delimited files are additional to the CSV (and LiPD) database described in the manuscript. To clarify, we have included the following sentences in section 2.2 Data Sources:

*"As part of the GBRCD compilation process, 114 previously unarchived records, as well as the seven previously unpublished records, were submitted to NOAA/WDS Paleoclimatology for archiving. These files are in the standard NOAA/WDS Paleoclimatology (tab-delimited) file format."*

Line 257 and elsewhere (not every instance is noted, use find and replace). Revise "between" to "among". Between is used for two items, among for more than two items.

We consulted the Merriam-Webster and Oxford dictionaries on the use of "between" vs "among", and while some may prescribe that "between" is only for two items, the dictionary definitions note that it is between two or more items, locations, etc. Common syntax usage makes "between" a more familiar and comprehensible choice of word for comparisons. In a select number of cases "between" has been replaced by "among" where it does not seem unduly out of place in the reading of the sentence.

Line 293 Spell out maximum and minimum.

Maximum and minimum have been spelt out in full.

Table 3 and text description of "Age" lines 266-270. Are the monthly data given as the middle of the month or the beginning of the month? After looking at several of the data files, the age or time step varies by record. Some records are resampled to even monthly time steps and others are not. This reviewer assumes the authors are using the original age given by the original authors. Please clarify this in the text and note that some users may need to resample or linearly interpolate to get data on the same time scale.

A description in 2.4 Data Processing describes circumstances where changes have been made to original age models, and is continued in Appendix C.

To ensure it is clear, additional sentences have been added in the text before Table 3:

*"For sub-annual resolution records the tie-point within each month varies depending upon the researchers' choice when creating their age models or interpolating their data. It is up to researchers to inspect the data to understand the 'Age' field for each record they intend to use."*

Table 7 geo_longitude Description. "All values are positive due to the GBR's location east of the Prime Meridian." This could be confusing, in general, longitude east is positive and east is negative, it has nothing to do with GBR's location. Please revise. Same for geo_latitude.

Our description of geo_latitude and geo_longitude follows the example set by CoralHydro2k (Walter et al. 2023). Additionally, it is the GBR's location, which is within 180$^{\circ}$ east of the Prime Meridian, that determines that all coral records in the GBRCD have a positive longitudinal value. Likewise, for latitude, the GBR's location south of the equator determines that all records in the GBRCD have a negative latitudinal value. This wording has been clarified in the geo_latitude and geo_longitude field descriptions in Table 7.

Table 11 Is there a field for records that have been resampled to even time steps? This reviewer did not see this mentioned in the text either.

Thank you for pointing out the oversight. It is likely that sub-annual measurements that have been resampled/interpolated to even time steps will not include the "_uneven" suffix (also in the CoralHydro2k database). Additionally, information on interpolation was previously included in the 'meths_chronologyNotes' field. To clarify this, we have updated the description of the "meths_chronologyNotes" field in Table 9 to:

*"Field includes the target SST dataset and interpolation information if supplied."*

Tables 9 and 12 The authors use 1 and 2 sigma for analytical precision, most likely given by the original authors. Why not make it easier and just make all precisions 1 sigma as in previous databases (Iso2k and Coralhydro2k)? This will help users who may not understand the differences.

As per your suggestion to match precision with other databases we have used 1 deviation for all analytical precision values. To achieve this, we have now halved the 2 standard deviations to only report 1 standard deviations for this field (also done in the CoralHydro2k database) and have updated the appropriate text in Table 9 and Table 12.

Table 13 calib_SSTdata and text. It would be useful to also include version numbers especially for ERSST since each version can be quite different. Additionally, OISST has different resolutions that would be useful to know.

calib_SSTdata includes version numbers in the record metadata if this was supplied in publications. The description in Table 13 provides a summary of the acronyms. To clarify the description of calib_SSTdata we have added:

*"SST dataset version information is included if the information was available."*

Figure 3 This is a nice figure but panels A and C are hard to read, perhaps switch to a log scale so the fossil corals can be seen for their group and resolution.

We have not adopted a log scale for the y-axis (number of records) to retain the visual representation of lack of data prior to 1500 CE. However, to make it easier to see the pre-1500 records, we have included an inset in Figure 3 a & c with a maximum y-axis value of 6. The related figure captions have been updated.